# Photoacoustic Tomography with Temporal Encoding Reconstruction (PATTERN) for cross-modal individual analysis of the whole brain

Yuwen Chen[1,2,15], Haoyu Yang[3,4,5,15], Yan Luo [1,2,15], Yijun Niu [3,4,15], Muzhou Yu[6], Shanjun Deng [7], Xuanhao Wang[8], Handi Deng[1,2], Haichao Chen[9], Lixia Gao [10], Xinjian Li [10], Pingyong Xu [11,12], Fudong Xue[11], Jing Miao[13], Song-Hai Shi[3,4,5], Yi Zhong[3,4,5], Cheng Ma [1,2] ✉ & Bo Lei [3,4,14] ✉

Cross-modal analysis of the same whole brain is an ideal strategy to uncover brain function and dysfunction. However, it remains challenging due to the slow speed and destructiveness of traditional whole-brain optical imaging techniques. Here we develop a new platform, termed Photoacoustic Tomography with Temporal Encoding Reconstruction (PATTERN), for non-destructive, high-speed, 3D imaging of ex vivo rodent, ferret, and non-human primate brains. Using an optimally designed image acquisition scheme and an accompanying machine-learning algorithm, PATTERN extracts signals of genetically-encoded probes from photobleaching-based temporal modulation and enables reliable visualization of neural projection in the whole central nervous system with 3D isotropic resolution. Without structural and biological perturbation to the sample, PATTERN can be combined with other whole-brain imaging modalities to acquire the whole-brain image with both high resolution and morphological fidelity. Furthermore, cross-modal transcriptome analysis of an individual brain is achieved by PATTERN imaging. Together, PATTERN provides a compatible and versatile strategy for brain-wide cross-modal analysis at the individual level.

In brain research, cross-modal analysis is becoming increasingly important to deepen our understanding of brain function and malfunction. Despite the substantial improvement in the imaging capabilities, the development of a whole-brain imaging system that is highly compatible with other modalities has been overlooked. Current whole-brain optical imaging techniques can visualize the distribution of neurons and biomolecules, neural circuits, and patterns of neuronal activity[1–8]. Specifically, the sectioning tomographic techniques such as fluorescence micro-optical sectioning tomography (fMOST)[1,4], serial two-photon tomography[6], and block-face serial microscopy tomography (FAST)[5], enable brain-wide fluorescence imaging at single-neuron resolution[3]. Light-sheet fluorescence microscopy (LSFM) coupled with tissue clearing methods is another effective approach to achieve whole-brain fluorescence imaging with higher imaging speed[2,7,9].

Although these optical imaging strategies are widely used in neural circuitry studies, they have serious limitations. For a single mouse brain, these methods usually take more than one week for sample preparation, processing, and imaging[5,10], which limits the overall throughput and challenges these methods from scaling up to

image larger animal models such as rats, ferrets, and non-human primates. More importantly, the likelihood of either altering the structure or biological state of the sample or completely destroying it during the sample preparation and imaging processes makes it challenging for researchers to conduct precise geometric and structural analysis and constrains subsequent measurements, such as metabolomic, genomic, and transcriptomic analyses[3,5,9,11]. While parallel measurements of different samples have enabled whole-brain high-dimensional integrated atlases in homogeneous and standardized animal models[12,13], analyzing cross-modalities in individual brains still requires the development of a high-throughput and non-destructive whole-brain imaging method that preserves both the structural and biological integrity.

Photoacoustic computed tomography (PACT) is potentially capable of fast, nondestructive, structural, and molecular imaging of individual animal brains[14–19]. In PACT, brain samples are illuminated by short laser pulses. By converting light energy into ultrasound emission via the photoacoustic (PA) process, PACT detects the ultrasound signal and recovers images with optical absorption contrast and ultrasonic resolution. Since PACT relies on scattered rather than ballistic photons for image formation, its penetration depth is well beyond the one-millimeter diffusion limit of traditional optical microscopy, in the range of several millimeters to several centimeters[20], depending on the required image resolution. This method can also image the distribution of extrinsic molecular probes or products of reporter genes by means of spectral or temporal unmixing[15,21–26]. Although the spatial resolution of PACT is poor compared to that of standard optical microscopy, in the range of several tens to several hundreds of micrometers, it can still enable a number of first-of-its-kind capabilities, as demonstrated by this paper.

Although existing PACT systems have shown the possibility of whole-brain imaging[22–24,26,27], their performance does not meet the standard for practical uses as a cross-modal brain analysis platform. Many PACT systems employ ultrasound transducer arrays that are geometrically focused in the elevational direction[19,28–30], resulting in a large missing cone in this direction, within which all spatial frequency components are lost. The presence of the missing cone in the spatial-frequency domain is a significant contributor to image artifacts and resolution degradation. As a result, for three-dimensional (3D) imaging, the elevational resolution is usually more than five times larger than the in-plane resolution. Moreover, the missing cone not only blurs the image but also renders certain features invisible due to the bipolar and coherent nature of the PA signals, a phenomenon known as the limited-view problem[31–33]. Although by scanning a specially designed transducer array, the missing cone can be eliminated or reduced[14,17,34], a compromise must be made among resolution, imaging speed, field of view (FOV), and sensitivity.

In this study, we propose a PACT imaging system employing a focused half-ring ultrasound transducer array with a translation-rotational scanning strategy[35–38]. The FOV of the image can cover a ferret or marmoset brain, up to 24 mm in diameter, with an isotropic resolution of approximately 140 μm over the entire FOV. The design allows us to acquire high-quality PA images at a relatively low cost and in a relatively short time. Taking advantage of the photobleaching of the fluorescent tags[39] that exhibit a characteristic temporal decay, we designed the scanning sequence to specifically extract the signals from the tags, using a single illumination wavelength. We utilized photobleaching, which is typically seen as a hindrance, in a new way to suppress the intrinsic tissue background, resulting in the generation of 3D PA images of enhanced detection sensitivity specifically for these fluorescent tags (Fig. 1a). We further enhanced the detection accuracy by deep learning. Accordingly, we named our new imaging technology Photoacoustic Tomography with Temporal-Encoding Reconstruction (PATTERN).

We demonstrated the compatibility and versatility of PATTERN by imaging common fluorescent proteins[40] such as iRFP713 and mScarlet,

to achieve highly specific imaging of brain regions and their projections. Thus, PATTERN provides a new optical approach to visualize brain-wide neural connectivity with nondestructive sample preparation, high imaging speed, moderate resolution, and reliable accuracy. Based on the minimal influence of PATTERN on brain tissue, we can combine it with other whole-brain imaging modalities, such as fMOST and LSFM, to obtain 3D images of the whole brain with both high resolution and morphological fidelity. This approach can effectively identify and correct the alterations caused by sample preparation to both morphology and fluorescence signals. Furthermore, PATTERN demonstrates the capability of analyzing brain-wide cross-modal data from individuals, encompassing various imaging modalities, biological measurements, or sequencing data. Specifically, we show PATTERN-based integration of 2D spatial transcriptomic data with whole-brain optical information, potentially enabling the creation of a 3D cross-modal connectivity and transcriptome map.

## Results

### Near isotropic photoacoustic imaging of PATTERN

The PATTERN system was upgraded from a conventional PACT system with a half-ring array transducer to minimize the cost and system complexity (Fig. 1b, Supplementary Fig. 1 and Supplementary Movie 1). The original PACT system[41] only scans linearly along the elevational direction in 3D imaging. The restricted acceptance angle of the transducer array leads to diminished resolution along the scanned direction. In PATTERN, we address this issue by rotating the system to mitigate the limited-view problem (Supplementary Note 1). The 3D images acquired at different scan angles are combined to reconstruct a 3D volume with near isotropic resolution (Fig. 1c–g). To maximize the signal-to-noise ratio (SNR), the image of each angle was filtered by a multi-angle filter and then summed (Fig. 2a and Supplementary Figs. 2, 3). We refer to each translation-rotational scan procedure as a "scan cycle".

The resolution enhancement was verified by imaging latex beads with diameters of 20 μm. A single scan cycle took 133 s, providing resolutions along x, y, and z of $137 \pm 15.1\,\mu m$, $115 \pm 24.3\,\mu m$ and $143 \pm 20.1\,\mu m$, respectively ($n = 345$ beads; mean ± standard deviation), which was almost isotropic as expected (Supplementary Fig. 4). In comparison, the elevational and in-plane resolutions were measured to be 1084 μm and 123 μm for the translational scan alone (Supplementary Fig. 2b, c). The spatial frequency distributions (SFDs) corresponding to several linear scans at various angles, as well as the ultimate image reconstructed at the end of a scan cycle, suggest that the missing spatial frequency components were recovered by the scanning process (Fig. 2b). Meanwhile, we show the dramatic difference between images of the same brain reconstructed by single-angle and multiangle scans (Fig. 2c). It is obvious that the elevational resolution is significantly improved, while an in-plane feature becomes sharper due to reduced contamination by the out-of-plane signals (Fig. 2d).

### Temporal encoding and unmixing of fluorescent tags by PATTERN

PA images have a strong intrinsic tissue background. The PA signal from aggregated somas was comparable to that of the fluorescent protein (Fig. 2e). To show the superiority of temporal unmixing over spectral unmixing in the translation-rotational scan scheme, we tested both methods. During the scan process, repeated laser exposure caused signal decay due to photobleaching (Supplementary Fig. 5a, b). The change in the signal strength distorts the measured PA spectrum, making spectral unmixing extremely complex. In addition, in our spectral sweeping window (680–1064 nm), there is a limited variety of usable fluorescent tags (Supplementary Fig. 5c). Conversely, the temporal unmixing method involves single-wavelength operation only, thus making full use of the pump laser (532 nm) to detect red

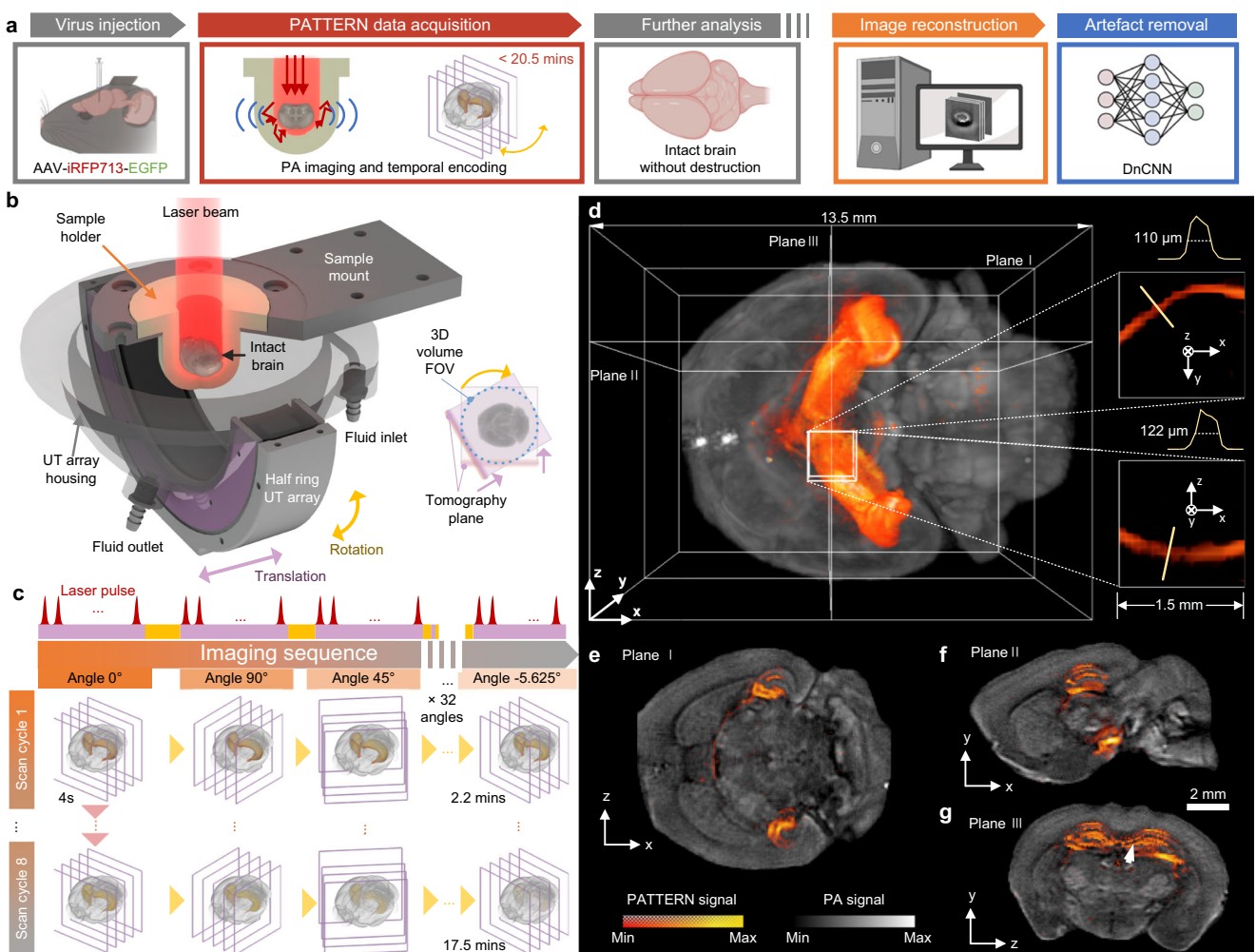

**Fig. 1 | The PATTERN concept. a** The pipeline for PATTERN imaging. Partially created with BioRender.com released under a Creative Commons Attribution-NonCommercial-NoDerivs 4.0 International license https://creativecommons.org/licenses/by-nc-nd/4.0/deed.en. **b** Schematic of the PATTERN system. The agar-based sample holder enables intact brain imaging without the need for embedding, and the inlet and outlet were perfused with phosphate buffer solution (PBS) or 4 °C artificial cerebrospinal fluid (aCSF) to maintain biological activity. Inset: the FOV of PATTERN is a cylinder with a diameter defined by the translation distance. **c** Imaging sequence of PATTERN. A total of 32 rotations and 1024 translation steps were performed in a single scan cycle. A typical bleaching process includes 8 scan cycles, taking approximately 17.5 mins. **d–g** PATTERN images of a mouse brain whose hippocampus was labeled with iRFP713 and EGFP. **d** Bottom view of a 3D-rendered volume. Insets: magnified views of a projection feature. **e–g** Tomographic views from three orthogonal perspectives related to planes in (**d**). The bleaching curve of the point indicated by the white arrow is shown in Fig. 2e.

fluorescent proteins (i.e., mScarlet). These factors prompted us to adopt the temporal unmixing method to improve the sensitivity to signals from the labels. Compared to using intensity alone without employing temporal encoding, PATTERN's detection sensitivity of fluorescent proteins is significantly enhanced (Fig. 2f). To clarify how we quantified this sensitivity improvement, we examine a thin slice (Fig. 1g). The conventional approach for distinguishing fluorescent labels from intrinsic background involves applying a threshold set at the maximum value of the intrinsic background signal. In Fig. 2f, this threshold is referred to as the "background roof" and is depicted as the red dashed line. Signals exceeding this threshold are identified as originating from fluorescent tags, while signals with amplitudes below the threshold are considered background. It is evident that this thresholding method is highly insensitive, resulting in the discarding of a substantial portion of the actual signal. In contrast, the novel temporal decoding method implemented by PATTERN ensures the effective removal of the unbleachable background. The background-rejection threshold can be set to three times the noise standard deviation (STD), denoted as the "PATTERN noise floor" and illustrated by the green dashed line in Fig. 2f. This approach significantly

improves sensitivity by preserving a larger portion of the real signal. We ensured that the small region chosen to compute the STD of the noise contained no signal by cross-referencing it with the confocal imaging results. By comparing the two thresholds, a gain of 12 was achieved, indicating the enhancement in detection sensitivity (Fig. 2f). Such enhancement was validated across the three different types of fluorescent proteins (Supplementary Fig. 6). Moreover, the resolution of the PATTERN-resolved features was the same as that of the intrinsic PA contrast (Fig. 1d).

Based on the rotation-translation scan strategy and the temporal unmixing method, we performed two optimization steps: (1) for a single sample, we applied multiple scan cycles (typically 8 cycles) to effectively bleach the fluorescent tags, ensuring a sufficient extent of bleaching (Fig. 1c). Moreover, the 3D images with different scan angles in adjacent cycles can also be fused into a 3D image with the ultimate high resolution. By doing so, only a few scan cycles can produce many high-resolution 3D images to improve the credibility of the bleaching curve fitting (Fig. 2a, e). (2) In a single scan cycle, the rotation angle followed a jumping sequence of 0°, 90°, 45°, −45°, …, which ensures that the k-space was filled relatively uniformly. In contrast, if we were

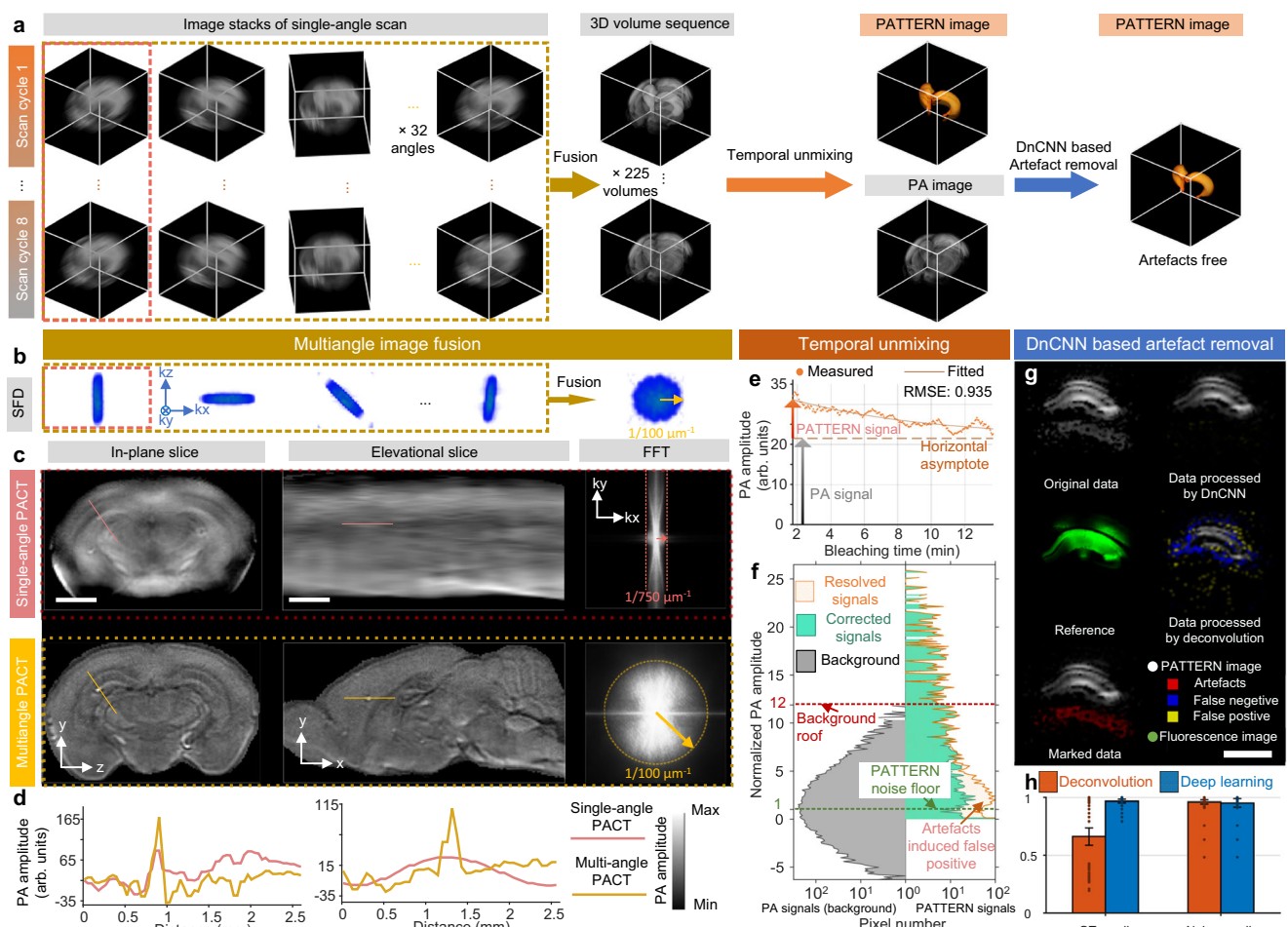

**Fig. 2 | PATTERN enables near isotropic and high-fidelity imaging of the whole brain and fluorescent tags. a** Data processing procedure of PATTERN. **b** SFD of each scan related to different angles in (**a**). **c** Resolution enhancement of the elevational direction by a rotation strategy. The first row, single angle scan reconstruction; the second row, multiangle scan reconstruction. The fast fourier transform (FFT) results indicate significant improvement, corresponding with (**b**). Scale bars, 2 mm. **d** Profiles of PA amplitude of lines indicated in (**c**). **e** Temporal unmixing results of the point related to the arrow in Fig. 1g. **f** Histograms of both PA signals (gray) and PATTERN signals (orange) in Fig. 1g; the corrected PATTERN

signals guided by confocal are marked individually in green. Green dashed line, the noise floor is estimated as three times the standard deviation of PATTERN signals' noise. Red dashed line, the roof of PA signals (maximum of background). **g** Diagram of the artifacts removal results using DnCNN with respect to the PATTERN signal of a hippocampus slice. Scale bar, 1 mm. **h** Comparison of ground truth (GT) recall and noise recall between the deconvolution-based method and deep learning method. $n = 77$ biologically independent image slices. Error bar length: 2 * standard error of the mean (SEM).

to scan in the angular direction in an incremental manner (0°, 5.625°, 11.250°, …), uneven filling of k-space in the azimuthal direction occurs due to the gradual signal decay from photobleaching, leading to a reduction in both image quality and fidelity of the bleaching curve (Fig. 2a and Supplementary Figs. 7, 8).

To prevent any ambiguity, we utilize the term "PA signal/image" in the current work to refer to signals and image features that correspond to the intrinsic tissue background. Such background contrast, depicted in grayscale across all subfigures (except Fig. 2g), is potentially attributed to lipids[19]. Conversely, we employ the term "PATTERN signal/image" to refer specifically to signals and image features that are derived from temporal encoding, as illustrated using pseudocolor superimposed on the grayscale background. Moreover, we use the term "fluorescence signal/image" to indicate signals and images that are generated through purely optical imaging techniques.

Due to the nondestructiveness of PATTERN, we were able to verify the fidelity of the signal from the labeled positions by invasive fluorescence imaging. This was achieved by sectioning the mouse brain after PATTERN imaging, followed by investigating the slice under a confocal microscope (Supplementary Fig. 9). We compared the histograms of the PATTERN signals before and after confocal-imaging-

guided corrections, which shows that PATTERN is accurate (Fig. 2f). Additional experiments indicated that for best performance, a "bleaching extent to noise ratio" (BNR) (see Methods) exceeding 47 dB was required, whereas the minimum BNR for the PATTERN approach was approximately 25 dB (Supplementary Fig. 10). The weak PATTERN signal slightly above the noise floor exhibits some false positives because of artifacts in the raw PA images[42].

For further improving our system, we next used confocal microscopy images as a reliable reference and employed a denoising convolutional neural network (DnCNN)[43] to remove the potential false-positive signals (Fig. 2a). Relying on a single-batch learning strategy, DnCNN was capable of extracting artifacts in the vicinity of true signals (Fig. 2g, Supplementary Fig. 9). The majority of the artifacts caused by the unideal response of the ultrasound transducers (Supplementary Fig. 11) were effectively eliminated without causing severe false-negative signals compared to traditional deconvolution methods (Fig. 2h).

## PATTERN-based whole-brain optical imaging
In light of the data above showing the capability of PATTERN for whole-brain optical imaging with isotropic resolution, large FOV, and

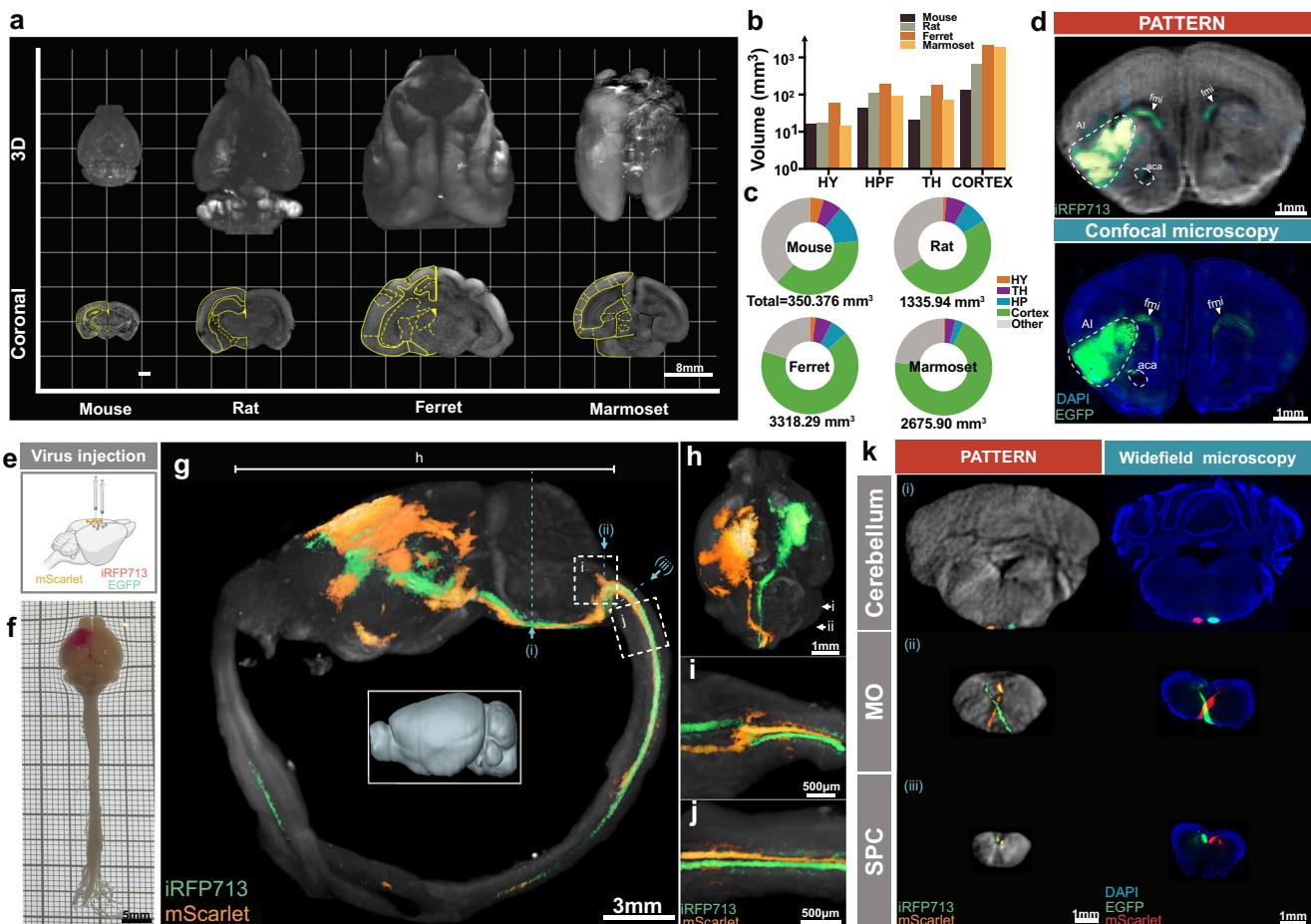

**Fig. 3 | PATTERN-based whole-brain optical imaging. a** Representative 3D whole-brain imaging of mouse, rat, ferret, and marmoset via PATTERN. **b, c** Geometric analysis of the representative mouse, rat, ferret, or marmoset brain according to data acquired by PATTERN. HY Hypothalamus, HPF Hippocampus Formation, TH Thalamus. **d** Validation of PATTERN-acquired fluorescent signal. **e** Dual-color labeling strategy for visualizing projections from the motor cortex to the spinal cord. Partially created with BioRender.com released under a Creative Commons Attribution-NonCommercial-NoDerivs 4.0 International license https://creativecommons.org/licenses/by-nc-nd/4.0/deed.en. **f** Bright field images of entire central nervous system. 3D-view images by PATTERN of the entire central nervous system (**g**), brain only (**h**), medulla oblongata (**i**), and spinal cord (**j**). **k** Coronal section via PATTERN (left column) and widefield microscopy (right column), medulla oblongata (MO), and spinal cord (SPC). Positions were labeled by roman numerals corresponding to (**g**).

high imaging speed, we next imaged the ex-vivo brains of different animal models. PATTERN achieved whole-brain imaging of mouse, rat, ferret, and marmoset brain with high geometric fidelity (Fig. 3a and Supplementary Movie 2) and enabled quantitative morpho-metric analysis of different brain regions (Fig. 3b, c). Utilizing photobleaching-based temporal encoding, the PATTERN system has been enhanced to detect fluorescent proteins within brain tissue effectively. To further validate the accuracy of our system for fluor-escent protein detection, we employed confocal data from the same brain samples, which contained both AAV-expressed iRFP713 (for PATTERN imaging) and EGFP (for confocal imaging). Using confocal images as the references, PATTERN images showed consistent fluorescent signals (Fig. 3d). This consistency encouraged us to explore more neuroscience applications. Firstly, PATTERN exhibited the ability to visualize the implants in the brain such as the optical fibers used in optogenetics and the electrodes used in electro-physiology (Supplementary Fig. 12a–d). Secondly, based on the reli-able detection of PA signals and the large FOV enough for simultaneously imaging two brains of mice (Supplementary Fig. 12e, g), PATTERN could also quantify viral vector expression in brains with the injection of different viral types (canine adenovirus type 2 (CAV2) or AAVretro) (Supplementary Fig. 12e, f), or different injection strategies (same total viral titer with different volume of

injection) (Supplementary Fig. 12g, h), which could assist the opti-mization of virus injection experiments. More importantly, PATTERN could image neural connectivity via visualizing brain-wide neuronal projections and we showed an example of the projection from the dorsal subiculum (dSub) to the downstream, such as mammillary bodies (MM) and entorhinal cortex (EC) (Supplementary Fig. 13). Remarkably, PATTERN could image the entire central nervous system of a mouse, allowing tracing the long-range projections, such as the projection from the motor cortex to the spinal cord (Fig. 3e–j and Supplementary Movie 3). We also validated that the details of these cortico-spinal projections acquired by PATTERN matched well with fluorescence imaging data (Fig. 3k and Supplementary Fig. 14). Overall, these results demonstrated that PATTERN provided a new method for whole-brain structural and fluorescent signal imaging within 20 min.

## PATTERN for visualizing neural connectivity of the brain
Revealing the connectivity of the brain is crucial for understanding its functions and dysfunctions[44,45]. Since PATTERN can achieve reliable 3D whole-brain optical imaging, we sought to explore its potential to visualize neural circuits. For that purpose, we injected an AAV vector to express iRFP713 fused with EGFP in the brain area of interest, allowing us to validate the PATTERN-traced projections by subsequent other

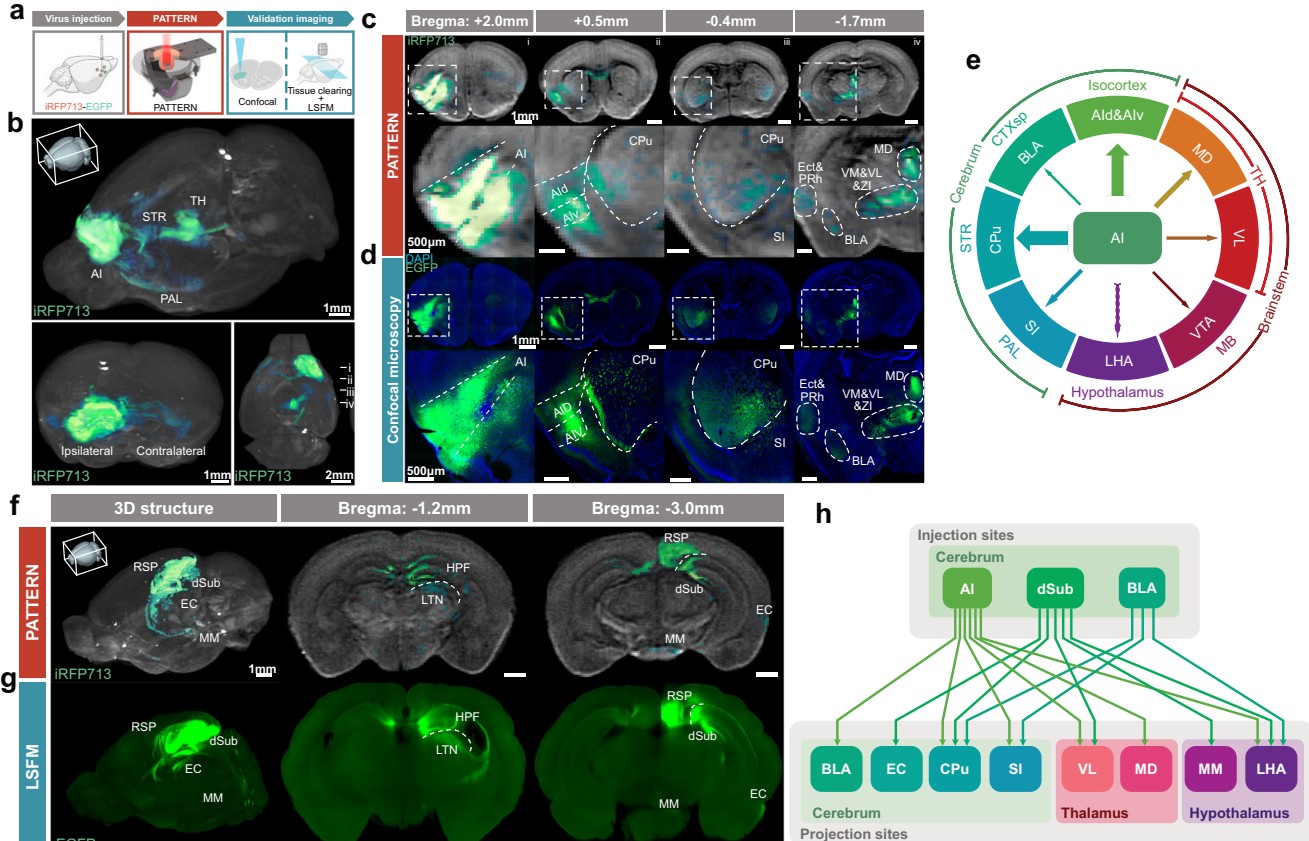

**Fig. 4 | PATTERN for visualizing neural connectivity of the brain. a** The pipeline for PATTERN imaging followed by validation imaging. Partially created with BioRender.com released under a Creative Commons Attribution-NonCommercial-NoDerivs 4.0 International license https://creativecommons.org/licenses/by-nc-nd/4.0/deed.en. **b** 3D-view imaging of the projection map of AI from three directions via PATTERN, striatum (STR), thalamus (TH) and pallidum (PAL). Coronal sections of PATTERN images (**c**) and confocal images (**d**) at the injection site (i) and some projection sites (ii–iv). The second row of each modality is the amplification of the signal area (outline with a white box), anterior insular cortex dorsal part (AId), anterior insular cortex ventral part (AIv), caudoputamen (CPu), sybstantia innominata (SI), ectorhinal area (Ect), periehinal cortex (PRh), ventromedial thalamic nucleus (VM), ventrolateral thalamic nucles (VL), zona incerta (ZI) and mediodorsal thalamic nucleus (MD). **e** The major downstream brain regions diagram of AI, lateral hypothalamic nucleus (LHA), midbrain (MB) and ventral tegmental area (VTA). The 3D and coronal images via PATTERN (**f**) and LSFM (**g**) retrosplenial area (RSP), entorhinal cortex (EC), medial mammillary nucleus (MM), hippocampal formation (HPF) and laterodorsal thalamic nucleus (LTN). **h** The projection map diagram of the connection atlas from the AI, dSub and BLA.

optical imaging methods (Fig. 4a). We first utilized PATTERN to image the projections of the anterior insular cortex (AI). We observed that the AI projects to the other parts of the cerebrum and brainstem such as the caudoputamen (CPu) in the striatum, the basolateral amygdala (BLA), and the thalamic nucleus (MD) in the thalamus (Fig. 4b, c and Supplementary Movie 4). The results were consistent with the confocal data of the same brain sample (Fig. 4d) and the data from Allen Brain Atlas[46,47] (Supplementary Fig. 15). Accordingly, we compiled a summary of the major downstream brain regions of AI (Fig. 4e) and the results were consistent with previous studies[48]. We also verified the ability of PATTERN to trace neural projections via comparing the data from LSFM imaging combined with tissue clearing and found matched signals in the same brain with AAV injection in the dorsal subiculum (dSub) (Fig. 4f, g, Supplementary Movie 5). Consequently, PATTERN could help us summarize the projections from the AI, dSub, and BLA (Supplementary Fig. 16) to their downstream (Fig. 4h). Thus, our results demonstrated that PATTERN could acquire reliable projection information and exhibited the capability to be a new tool for whole-brain connectivity analysis.

**Cross-modal whole-brian optical imaging via the combination of PATTERN with fMOST or LSFM imaging**
Compared to other brain-wide optical approaches[3,7,9,11], a significant advantage of PATTERN is the capability to nondestructively image the

whole-brain structure and fluorescence distribution with high geometric fidelity. Thus, our system showed the potential for the analysis of cross-modal data from the same brain when combined with other imaging modalities. We next utilized the 3D structure acquired by PATTERN as a template to correct the anisotropic deformations caused by sample preparation of fMOST and tissue clearing, enabling precise morphological analysis. Accordingly, we proposed a pipeline for the correction, in which the tested brain was initially imaged by PATTERN and then by another imaging method. These cross-modal data allowed us to align the deformed fMOST or LSFM images to the PATTERN-acquired images (Fig. 5a). Based on the contour of PATTERN data, such alignment[49] rectified the deformations in the fMOST[50] (Fig. 5b, c and Supplementary Movie 6) and LSFM[51] data (Fig. 5d, e and Supplementary Movie 6). Furthermore, this strategy was applicable in brains with considerable heterogeneity, where a reference template for correction and alignment was lacking. We imaged an aged rat brain with neurodegeneration (Alzheimer's disease)[52] by PATTERN, followed by tissue clearing and LSFM imaging[51]. Utilizing the self-to-self template acquired by PATTERN, we could correct the anisotropic morphological changes (Fig. 5f and Supplementary Movie 7) caused by sample preparation (Supplementary Fig. 17).

In addition to morphological deformation, the sample preparation for sectioning tomography and tissue clearing may also reduce fluorescence integrity or introduce autofluorescence noise[7,9,11]. To

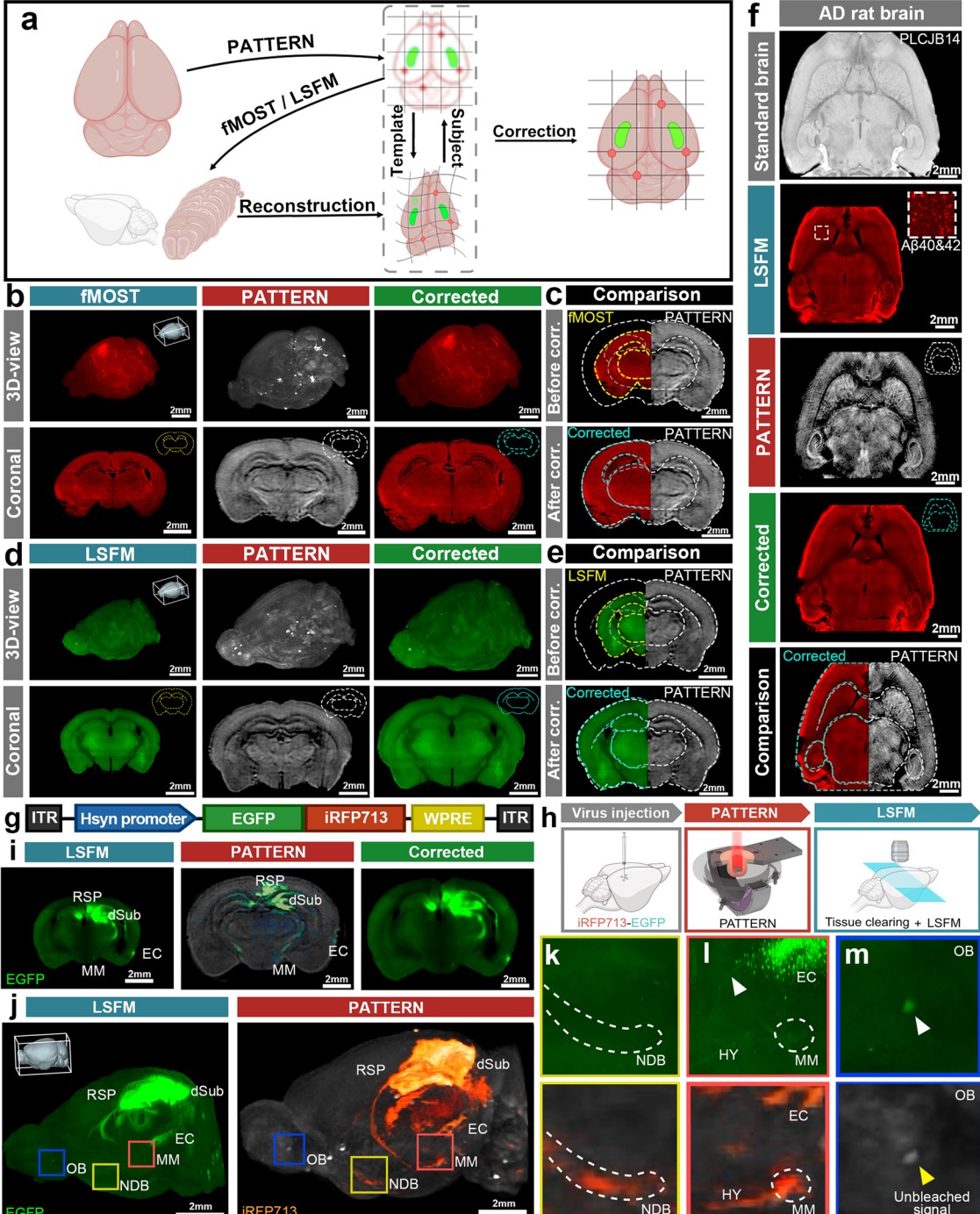

**Fig. 5 | Combination of PATTERN imaging with fMOST or LSFM. a** Schematic of the cross-modal correction strategy. Partially created with BioRender.com released under a Creative Commons Attribution-NonCommercial-NoDerivs 4.0 International license https://creativecommons.org/licenses/by-nc-nd/4.0/deed.en. **b–f** Cross-modal correction applied to fMOST and LSFM images (**b**), 3D and representative coronal sections of the brain that underwent fMOST; (**d**), 3D and representative coronal sections of the brain that underwent uDISCO; (**c**) and (**e**), Comparison of coronal sections before and after correction. **f** Horizontal sections of an AD rat brain that underwent iDISCO. The yellow dashed line represents the contour of fMOST or LSFM, white represents PATTERN, and blue represents the corrected result. **g** Schematic of AAV expressing EGFP and iRFP713 fusion protein.

**h** Experimental flowchart for correction of fluorescence signals via PATTERN. Partially created with BioRender.com released under a Creative Commons Attribution-NonCommercial-NoDerivs 4.0 International license https://creativecommons.org/licenses/by-nc-nd/4.0/deed.en. **i** Representative coronal sections of brain with fusion protein that underwent PEGASOS. **j** 3D structure of the brain with fusion protein. **k–m** Magnified view of the regions indicated by the colored box in (**j**). White arrows represent the false-positive signals; yellow represents the unbleachable signal, dSub dorsal subiculum, EC entorhinal cortex, MM medial mammillary nucleus, RSP retrosplenial area, NDB nucleus of the diagonal band, OB olfactory blub, HY hypothalamus.

address this issue, we used the designed AAV vector expressing a fusion protein of iRFP713 and EGFP (Fig. 5g), which could be detected by both PATTERN and other whole-brain imaging methods with identical spatial distribution. We then injected the AAV into the dSub and performed PATTERN analysis before tissue clearing with PEGASOS[53] and LSFM (Fig. 5h). We found that some PATTERN-detected projections that were reported in the Allen Brain Atlas[46,47] (Supplementary Fig. 16a) were not detected by LSFM imaging, which might be caused by fluorescence reduction or autofluorescence interference during tissue clearing[11]. Using our cross-modal correction (Fig. 5i, j), PATTERN could provide a preliminary view and guidance for analyzing these possible false-negative signals (Fig. 5k, l). In addition, our system could also distinguish the unbleachable impurities from the fluorescent proteins based on temporal unmixing (Fig. 5m). Overall, PATTERN shows compatibility with other imaging modalities and the potential to create a brain atlas with both high resolution and morphological fidelity (Supplementary Fig. 18a–e), which provides the basis for cross-modal analysis of individual brains.

### PATTERN-based cross-modal transcriptome analysis of an individual brain

The involvement of the spatial transcriptome in the cross-modal analysis is crucial for understanding cell identity and function in the tissue context. Despite recent efforts in cross-modal brain-wide analysis[13,54], there are still limitations in combining whole-brain fluorescence information with omics analysis in the same brain. Based on the non-destructiveness of PATTERN imaging, we sought to incorporate PATTERN-acquired 3D whole-brain fluorescence data into gene activity profiles to instruct 2D transcriptome analysis (Fig. 6a). We first verified that imaging by PATTERN did not cause significant changes to the sample at the protein and RNA levels (Supplementary Fig. 18f, g). Thus, we designed a pipeline in which whole-brain fluorescence imaging by PATTERN was performed before tissue slicing for spatial transcriptome sequencing (Fig. 6a, Supplementary Fig. 18h). Consequently, we could align spatial transcriptome data to PATTERN-acquired images (Fig. 6b, c). Guided by fluorescent signal, we could select the region of interest from the whole profiles for subsequent analysis (Fig. 6d). Following this pipeline, we injected AAV-U6-shRNA(c-Fos)-CMV-iRFP713 into part of the hippocampus, then tested whether and how the knockdown of c-Fos, an immediate early gene related to learning and memory[55], changes the transcriptome of hippocampal CA1 region after behavioral stimuli (Fig. 6a). We focused on the stratum oriens (so) and the stratum pyramidalis (sp) of the hippocampal CA1. The result showed that all data from these regions were clustered into 5 subclusters by using Seurat spatial transcriptome analysis method (Fig. 6e). In particular, two subclusters (subcluster-3 from CA1so and subcluster-5 from CA1sp) were found in the groups with AAV infection rather than control groups without virus injection (Fig. 6h). According to PATTERN-detected fluorescent signal, we could identify the iRFP713-positive region as the c-Fos knockdown region and found that most spots from these two subclusters were iRFP713-positive (Fig. 6f, g) and the spatial distribution of such spots matched well with iRFP713-positive region of CA1 (Fig. 6h). Supportively, the gene expression pattern of those two subclusters and PATTERN-selected iRFP-positive spots were also consistent (Fig. 6i). These results demonstrated that this 2D spatial transcriptome analysis can be guided by PATTERN-acquired 3D fluorescence information. Together, PATTERN provides a comprehensive solution for integrating brain-wide 3D fluorescence information with molecular omics analysis to achieve multidimensional and cross-modal exploration.

## Discussion

Brain-wide and cross-modal analysis is a promising approach for further understanding brain function at different levels. Existing whole-brain optical imaging approaches have provided opportunities to achieve multimodal brain atlases in standardized and homogeneous animal models[12,13]. However, when it comes to studying neural diseases or new mammalian models with large individual heterogeneity, achieving individual-level cross-modal analysis still faces significant challenges. PATTERN, our newly developed whole-brain PA imaging platform, addresses these challenges by allowing minimal sample preparation damage and high-speed imaging of the whole brain and spinal cord. This nondestructive approach allows for subsequent analysis of the same sample, facilitating cross-modal analysis of an individual brain. The development of PATTERN enables brain-wide optical and multi-omics analysis to capture individual variation, ultimately leading to a more comprehensive understanding of brain diversity.

### Comparison of PATTERN with other PA brain imaging studies

Given its capacity to identify optical absorption contrasts in deep tissue with high resolution, PA imaging has gained much attention in brain research. Recently, various PA imaging systems have been developed to meet the specific needs of different neuroscience studies. Contrasts between oxyhemoglobin and deoxyhemoglobin have rendered PA imaging an effective tool for visualizing the distribution of oxygen saturation[14,19], especially when combined with functional magnetic resonance imaging (fMRI)[56]. Using exogenous fluorescent probes, PA imaging has been also explored for visualizing enhanced molecular details deep within the brain[19]. With the enhancement of temporal resolution, PA imaging has also demonstrated its capability to detect GCaMP signals[27]. However, the ability and sensitivity of detecting fluorescent proteins, which are prominent requirements for neuroscience studies, are still not further optimized in most PA imaging techniques. Thus, the primary focus of the PATTERN system was to improve sensitivity and image fidelity across a broader range of fluorescent proteins, thereby extending the potential applications of PA imaging in neuroscience studies.

To meet the requirement of high-resolution and high-throughput whole-brain PA imaging, a system with point-like ultrasound transducers is preferred for the best imaging quality (isotropic resolution)[14,27,34,57]. However, the reported PA systems of this kind, to our knowledge, are not suitable for the tasks reported in the current work. Difficulties include low resolution or sensitivity to clearly distinguish brain regions, limited FOV for large brains, and excessive laser exposure for fluorescent proteins (Supplementary Fig. 19). Challenges arise due to the difficulty in manufacturing sensitive, broadband, and small-footprint ultrasound transducers. In this study, we employed a relatively simple and cost-effective imaging configuration to achieve both good image quality and high sensitivity to molecular probes. The filtered multi-angle reconstruction procedure, designed specifically for the translation-rotational scanning strategy (Supplementary Fig. 20), can be better understood by visualizing it in the spatial frequency domain, which allows for the optimization of scanning parameters[38]. Additionally, several measures can be implemented to enhance PATTERN's performance and impact. Firstly, faster lasers or potentially, some deep-learning-based image fusion approaches[58] can be employed to accelerate the imaging process, while the imaging resolution can be improved by using ultrasound transducers with larger bandwidths. Specifically, the utilization of transducers with a frequency response ranging from direct current (DC) to 22 MHz has demonstrated the ability to achieve a resolution of approximately 50 μm[57], which is comparable to the resolution achieved in all-optical, large-FOV brain imaging[58,59]. Secondly, regardless of the detailed implementations, the PATTERN concept is adaptable to PAT platforms involving various scan strategies and could potentially enhance their sensitivity to fluorescent proteins. Thirdly, with the bleaching non-linearity of the fluorescent proteins calibrated, their molecular concentrations can potentially be quantified by PATTERN (Methods and Supplementary Figs. 21 and 22). Lastly, the background rejection

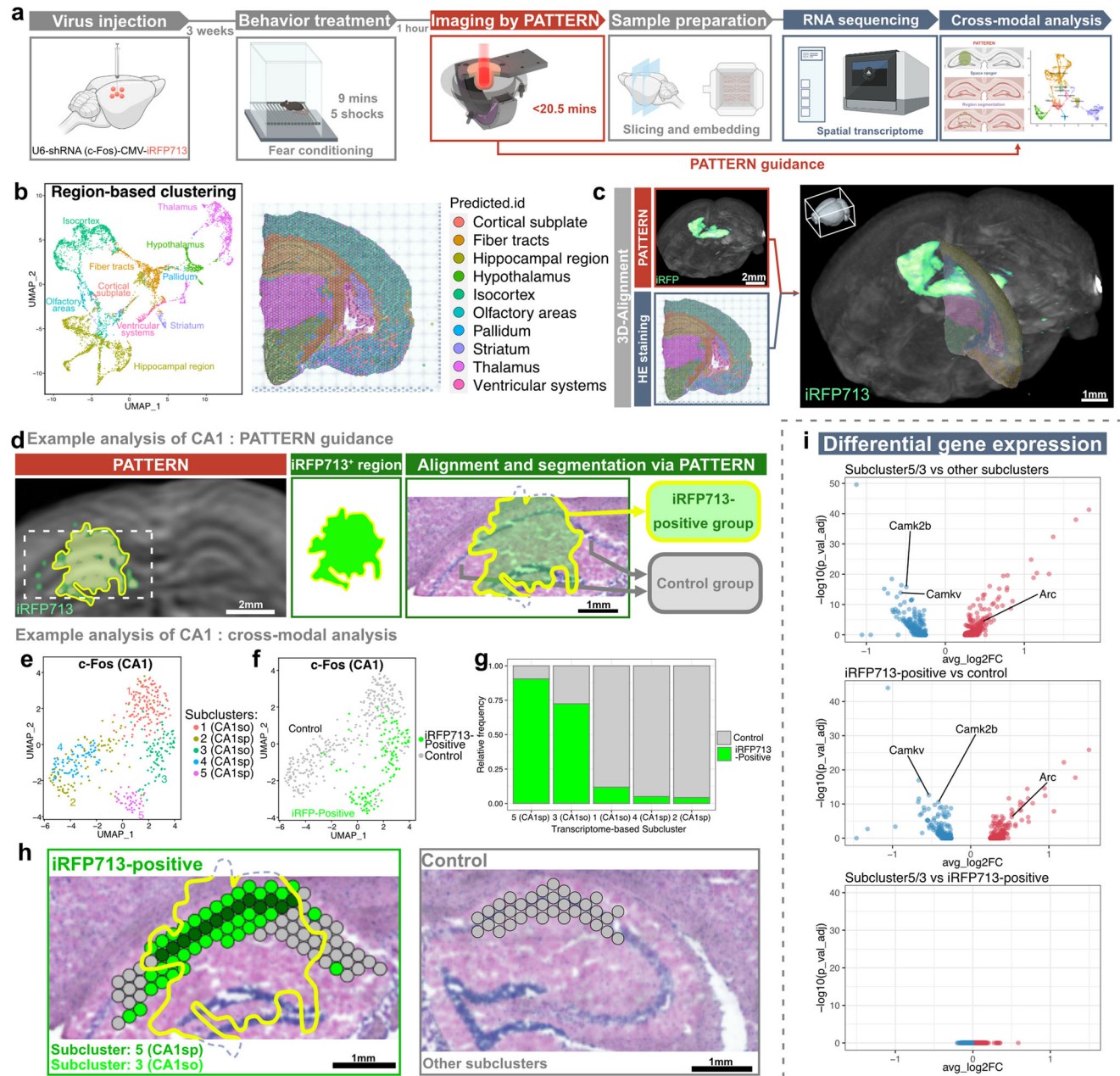

**Fig. 6 | PATTERN for cross-modal transcriptome analysis of an individual brain.**
**a** Pipeline of combining PATTERN and spatial transcriptome. Partially created with BioRender.com released under a Creative Commons Attribution-NonCommercial-NoDerivs 4.0 International license https://creativecommons.org/licenses/by-nc-nd/4.0/deed.en. **b** Region-based UMAP clustering and spatial map of a hemisphere. **c** 3D alignment of spatial transcriptome data to whole-brain images by PATTERN. **d** Pipeline of fluorescence alignment for the CA1 example. UMAP clustering of CA1 spots labeled by brain-region (**e**) or fluorescence (**f**). **g** Relative ratio of fluorescence-positive spots in each subcluster. Specifically, a CA1sp subcluster and a CA1so subcluster were labeled intensively. **h** Spatial identification of subclusters and manipulated regions. **i** Differential gene expression of different subclusters. Transcriptome-based selected subclusters are highly correlated with fluorescence-positive subclusters at the gene expression level. A two-sided Wilcoxon Rank Sum test was performed and the *p* value was adjusted based on bonferroni correction.

capability of PATTERN can be further explored for in vivo applications (Fig. 2e and Supplementary Fig. 23).

It has been shown that the use of photoswitchable proteins significantly improves intrinsic tissue background rejection, thus enhancing sensitivity to molecular labels[22]. However, compared to traditional fluorescent proteins[40], further development of photo-switchable proteins is required to address the diverse needs in various neuroscience studies. Thus, in PATTERN, we employed the photobleaching process instead of the photoswitching process, to ensure compatibility with routine labeling strategies (Supplementary Note 2). Compared to the multispectral unmixing method, temporal encoding is immune to the spectral coloring effect[22,23] and

involves only a single-wavelength operation. Currently, temporal encoding schemes typically require a relatively weak bleached signal so that the light fluence distribution inside the sample is not significantly affected. When the above assumption is invalid, errors can be generated, appearing as ripples superimposed on the measured bleaching curve in Fig. 2e. When such unwanted modulations are strong, the reliability of the temporal-encoding scheme decreases in analogs to the spectral-coloring problem in multi-spectral PA imaging[26]. For molecular probes that are difficult to photo-bleach, the traditional spectral-domain demodulation scheme can be used. Using the spectral unmixing method, we successfully imaged the distribution of DiR, a chemical dye that is

difficult to bleach in comparison to fluorescent proteins (Supplementary Fig. 12a).

## Comparison and combination of PATTERN with other whole-brain optical imaging technologies

Due to the limitations of optical microscopy imaging depth, serial sectioning and tissue clearing methods are used to achieve complete whole-brain imaging with high spatial resolution (Supplementary Table 1)[2,3,6,9]. However, both strategies require a sample preparation process that involves morphological and biological changes, significantly influencing accurate geometric measurement, precise fluorescence signal analysis, and automatic registration of brain samples. Moreover, many of these sample preparation methods discard biological activity or damage the molecular information of the original sample, hindering the possibility of testing the sample in other modalities.

Compared with these strategies, PATTERN could also achieve reliable 3D visualization of brain-wide neural projections even extending from the cortex to the spinal cord with isotropic resolution. More importantly, the non-destructive sample preparation process and high-speed imaging process minimize the physical and physiological perturbations to the tested brain sample. Consequently, our system provides a novel approach to brain-wide optical imaging with a larger FOV, higher imaging speed, and minimal tissue perturbation.

Nonetheless, PATTERN is not intended to replace or compete with existing high-resolution brain-wide imaging techniques but aims to work in combination with existing high-resolution brain-wide imaging techniques to achieve a more efficient and precise whole-brain optical analysis. PATTERN can capture images of freshly removed and untreated brain tissue, without causing structural and biological changes, allowing the same brain sample to be subsequently imaged by other high-resolution methods such as fMOST and LSFM with tissue clearing. By combining these cross-modal data, we could utilize the PATTERN-acquired image to rectify morphological and signal changes for other whole-brain imaging modalities, resulting in 3D whole-brain data with both high resolution and morphological fidelity. While aligning the brain from inbred animal models to a standard-brain template can partially correct deformation, challenges remain in animal models with high heterogeneity. Our system can effectively tackle these challenges in higher mammalian models and brain disease models.

In addition, high-resolution whole-brain imaging can often incur significant time and economic costs. Currently, there are no effective approaches to assess the information on injection sites of vectors, fluorescence expression intensity or signal loss during sample preparation. Coexpressing an infrared protein and a visible protein in the brain sample provides a viable means of pre-evaluating the quality of fluorescent labeling via PATTERN imaging. As demonstrated in the current work, photobleaching of the infrared protein does not affect the downstream fluorescence imaging of the visible proteins. Due to its capacity to obtain the whole-brain 3D fluorescence distribution in a fast, nondestructive, and high-throughput manner, PATTERN provides a new way of quickly and conveniently previewing the fluorescence signal quality of samples. It provides effective evaluation and guidance before conducting subsequent high-resolution imaging experiments or other high-cost experiments, such as single-cell sequencing and spatial transcriptomics. Therefore, incorporating PATTERN into the specimen analysis pipeline enhances the success rate and reduces overall costs. Notably, emerging magnetic resonance (MR)-based reporter gene imaging demonstrates the potential for non-destructive, in vivo whole-brain detection of gene-expression patterns[60]. However, the diversity of available labels for this approach is still limited.

## Cross-modal analysis guided by PATTERN

Although cross-modal analysis among different samples shows the power to uncover brain functions and dysfunctions[12], challenges remain in applying both whole-brain fluorescence imaging and analysis of other modalities to an individual brain. By combining PATTERN imaging and spatial transcriptome analysis of the same brain sample, we confirmed the possibility of using the 3D image obtained by PATTERN to guide transcriptomic analysis, enabling a more comprehensive understanding of spatial transcriptomic data. Given the numerous bioindicators for PA imaging that have been developed[17,61,62], PATTERN-based cross-modal analysis has the potential to analyze multiple modalities in conjunction with 3D fluorescent information in various animal models, since a similar approach was performed in Alzheimer's disease models at the 2D level[63]. Thus, PATTERN provides a versatile tool for analyzing individual brains using multiple modalities, enabling a flexible cross-modal analysis of an individual brain.

# Methods
## PATTERN

PATTERN uses a rotatable PACT system as its basic setup, which includes an excitation laser, scanning stages, and an ultrasound detection module (Supplementary Fig. 1). To ensure optimal system performance and user-friendliness, we incorporated a half ring array transducer with a 55 mm radius from ULSO TECH Co., Ltd. into the design. This transducer has a central frequency of 5.5 MHz, a detection bandwidth of 60% at −6 dB, and comprises 128 elements, each with a 1.32 mm (width) by 20 mm (height) aperture, which provides a cylindrical focus with numerical aperture. The ultrasound signal was recorded using a low-noise data acquisition system equipped with 128 channels (MarsonicsDAQ128, Tianjin Langyuan Inc.) with a sampling rate of 40 MHz and a dynamic range of 14 bits. The DAQ software allows for the real-time reconstruction of individual frames of PA signals, enabling the verification of data quality.

To achieve the specified imaging strategy, two motors were employed: a direct-drive motor (ADRS-200-M-A-NS, Aerotech Inc.) and a linear motor (ANT 25 L, Aerotech Inc.) for rotation and translation of the ultrasound transducer, respectively. The linear motor's translation speed was set to 6 mm/s, corresponding to a step length of 0.6 mm, which is smaller than the raw elevational resolution. Furthermore, the translation range was adjusted to ±9.6 mm for single mouse brain and whole CNS imaging and ±12 mm for multi mouse brains, rat brain, and ferret brain imaging, in line with the FOV size. To minimize the potential side effects of PA signal bleaching during the scanning process, a jumping sequence was applied for every scan cycle. The 32 rotation angles of the scan cycle were followed by a series of jumps to new angles in the following order: 0°, 90°, 45°, −45°, 22.5°, −67.5°, 67.5°, −22.5°, 11.25°, −78.75°, 56.25°, −33.75°, 33.75°, −56.25°, 78.75°, −11.25°, 5.625°, −84.375°, 50.625°, −39.375°, 28.125°, −61.875°, 73.125°, −16.875°, 16.875°, −73.125°, 61.875°, −28.125°, 39.375°, −50.625°, 84.375°, and −5.625° (Supplementary Fig. 7). It is worth noting that the rotation axis was carefully adjusted to pass through the central point of translation, and its accuracy was verified via bead imaging.

The excitation of PA signals was carried out using an optical parameter oscillator (LP604, Solar Laser) for iRFP and its pump source (LQ929B, Solar Laser) for mScarlet. The optical parameter oscillator delivered 680–1064 nm wavelengths with a pulse repetition rate of 10 Hz and a pulse width of 10 ns. In contrast, the pump source produced wavelength of 532 nm. During imaging, the laser fluence at the sample surface was 3.2 mJ/cm² at 690 nm for iRFP713, 3.4 mJ/cm² at 697 nm for SNIFP, 2.0 mJ/cm² at 532 nm for mScarlet, and 3.4 mJ/cm² at 800 nm for cross-modal registration. These fluence values were empirically determined to ensure both detectable bleaching extent of fluorescent proteins (favor high fluence, Supplementary Fig. 10) and high image quality of the bleached features (favor low fluence, Supplementary Fig. 7). To conveniently adjust the light spot size on the sample, a customized expander comprising three lenses (GCL-010112A, GCL-010166A, and GCL-010329, Daheng Optics) was used to expand the beam. The sample holder contained 2.25% w/w agarose (agarose G-10, BIOWEST) mixed with 3% v/v Intralipid-30% (Kelun Pharmaceutical Co., Ltd.) to increase

light scattering, allowing some incident light to scatter back to the sample. Additionally, a customized optical shutter assembled using a server motor (GM6020, DJI) and a 3D-printed blade was used to regulate the laser's exposure to the sample.

In addition, a perfusion module was utilized to maintain a liquid environment consistent with body fluid. For the PA imaging procedure followed by other imaging modalities, the imaging chamber was perfused with PBS; for imaging followed by other biological measurements, aCSF was used to minimize biological changes at the molecular level of the brain tissue[27] (Supplementary Fig. 18f–h).

### Data reconstruction and postprocessing

The recorded data underwent reconstruction using the delay and sum (DAS) algorithm, resulting in the production of 2D PA image stacks. DAS was used because of the fast reconstruction speed without sacrificing image SNR for high throughput imaging. Subsequently, these stacks were interpolated using an FFT method to generate 3D images with relatively poor elevational resolution. To improve the elevational resolution, a multiangle fusion procedure was applied consisting of the following steps: (1) Rotating and registering 3D images of 32 different angles into the same coordinates. (2) Transforming the images into the frequency domain, followed by filtering and summing. (3) Inversely transforming the uniformly-filled spatial frequency signal back into the spatial domain, resulting in a near isotropic 3D volume (Supplementary Fig. 2). The filter in (2) was designed to normalize the uneven sampling density in k-space. Specifically, at each scanning angle, a thin disk oriented along that angle is accessed within k-space. After completing a full scanning cycle, the disk undergoes a full rotation. When these disks are superimposed, the central portion is weighted more heavily. As a result, we attenuated the amplitude of each frequency component by its overall weight, while preserving the original phase. To prevent the introduction of extraneous noise, amplitudes beyond the frequency support of the filter were set to zero:

$$H(\theta) = S(\theta) \cdot \frac{1}{\sum_\theta S(\theta)}, \tag{1}$$

where $S(\theta)$ represents a transfer function corresponding to a translationally-scanned tomogram at angle $\theta$:

$$S(\theta) = \text{step}\left(k_x \cos(\theta) - k_y \sin(\theta) + W_e\right) \\ - \text{step}\left(k_x \cos(\theta) - k_y \sin(\theta) - W_e\right) \tag{2}$$

$W_e$ is the cutoff spatial frequency in elevational direction, and step() denotes the step function.

All of these procedures were performed using a GPU-accelerated program coded in Python 3.8. The voxel size was set to $75 \times 75 \times 75 \ \mu m^3$ to balance the reconstruction quality against video memory consumption.

The multiangle fusion procedure was repeated for every adjacent 32 angles in the whole scanning process, with eight scan cycles usually performed, creating 225 3D volumes in total (angle number per scan cycle × (scan cycle number-1) + 1 = 32 × (8-1) + 1 = 225). To ensure more accurate analysis, the last 224 volumes were registered to the first volume (unbleached state) using ANTS, an open-source registration software. Thus, a bleaching curve of each voxel was acquired with a temporal sample point of 225. These curves were then used to calculate the PATTERN signals. In our algorithm, we fit the PA amplitude decay curve using

$$A(t) = a \cdot \exp(-bt) + c, \tag{3}$$

where the parameter $b$ represents the bleaching rate, $a$ is the PA signal strength of the fluorescent tag which contributes to the useful signals,

referred to as "PATTERN signals" in the current work, and $c$ is the PA signal strength of any unbleached background chromophores, referred to as "PA signals" in the current work. To optimize computational speed, the process of curve fitting is divided into the following steps: Initially, an exhaustive search for the bleaching rate $b$ was conducted in parallel using GPU. The searching interval, $0 \le b \le 0.09$ per translational scan, was confirmed to fully cover the bleaching rates of different brain samples. A set of bleaching rates, typically comprising 12 rates to achieve a balance between accuracy and calculation speed, was employed to generate an equal number of corresponding bleaching curves. Subsequently, the correlation coefficient between the experimental data and each preset curve followed by its absolute value was calculated to estimate the confidence of the corresponding bleaching rate. Ultimately, the preset curve associated with the rate with the highest confidence was used to determine both the amplitude of the decreasing part (PATTERN signal) and the amplitude of the constant part (PA signal) through linear unmixing. To guarantee the stability of the calculation, when the bleaching rate was determined to be exceptionally small (e.g., $b = 0.0001$), the PATTERN signal was set to zero. This precaution was taken to prevent the inversion of a low-rank matrix. Physically, setting the PATTERN signal to zero signifies classifying the voxel as unbleachable. The PATTERN signal amplitude was then transformed into PATTERN signal intensity using absolute value manipulation (Supplementary Fig. 5d). The calculation was applied voxel by voxel using GPU for acceleration. A typical volume of $256 \times 256 \times 256$ took ~11 min using TITAN RTX.

The obtained results underwent further processing to enable visualization. Initially, the imaging feature, such as a mouse brain along with the agarose-based sample holder, was segmented using Amira software. Subsequently, a 3D Monte Carlo simulation, conducted using MCXLAB[64] (Supplementary Methods), was employed to simulate the light fluence distribution, considering $\mu_a = 0.005$ and $g = 0.9$ for the brain, and $\mu_a = 0.002$ and $g = 0.98$ for the sample holder (agarose). We utilized a uniform light source covering the inner surface of the sample holder to replicate the experimental conditions. The resulting light fluence distribution was then applied to compensate for the optical attenuation within both the PA image (background) and the PATTERN image (fluorescent signal). The compensated results were rendered using Amira.

MATLAB was employed to perform additional data analyses, such as k-space visualization and histograms, while ImageJ was used to acquire line profiles of slice images.

### Quantification and signal non-linearity

The PA amplitudes $P(t)$ (as a function of the scanning time $t$) can be expressed as[65]:

$$P(t) \propto \Gamma \eta_{\text{th}} C_0 I \exp(-kI^\beta t), \tag{4}$$

where $\Gamma$ is the Grueneisen coefficient, and $\eta_{\text{th}}$ is the percentage of the absorbed photon energy that is converted into heat. $C_0$ is the concentration of molecules, and $I$ is the excitation intensity. The factor $kI^\beta$ represents the bleaching rate in which $k$ is a constant factor and $\beta$ is the intensity power dependence with $\beta \ge 0$. In PATTERN, a 3D image is computed by integrating a number of 2D frames, mathematically:

$$A(t) \propto \int_t^{t+T} P(\tau) d\tau \tag{5}$$

Here $A(t)$ is the amplitude of the signal acquired by the multiangle fusion process, and $T$ is the scanning time of a single cycle.

Collectively, we get:

$$A(t) \propto \frac{\Gamma \eta_{\text{th}} C_0 I}{k I^\beta} \left( 1 - \exp\left( -k I^\beta T \right) \right) \exp\left( -k I^\beta t \right) \tag{6}$$

This equation represents the exponential decay of raw PA signals over the entire scanning procedure (including many cycles), a relationship that has been experimentally validated and applied in our study. Furthermore, the interconnection between amplitude and bleaching, mediated by the excitation intensity, offers a methodology for calibrating the parameter $\beta$ using the data presented in Supplementary Fig. 6 through the subsequent steps:

(1) Fit $A(t)$ of different positions with exponential function $A(t) = a \cdot \exp(-bt)$, obviously we get

$$k I^\beta = b, \tag{7}$$

and

$$\frac{\Gamma \eta_{\text{th}} C_0 I}{k I^\beta} \left( 1 - \exp\left( -k I^\beta T \right) \right) = a \tag{8}$$

(2) Calculate the equivalent excitation intensity $I_{eq} \propto I$:
Substitute the Eq. 7 into Eq. 8, we can calculate excitation intensity $I$ using:

$$I = \frac{1}{\Gamma \eta_{\text{th}} C_0} \cdot \frac{ab}{1 - \exp(-bT)} \tag{9}$$

Notice that $\Gamma$ and $\eta_{\text{th}}$ are constants for the same type of protein, and $C_0$ was kept as a constant for the whole tube, we define the equivalent excitation intensity $I_{eq}$ as fellow:

$$I_{eq} = \frac{ab}{1 - \exp(-bT)} \tag{10}$$

Obviously, we get $I \propto I_{eq}$.

(3) Calculate the intensity power dependence factor $\beta$ using least squares method:

$$\begin{bmatrix} \ln(k) - \beta \ln\left( \Gamma \eta_{\text{th}} C_0 \right) \\ \beta \end{bmatrix} = \left( \mathbf{X}\mathbf{X}^T \right)^{-1} \mathbf{X} \ln(\mathbf{b}_{n \times 1}), \tag{11}$$

in which $\mathbf{X}$ is a matrix of 2 by n:

$$\mathbf{X} = \begin{bmatrix} \mathbf{1}_{1 \times n} \\ \ln\left( \mathbf{I_{eq}} \right) \end{bmatrix}_{2 \times n} \tag{12}$$

By calibrating the nonlinearity parameter $\beta$, we developed a methodology for quantifying $C_0$, as detailed in the following steps:

(1) Use PATTERN approach to get PATTERN signal amplitude $a$, bleaching rate $b$ and PA signal (background) of a voxel.
(2) Calculate the equivalent molecule concentration $C_{eq} \propto C_0$:

Eliminate excitation intensity $I$ with Eq. 7 and Eq. 8 and extract $C_0$:

$$C_0 = \frac{k^{\frac{1}{\beta}}}{\Gamma \eta_{\text{th}}} \cdot \frac{ab^{\frac{\beta-1}{\beta}}}{1 - \exp(-bT)} \tag{13}$$

Notice that $\Gamma$, $\eta_{\text{th}}$, $k$ and $\beta$ are all constants for the same type of protein, we define equivalent molecule concentration $C_{eq}$

as follow:

$$C_{eq} = \frac{ab^{\frac{\beta-1}{\beta}}}{1 - \exp(-bT)}, \tag{14}$$

which is related to the real molecule concentration ($C_{eq} \propto C_0$).

## Registration

Cross-modal 3D data registration was performed using the open-source software ANTs. First, a PA template volume was generated by additional scan cycle results with excitation at 800 nm or unmixed PA image results. The 800 nm result was favored over the unmixed PA result due to lower noise contamination. When scanning time was limited strictly (i.e., experiment in Fig. 5), the unmixed PA result was used. The light-sheet, fMOST, and PATTERN results were initially resampled to a voxel size of $12.5 \times 12.5 \times 12.5\ \mu m^3$. Then, the 'ElasticSyn' transform type was employed to align the light-sheet/ FMOST results with the PA template. Additionally, the MRI data were resampled to $75 \times 75 \times 75\ \mu m^3$ and aligned with the PA template. To align the confocal signal with the 3D PATTERN results, well-trained human experts manually selected a corresponding slice in the PATTERN data. Next, 2D registration between the DAPI staining channel in confocal microscopy and the matching position of the PA template volume was performed using MATLAB. This yielded a transformation matrix that was subsequently applied to the iRFP signal in confocal microscopy. For transcriptome analysis, the chosen tissue slices underwent hematoxylin-eosin (HE) staining prior to the transcriptome test. Manual alignment of PATTERN data and HE staining results was performed by human experts to guide the alignment of transcriptome results.

The brain atlases of mice, rats, ferrets, and marmosets were aligned with the PA template to obtain a transfer matrix that was applied to annotate atlas locations and calculate the brain region volumes[46,66–68]. The volume of virus infection was derived by counting the number of voxels whose fluorescence results were above a pre-defined threshold.

## Bleaching extent and PATTERN sensitivity

The degree of photobleaching is influenced by local optical fluence and the concentration of fluorescent proteins. We modulated photobleaching by varying the light fluence at different positions within the scattering media. The optical properties of the phantom were designed to mimic those of brain tissue[69] In our experiment, we used a suspension of intralipid-30% with a v/v ratio of 3.6% to imitate brain tissue. We placed a PTFE tube filled with purified fluorescent proteins, which had an inner diameter of 0.3 mm, in the intralipid suspension. We tested three types of fluorescent proteins: iRFP713, mScarlet, and SNIFP, excited at wavelengths of 690 nm, 532 nm, and 697 nm, respectively.

Due to surface tension, the surface of the suspension exhibited a slight curvature. To calculate the distance, we selected a relatively flat portion of the surface (Supplementary Fig. 6a). In the image, the tube was manually segmented and served as the reference standard in our evaluation, providing reigon of interest (ROI) for subsequential analyse as well. We calibrated the bleaching extent against the noise floor of our system, defined as the extent of bleaching to noise ratio (BNR). In this context, "bleaching extent" is defined as the magnitude of signal reduction:

$$BNR = 20\lg\left( \frac{\max_t(PA\ amplitude) - \min_t(PA\ amplitude)}{\text{std}(noise)} \right) \tag{15}$$

The standard deviation was estimated using a manually selected region that did not contain fluorescent proteins.

Common accuracy analyses involving receiver operating characteristic (ROC) curve and The area under the curve (AUC) were performed.

For both the traditional method and the PATTERN method, we characterized the sensitivity using SNR. Signals outside the ROI were confirmed to be generated from the system noise, and inside was the superposition of protein signals and system noise. We computed SNR according to the following equation:

$$SNR = \frac{P_{in\ ROI} - P_{out\ ROI}}{P_{out\ ROI}} \qquad (16)$$

where $P$ is the power of the corresponding signal, calculated by

$$P_{region} = \frac{\sum_{region} pixel\ value^2}{pixel\ number} \qquad (17)$$

### Rotation order

The rotation order of PATTERN was designed and verified via simulation using MATLAB (Supplementary Figs. 7, 8). All parameters used in the simulation were consistent with those employed for the real samples. For comparison purposes, a sequential scan scheme, −90°, −84.375°…, 0°…, 84.375°, was simulated as the control group. During the simulation, we assumed that a point source ($1 \times 1$ pixel at the center of a $256 \times 256$ grid) was bleached during a scan cycle. We assessed the image degradation resulting from photobleaching by comparing the reconstructed image with the ground truth using cosine similarity. Next, a line source ($1 \times 256$ pixels at the middle of a $256 \times 256$ grid) was bleached during an eight-cycle scan. The center pixels were counted to plot the 'bleaching curve'. By altering the orientation of the line source, we generated a series of distorted bleaching curves. The envelopes of these curves, plotted alongside the actual bleaching curve, demonstrate that distortion exacerbates with increasing bleaching rates. Utilizing these distorted bleaching curves, we quantified the estimation errors for both the PATTERN signal and the bleaching rate, presenting them as an upper limit of the relative error.

### Dataset construction

The brain samples were first subjected to PATTERN to collect the iRFP713 signal prior to sectioning and confocal microscopy to collect the EGFP signal. The two modalities were then equally normalized. Well-trained human experts would subsequently manually annotate the artifacts in the PATTERN signals, based on the results of confocal microscopy, which serve as the reference. In addition, this labeling method is able to avoid the generation of false positives, yet there still exists the potential for unlabeled artifacts, which can be further improved. The artifact-free images after correction were matched with the original data and incorporated into the dataset. For each brain, approximately 100 pairs of coronal sections from different locations were generated, and each dataset for a distinct brain region consisted of data from at least three brains. The PATTERN signal processing and dataset generation were managed in Amira 2019, and the confocal signal was processed in ZEISS ZEN3.6.

### DnCNN method

A deep-learning method based on denoising convolutional neural networks (DnCNN) was adopted to remove the artifacts. The network is divided into three parts: a layer of Conv+ReLU, several layers of Conv+BN+ReLU, and a layer of Conv, with a total of 20 layers of three parts. During the training phase, only a single labeled image was input into the network each time, and the network parameters were updated. The neural network implicitly learned the artifacts of each image to achieve the distribution of the artifacts across the entire dataset. The Mean Squared Error (MSE) loss function was employed as the training criterion over 50 epochs, during which the network effectively filtered out pure artifacts to generate the output. For each brain region, a dataset consisting of more than 100 pairs of images with and without artifacts was used for training, and more than 50 pairs for testing. Some of the representative hyperparameters used during training are provided below: batch size = 1, training epochs = 50, and learning rate = $1 \times 10^{-3}$. When the epoch reached 30, the learning rate was reduced by a factor of 10. To make the network more generalized, traditional data augmentation strategies were also applied, including flipping, rotating, and intensity changes. During the evaluation phase, images with artifacts were inputted, and the trained DnCNN output the pure artifacts of the image. Subtracting the input and output could obtain a clean image free of artifacts. Three indicators were used to measure the artifact-removal effect on the DnCNN method and the deconvolution method. GT recall represents the true signal retention rate; noise recall is the artifact removal rate; MSE loss represents the difference between the image after removing artifacts and the real image.

### Animals

Throughout the development, testing, and application process of our entire system, we used various animals including male and female mice, male and female rats, male and female ferrets, and a female marmoset. All experimental procedures of rodents and ferrets were approved by the Institutional Animal Care and Use Committee (IACUC) of Tsinghua University and were performed using the principles outlined in the Guide for the Care and Use of Laboratory Animals of Tsinghua University. All wild-type C57BL/6J mice, BALB/c mice, SD rats, and transgenic mice were purchased and maintained under standard conditions by the Animal Research Center of Tsinghua University. Animals were housed five (for mice) or two (for rats) per cage with free access to food and water while under a 12-h light-dark cycle (light on from 7 p.m. to 7 a.m.). The AD rat we used is the wild-type SD (Sprague Dawley) rats, and AppNL-G-F rats were bred in the animal facility of Tsinghua University. For the fMOST brain and AD rat brain, only the contour information of the brain is addressed in this article, further physiological investigations will be reported in future studies. Transgenic mice were genotyped through PCR using genomic DNA and Jackson Laboratories-provided primers. All studies and experimental protocols were approved by the Institutional Animal Care and Use Committee (IACUC) at Tsinghua University and the animal protocol number is 21-ZY1. Most of the data came from separate samples. Especially, the imaging data from a brain that was injected with AAV-hsyn-iRFP-EGFP in AI was used in Fig. 3d for signal comparison and Fig. 4b–d for visualization of AI projectome. The imaging data from a brain which was in was injected with AAV-hsyn-iRFP-EGFP in dSub was both used in Fig. 4f, g for visualization of dSub projectome and Fig. 5j, k for signal correction.

The marmoset brain sample we used was from Gao Lixia's lab at Zhejiang University. It was housed, maintained, and bred at the Zhejiang University Interdisciplinary Institute of Neuroscience and Technology (ZIINT) Non-Human Primate Center located at the Huajiachi Campus, Hangzhou, Zhejiang Province, China. All experimental procedures were approved by the Zhejiang University Animal Care and Use Committee.

### Viral constructs

The AAV2/9-hsyn-iRFP713, AAV2/9-hsyn-EGFP, AAV2/9-hsyn-DIO-iRFP, AAV2/9-hsyn-DIO-EGFP, AAV2/9-hsyn-EGFP-iRFP, AAV2/9-hsyn-mScarlet, AAV2/9-hsyn-Cre, AAV.PHPeB-GfaABC1D-iRFP713, AAV.PHPeB-GfaABC1D-EGFP (OBiO), CAV-CMV-Cre (WZ Biosciences Inc.), AAVRetro-Cre-mcherry (Taitool), AAV2/9-U6-shRNA(c-Fos)-CMV-iRFP713 (Brain Case), The viral concentration of all viruses was adjusted to $2–5 \times 10^{12}$ viral genomes (vg)ml$^{-1}$ for injection. Viruses were subdivided into aliquots and stored at −80°C until use.

## Stereotactic injection and optical fiber or steel wire implant

Mice were anesthetized with 2.5% avertin in saline (350 mg/kg, 350 ul/25 g). Bilateral craniotomies were performed using a 0.5 mm diameter drill, and corresponding volumes of virus was injected into the target brain region using a 10 μl nanofil syringe controlled by UMP3 and Micro4 system (WPI) with a speed of 60 nl/min. The injections were bilaterally targeted to −2.0 mm AP, ±1.5 mm ML, and −1.4 mm DV, 200 nl, for CA1 region; −1.9 mm AP, ±1.3 mm ML, and −2.0 mm DV, 200 nl, for dDG region; −1.4 mm AP, ± 3.3 mm ML, and −4.84 mm DV, 50 nl, for BLA region; +2.0 mm AP, ± 2.2 mm ML, and −3.1 mm DV, 200 nl, for AI region; −2.6 mm AP, ± 4.5 mm ML, and −3.9 mm DV, 100 nl; −2.6 mm AP, ± 4.5 mm ML, and −3.9 mm DV, 100 nl, for LEC region; +0.3/−0.1/−0.5/−0.9/−1.3 mm AP, ±1.5 mm ML, and −1.0 mm DV, 5 sites each side, 400 nl each injection site, for motor cortex. In some experiments, we also used small volume injections to achieve partial infection. After the injection, the needle remained in place for 10 minutes to ensure that the virus spread to the targeted area before it was slowly withdrawn. For intraspinal injection, mice were first anesthetized by 2.5% avertin in saline. Fur on the back were shaved and an incision was made along the rostral-caudal axis to expose the intrathecal spaces corresponding to L2-L6. Muscle was appropriately cut off to expose the injection site. Two injections were made at both sides of intrathecal spaces. 800 nl virus was injected per injection at the speed of 5 nl/sec below the surface of the spinal cord 300 to 500 μm. The needle was left 3 min after injection.

For mice optic fibers and steel wire implanting, a Doric fiber-optic patch cord (200 μm core diameter; Doric Lenses) or a steel wire (80 μm core diameter) was inserted into the brain with DiR dye (DiIC$_{18}$(7); 1,1′-dioctadecyl-3,3,3′,3′-tetramethylindotricarbocyanine iodide) on its surface. A surgical screw was threaded into the skull to provide an extra anchor point. Dental cement (Teets Cold Cure; A-M Systems) was applied to securely fix the optical fiber implant. After surgery, the mice were allowed to recover for at least 2 weeks before conducting all subsequent experiments.

## Western blotting

Isolated hippocampi or other brain regions were homogenized with cell lysis buffer (Beyotime, Cat. No. P0013) with protease inhibitors. The protein concentration was determined using the BCA protein assay kit (Beyotime, Cat. No. P0012). For the detection of the levels of total actin, AMPK, PAMPK, PMK2, PPMK2, 7 μl of homogenates was separated by SurePAGE (GenScript, Cat.No. M00665) and transferred to nitrocellulose membranes (Pall Corporation, Cat. No. 66485). Membranes were blocked with BSA solution (5% BSA in TBS and 0.1% tween 20) for one hour at room temperature. Subsequently, membranes were individually incubated with primary antibodies against actin, AMPK, PAMPK, PMK2, PPMK2, (mouse anti-β-actin, Biodee, DE0620, RRID:AB_2737288, 1:8000; rabbit anti-AMPK, Cell Signaling Technology, CST2532, RRID:AB_330331, 1:1000; rabbit anti-PAMPK, Cell Signaling Technology, CST2535, RRID:AB_331250, 1:1000; rabbit anti-PKM2, Cell Signaling Technology, CST3198, RRID:AB_2252325, 1:1000; rabbit anti-PPKM2, Cell Signaling Technology, CST3827, RRID:AB_1950369, 1:1000) overnight at 4 °C. All the HRP-conjugated secondary antibodies (horse Anti-mouse IgG, HRP-linked Antibody, Cell Signaling Technology, 7076 s, RRID:AB_330924, 1:2000; goat Anti-rabbit IgG, HRP-linked Antibody, Cell Signaling Technology, 7074 s, RRID:AB_2099233, 1:2000) were used at 1:2000 dilutions for membranes incubation at room temperature for one hour. Image quantification analysis of bands of the western blots were calculated by the ImageJ software (National Institutes of Health).

## Fear conditioning training

The fear conditioning test was conducted by using the HABITEST Modular Behavioral Test System. A Coulbourn Habitest chamber (27 cm × 28 cm × 30.5 cm) had a stainless-steel rod floor that was connected to a shock generator in a sound-attenuating box. In the five-shock contextual fear conditioning task, the footshock (2 s, 0.8 mA) was delivered at 180 s, 240 s, 300 s, 360 s, and 420 s. Mice remained in the conditioning chamber for a total of 450 s.

## Histology

Tissue Preparation. Mice were deeply anaesthetized with 2.5% Avertin in saline (500 mg/kg, 500 ul/25 g) and transcardially perfused, first with cold PBS and then with cold 4% paraformaldehyde (PFA). For subsequent western blot or spatial transcriptome analysis to do the brain, the mice were perfused with cold aCSF only. The brains were extracted, and postfixed in PFA overnight at 4 °C. Next, the brains were coronally sectioned at 60-μm sections through the targeted brain regions using a vibrating blade microtome.

Immunohistochemistry. Floating sections were used in all the following immunostaining experiments. Unless otherwise stated, all incubations occurred at the room temperature. For Iba-1 immunostaining, sections were washed 3 times in 1 × PBS and rinsed in 1% Triton X-100 for 15 min before a blocking step in PBS with 0.5% Triton X-100 and 10% normal donkey serum for 1 h. Incubation with primary antibody was performed at 4 °C for 48 h (rabbit anti-Iba-1, Cell Signaling Technology, 17198, RRID:AB_2820254, 1:1000) in PBS with 0.5% Triton X-100 and 1% normal donkey serum. Sections were then washed 3 times in PBS and incubated with secondary antibody (donkey anti-GP IgG Cy3, 706-165-148, RRID: AB_2340460, Jackson ImmunoResearch, 1:500) for 2 h. Sections were then washed 3 times in PBS and incubated in DAPI that was diluted in PBS (1:3000) for 10 min. Next, sections were again washed in PBS for 3 times before being mounted onto slides and coverslipped with anti-fade mounting medium (Invitrogen).

Fluorescence imaging. Fluorescence was detected using an Olympus SpinSR spinning disk confocal microscopy imaging system for Figs. 3d, 4d, and Supplementary Fig. 15 with the aid of the Olympus software. A Zeiss axio scan z1 was used for imaging for Figs. 3o, 6c–e, and Supplementary Figs. 14 and 16 with the aid of the ZEN software (black edition). Images were acquired with a 10× objective and colocalization was confirmed by a 3-D reconstruction of z series images.

## fMOST

Protocols for Fluorescent micro-optical sectioning tomography (fMOST) imaging were based on previously published methods with minor modifications[50]. Briefly, the mice were intracardially perfused with 4% paraformaldehyde in 0.01 M phosphate-buffered saline, rinsed with PBS buffer solution, gradient dehydration with ethanol, gradient penetration with LR White resin and embedded with thermostatic polymerized resin at 45 °C to achieve complete morphological maintenance characteristics and good mechanical cutting performance. TDI fMOST two-color imaging system was used to cut through the embedded mouse brain with a z-axis resolution of 1 μm and a lateral resolution of $0.35 \times 0.35$ μm$^2$. Imaging buffer containing propyl iodide (PI) was used to obtain cell density information under 561 nm excitation light. The final presented result was down-sampled to $1.75 \times 1.75 \times 50$ μm$^3$. The sampling and fMOST imaging work was performed by Song-Hai Shi lab at Tsinghua University.

## Tissue clearing

uDISCO. The ultimate three-dimensional imaging of solvent-cleared organs (uDISCO) method was performed using the previously published methods[51].

Briefly, immediately following transcardiac perfusion with PBS and 4% PFA, brains were fixed in 4% PFA for 24 h. The fixed samples were incubated in tert-butanol with a gradient concentration at 34–35 °C, followed by immersion in DCM for 45–60 min at room temperature. Then, samples were incubated in BABB-D at room temperature until transparency was achieved. Samples could be stored in BABB-D at room temperature in dark before imaging. The sampling

step was performed by the instrument platform at Center of Biomedical Analysis, Tsinghua University.

**iDISCO.** Immunolabeling-enabled three-dimensional imaging of solvent-cleared organs (iDISCO) method was performed on the AD rat brain. The protocol was adapted from a previous study[52] with some modifications. All the procedures of tissue clearing and antibody incubation were appropriately extended to enhance the penetration of antibodies into the rat brain. The primary antibody is a murine chimaeric IgG2a/κ antibody that targets Aβ40 and Aβ42[70] with a volume of 100 μl before diluting, per sample. The second antibody is Alexa Fluor 647 donkey anti-mouse IgG (Invitrogen, Lot:819571) for a volume of 100 ul before diluting per sample. These antibodies were generously provided by Bai Lu's lab at Tsinghua University.

**PEGASOS.** Polyethylene glycol-associated solvent system (PEGASOS) was used as the tissue clearing technique according to previously published methods[53]. Immediately following transcardiac perfusion with PBS and 4% PFA, brains were fixed in 4% PFA for 24 h and then treated with Quadrol decolorization solution for 48 h at 37 °C. Next, samples were immersed in gradient delipidation solutions at 37 °C for 2 days under constant shaking, followed by dehydration solution treatment for 2 days. Finally, samples were immersed in BB-PEG clearing medium for a minimum 1 day until reaching transparency. Agar embedding was performed during the sample preparation of Abnormal Brain in order to maintain morphological stability. Samples were then preserved in the clearing medium BB-PEG at room temperature. The sample preparation step was conducted at the instrument platform at Center of Biomedical Analysis, Tsinghua University.

### Light-sheet microscopy
Whole brain fluorescence z-stack imaging was performed using a light sheet fluorescence microscope (Zeiss Lightsheet Z.1, Imaging objective 5X/0.16, Illuminating objective 5X/0.1). Stitching and three-dimensional reconstruction were performed with Imaris9.7.2 (Bitplane).

### MRI experiment on mouse
Briefly, the mice were scanned in horizontal MRI scanners (9.4 T/30 cm, Bruker BioSpec 94/30, Germany, software ParaVision for MRI acquisition). Anesthesia was induced with 3% isoflurane (R5835, RWD Life Science) and maintained during scanning using 1.5% isoflurane supplemented with 93% oxygen. The body temperature was kept at 37 °C through the circulating water tank (SC100-S5P, THERMO HAAKE, USA), and the respiratory status is monitored in real time through the ERT module (Model 1030, SA Instruments Inc., USA). A T2-weighted structural image was acquired by using T2 _Turbo _ RARE sequecne with following parameters: Number of slices = 59, TR = 5849 ms, TE = 33.79 ms, flip angle = 90°, FOV = $16 \times 15 \, mm^2$, matrix size = $212 \times 212$, slice thickness = 0.3 mm ETL = 10, NEX = 5, TA = 10 min and 14 s. Total imaging time including animal positioning was around 1 h. The experiment was performed at the Tsinghua Laboratory of Brain and Intelligence, Beijing.

### Visium sequencing samples preparation
In order for the virus to be fully expressed, the conditioned fear experiment is performed three weeks after the injection. And the perfusion was scheduled one hour after the CFC to sequence c-Fos and related genes. The mice were only perfused with cold aCSF. The brain was separated rapidly and imaged by PATTERN. For the convenience of cutting the brain, we first pre-cooled the brain on dry ice for a few seconds. Then, the target brain was cut into the right shape and put into the dry ice waiting for the next sample. A set of samples was first

embedded in an 8 mm × 8 mm × 4 mm silicone tank to limit the total volume and was finally embedded in a full-size box.

### Visium sequencing libraries preparation
The Visium Spatial Gene Expression Slide & Reagent kit (10X Genomics) was used to construct sequencing libraries according to the Visium Spatial Gene Expression User Guide (CG000239, 10X Genomics). A 10um frozen tissue section was placed on one of the Visium gene expression slide capture areas in a slide. After tissue Hematoxylin and Eosin (H&E) staining, bright-field images were acquired as described in the Spatial Transcriptomics procedure. Tissue permeabilization was performed for an optimal minute, as established in the TO procedure. Then reverse transcription experiment was conducted and sequencing libraries were prepared following the manufacturer's protocol.

### Sequencing
Sequencing was performed with a Novaseq PE150 platform according to the manufacturer's instructions (Illumina) at an average depth of 300 million read-pairs per sample.

### Seurat analysis
The Seurat package was used to perform gene expression normalization, dimensionality reduction, spot clustering, and differential expression analysis. Briefly, spots were filtered for minimum detected gene count of 100 genes. Normalization across spots was performed with the SCTransform function and 3000 highly variable genes were selected for principal component analysis. For spot clustering, the first 20 PCs were used to build a graph, which was segmented with a resolution of 0.5. The Wilcox algorithm was used to perform differential gene expression analysis for each cluster via the FindAllMarkers function. Genes with fold change >2 and adjust $p < 0.05$ were defined as significantly differentially expressed genes. Notably, Seurat::FindAllMarker was used to find differential genes expression in Fig. 6i. A two-sided Wilcoxon Rank Sum test was performed and the $p$ value was adjusted based on bonferroni correction.

### Statistics
Statistical analyses were performed in GraphPad Prism. All data were analyzed with an unpaired $t$ test, one-way ANOVA, or two-way ANOVA where appropriate. The data were shown as the mean ± SEM. and n.s. (ns) indicates non-significance ($p > 0.05$). The significance level were set to $P = 0.05$. Significant levels for comparison: $^*p < 0.05$; $^{**}p < 0.01$; $^{***}p < 0.001$; $^{****}p < 0.0001$.

### Schematic
Figures 3e, 4a, 5a, 5h, and 6a were created with BioRender.com.

### Reporting summary
Further information on research design is available in the Nature Portfolio Reporting Summary linked to this article.

## Data availability
Raw data of spatial transcriptome have been deposited in the National Center for Biotechnology Information's Sequence Read Archive with accession numbers PRJNA1091401. The processed spatial transcriptome analysis is available at https://github.com/CaA2318777/PATTERN/tree/main/SpatialTranscriptome. Processed PA imaging data generated in this study have been deposited in the figshare database under the accession link https://figshare.com/s/c0f6139f729b97b21028. Source data are provided as a Source Data file. Raw data of PA and other imaging modality are available upon responsible request due to their large size. Source data are provided with this paper.

## Code availability

We have uploaded all the code produced in this project to a public database. The link to access them is https://github.com/CaA2318777/PATTERN.

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

## Acknowledgements

We greatly thank Xiaolin Wang, Efan Wang, Qiangqiang Zhang, and Jian Ma from Song-Hai Shi Lab at Tsinghua University for sharing the brain samples. We greatly thank Yixiao Gao and Dongqin Cai from Tsinghua University for their help with the sample preparation of the marmoset brain. We greatly thank Shudan Wang from Bai Lu lab at Tsinghua University for sharing the AD rat brain samples. We also thank all the members of the Bo Lei Group, Yi Zhong Lab, and Cheng Ma Lab for their support. This work was supported by grants from the Tsinghua University Initiative Scientific Research Program (20221080072, to C.M.), the National Science Foundation of China (32021002, to Y.Z. and 61971265, to C.M.), the STI2030-Major Projects (2022ZD0204900, to Y.Z.), National Science and Technology Major Project (2022ZD01163013, to B.L.), the Tsinghua-Peking Center for Life Sciences (to Y.Z.), the Initiative Scientific Research Program of Institute for Intelligent Healthcare of Tsinghua University (to C.M.), the Innovation Project of Tsinghua-Foshan Institute of Advanced Manufacturing (to C.M.), the National Key R&D Program of China (2022YFC3400600-2 to P.X.), and the National Natural Science Foundation of China (21927813 to P.X.)

## Author contributions

C.M. and B.L. conceptualized the project. H.Y., Y.C., Y.L., Y.N., C.M., and B.L. designed the study. H.Y., Y.C., Y.L., Y.N., Y.Z., S.S., C.M., and B.L. wrote and reviewed the paper. H.Y., Y.N., and H.C. conducted biological experiments. L.G. and X.L. conducted experiments on marmoset. X.W., Y.L., and H.D. did the feasibility study of PATTERN Y.C. and H.D. built the system. H.Y., Y.N., J.M., Y.C., and Y.L. analyzed all imaging data and prepared the data set. P.X. and F.X. provided protein samples for testing with PATTERN. S.D. analyzed transcriptomic data. Y.N., B.L., and M.Y. designed and performed de-noising experiments. All authors discussed and edited the manuscript.

## Competing interests

C.M. has a financial interest in TsingPAI Technology Co., Ltd., which provided the data acquisition unit (DAQ) used in this work. Other authors declare no conflicts of interest.

## Additional information

[1]Department of Electronic Engineering, Beijing National Research Center for Information Science and Technology, Tsinghua University, Beijing 100084, PR China. [2]Institute for Intelligent Healthcare, Tsinghua University, Beijing 100084, PR China. [3]School of Life Sciences, Tsinghua University, Beijing 100084, PR China. [4]IDG/McGovern Institute of Brain Research, Beijing 100084, PR China. [5]Tsinghua-Peking Center for Life Sciences, Tsinghua University, Beijing 100084, PR China. [6]School of Computer Science, Xi'an Jiaotong University, Xi'an 713599, PR China. [7]School of Life Sciences, Sun Yat-sen University, Guangzhou 510275, PR China. [8]Research Center for Humanoid Sensing, Zhejiang Laboratory, Hangzhou 311100, PR China. [9]School of Medicine, Tsinghua University, Beijing 100084, PR China. [10]Department of Neurology of the Second Affiliated Hospital and Interdisciplinary Institute of Neuroscience and Technology, School of Medicine, Zhejiang University, Hangzhou 310029, PR China. [11]Key Laboratory of Biomacromolecules (CAS), CAS Center for Excellence in Bioma-cromolecules, Institute of Biophysics, Chinese Academy of Sciences, Beijing 100101, PR China. [12]College of Life Sciences, University of Chinese Academy of Sciences, Beijing 100101, PR China. [13]Canterbury School, New Milford, CT 06776, USA. [14]Beijing Academy of Artificial Intelligence, Beijing 100084, PR China. [15]These authors contributed equally: Yuwen Chen, Haoyu Yang, Yan Luo, Yijun Niu. ✉e-mail: cheng_ma@tsinghua.edu.cn; b.lei.2022@hotmail.com

