## [Peer Review File · Nature Communications]

Photoacoustic Tomography with Temporal Encoding
Reconstruction (PATTERN) for Cross-Modal Individual Analysis
of the Whole BrainREVIEWER COMMENTS

Reviewer #1 (Remarks to the Author):

This paper presents a novel implementation of photoacoustic tomography (PAT) that relies on temporal encoding instead of spectral encoding to achieve molecular imaging using exogenous chromophores (fluorophores). The basic instrumentation of using a half-circle sensor array for 3D imaging is not all that novel, and rotating and translating the array with spatial compounding to improve image quality and resolution is an incremental improvement. However, the idea of temporal encoding aided by deliberate photobleaching of fluorophores is elegantly simple and can aid in molecular and function photoacoustic (PA) imaging of a variety of readily available fluorophores provided that produce a strong enough PA signal and that the photobleaching is fast enough. The authors have prepared a high-quality manuscript with extensive data using multiple animal models to demonstrate the strengths of their new approach including imaging with genetically encoded probes. As such, I think that this submission is appropriate for this journal. However, several clarifying questions need to be addressed prior to publication. Therefore, I recommend relatively major revisions prior to consideration for publication.

- The main premise here is that photobleaching of fluorophores produces a time-dependent (exponential) decay in the PA signal whereas endogenous chromophores like blood (background) signal remains unchanged over time. However, this is a function of how much incident optical energy is reaching different locations within the biological sample. There appears to some simple correction for local light fluence using a Monte Carlo (MC) simulation. These details are lacking. Furthermore, it is not clear if this MC simulation adequately corrects for optical heterogeneity in real biological samples.
- There is also no analysis of optimal bleaching rates (and its relation to this imaging approach and acquisition rates) as well as how the bleaching varies as a function of distance from surface in real biological materials or what is the minimum PA SNR needed at the onset for the PATTERN approach.
- Exponential fitting for PATTERN – was this the same for all biological models (different length scales) or was this done separately for each animal model?
- PATTERN is not going to provide cell-level resolution, and that may be ok as this approach is complimentary to other destructive techniques. Authors speculate that increasing the center frequency and bandwidth of the transducers may improve resolution. This while true, may not always be good approach for PAT, as low frequencies are equally important for accurate representation of features and shapes. Really, bandwidth is the most important factor and then perhaps PA SNR.
- Please provide a table that compares the acquisition speed, resolution, max depth, and FOV for PATTERN, fMOST, and LSFM (also destructive vs non-destructive, fixed vs fresh) for each biological model for comparison. This might be a nice addition for readers to see the pros and cons at glance.
- Please clearly articulate why the this approach for acquiring PAT data is better than other implementations that use for instance an array of sensors arranged in a hemispherical bowl.

- Multi-angle filter appears to be implemented in K-space as part of the image reconstruction workflow, but this is not clearly explained. This seems to be an important component. Please provide a detailed explanation on how this is implemented.
- How does the current approach compare to filtered backprojection reconstruction in process and results?
- Why use delay-and-sum for the initial reconstruction? This is known to produce low-quality images. Better performing methods might allow 3D volumes with fewer angles and faster acquisition times?
- Fig. 2E, please change the x-axis to units of time.
- P11: “Compared to using intensity alone...a gain of 12 was achieved in sensitivity (Fig. 2f)” this is not clear, nor do I see how the figure shows a factor of 12 improvement in sensitivity. Please clarify.
- P33:” ...aCSF was used to minimize biological activity loss” – what activity is being referred to here, does this mean that the whole excised brain remains viable in aCSF?
- For DnCNN, please provide pertinent details on number of training and testing data, number of hyperparameters, what strategies were used for data augmentation, etc.

Reviewer #2 (Remarks to the Author):

I am highly impressed by the innovative work of Dr. Ma’s team in developing the PATTERN photoacoustic imaging technology. Their manuscript presents a significant advancement in the field, demonstrating the capability of PATTERN for high-resolution, non-destructive, and high-speed imaging of fluorescent proteins in fresh ex vivo brains. The paper is well-written and effectively communicates the technical details and results. I offer several major and minor comments that I believe will further enhance the quality and impact of this excellent study.

Major Comments:

1. Sensitivity Limit of PATTERN:

The authors successfully highlight the substantial enhancement in detection sensitivity achieved through the photobleaching approach in PATTERN. To gain a deeper understanding of PATTERN's performance, I suggest exploring the sensitivity enhancement across different fluorescent proteins. Additionally, investigating its performance in the presence of strong background signals, such as blood, would provide insights into its potential for in vivo applications. A comparison of PATTERN's sensitivity with that of photoswitching-based techniques, which offer reversible sensitivity enhancement, would strengthen the manuscript. Furthermore, an investigation into the extent of photobleaching required to achieve the desired sensitivity, through phantom studies with purified fluorescent proteins in scattering media, would add valuable information.

2. Signal Non-linearity of PATTERN:

The authors' creative use of photobleaching to enhance detection sensitivity in PATTERN raises questions about the non-linear relationship between its signals and local optical fluence. Photobleaching was explored before for PAM as a non-linear approach to enhance the spatial resolution (Physical review letters 112, no. 1 (2014): 014302.). It is innovative to turn photobleaching, an otherwise unwanted side effect, into a useful tool for improving detection sensitivity. Because the photoacoustic rate (or the signal deduction speed) usually has a power dependence on the optical fluence (may be larger than 1 for some strong optical absorbers), the PATTERN signals should have a non-linear dependence on the local optical fluence. To address this, I recommend a quantitative investigation into the non-linear behavior of PATTERN signals such as the signal decay with increasing depth, particularly in deep brain targets with reduced optical fluence. Comparing the non-linear behavior of PATTERN signals to traditional PA and fluorescence signals would shed light on its implications for cross-model validation and analysis.

Minor Comments:

1. Clarification Needed:

- Page 4, line 10: The term "a large missing cone" should be explained more comprehensively for readers who are not experts in imaging, specifically in relation to spatial frequencies (k space).
- Page 27: Further clarification is required regarding how the translation and rotation effectively address the limited view problem in PATTERN, especially concerning the detection of vertical structures along the acoustic axis.

2. Figure and Terminology:

- Page 8, Figure 1a: Consider whether PATTERN should follow the "brain retrieval".
- Background Signals: Please provide additional details about the PA contrast of background signals, including their composition (somas) and potential sources (lipids, cell plasma, water).

3. Reconstruction and Processing:

- Positive PA Signals: Elaborate on the specific technique used for image reconstruction (e.g., DAS) and whether the Hilbert transform is applied to eliminate negative signals, along with its potential impact on spatial resolution.

4. Jumping Sequence and Quantitative Analysis:

- The innovative jumping sequence for reducing k -space non-uniformity warrants further discussion, particularly regarding its potential implications on the quantitative analysis of PATTERN signals.

5. Deep Learning and Elevational Resolution:

- The use of DnCNN for enhancing PATTERN signals raises an interesting possibility of leveraging multi-angled scanning results as the 'ground truth' to improve the elevational resolution of single-angled scanning results. Exploring this idea further could potentially lead to enhanced elevational resolution without extensive scanning or improved results with reduced scanning angles.

In conclusion, I want to commend the authors on their outstanding work in developing PATTERN, a novel photoacoustic imaging technology with promising implications. I believe addressing the major and minor comments mentioned above will not only refine the manuscript but also contribute to its scientific rigor and potential impact. I highly recommend considering these revisions to further elevate the clarity and comprehensiveness of this remarkable study, positioning it as an excellent fit for publication in Nature Communications.

Reviewer #3 (Remarks to the Author):

In the work with the title 'Photoacoustic Tomography with Temporal Encoding Reconstruction (PATTERN) for Cross-Modal Individual Analysis of the Whole Brain', Yang et al. introduce an imaging platform (PATTERN) for the non-invasive, fast, three-dimensional ex-vivo imaging of small animal brains. The manuscript is generally well-written with high quality figures but I think some more clarifications and additional work (major revision) would be needed:

Introduction

- The authors present their method as an improvement compared to optical imaging methods without explaining why optical imaging would be accepted as the gold-standard small animal imaging method. In other words, a brief overview (with main pros and cons of each one) regarding other imaging technologies (e.g., magnetic resonance imaging, MRI) available for whole-brain small animal imaging is needed.

- Line 24: I suppose that the word 'metabonomic' should be changed into 'metabolomic'.

- Indeed, the method provides high quality morphological brain imaging. However, based on the fact that several studies focusing not only on morphological but also functional brain imaging with photoacoustics have been already published (DOI: 10.1038/s41551-019-0372-9, DOI: <https://doi.org/10.1038/s41377-022-01026-w>, DOI: 10.1016/j.celrep.2019.02.020), I would like to ask the authors whether the system could be also used for functional brain imaging. If yes, could you please provide us with some data? If not, could you please explain more clearly what is the exact novelty of the work? Lower cost? Faster scanning times? Isotropic resolution? Higher resolution? A thorough comparison with other photoacoustic brain studies would be required.

Results

- Near isotropic photoacoustic imaging of PATTERN: The authors provide resolutions of approximately 100-150µm. Based on the fact that several even clinical photoacoustic systems, which would definitely have (or even need) lower resolutions to image human tissues, almost achieve these resolutions (also in real-time), I would like to ask whether the authors believe that the spatial resolutions achieved are enough to image the fine small animal brain structures. What do the other preclinical photoacoustic platforms achieve? For example, what is the smallest biologically- (or disease-) relevant small animal brain structure that could be resolved with the developed system?

- Temporal encoding and unmixing of fluorescent tags by PATTERN: The authors characterize the spectral unmixing 'extremely complex'. I find it absolutely useful to provide more objectified data on the superiority of the temporal unmixing approach followed. Which are exactly the weaknesses of the spectral unmixing approaches?

- Temporal encoding and unmixing of fluorescent tags by PATTERN: I would be grateful to have some more information on how the 'artefacts' were identified? Could you provide extra clarifications on the usual artefacts observed with the technology presented?

- PATTERN-based whole-brain optical imaging: I believe that the best way to characterize the technology developed is photoacoustic and not optical (2nd line). This is why the main comparisons should be done against other photoacoustic (and not optical) imaging platforms.

- PATTERN for visualizing neural connectivity of the brain: Since exploration of neural connectivity is indeed important, the reader should be provided with more information about its importance.

Discussion

- Comparison and combination of PATTERN with other whole-brain optical imaging technologies: As already mentioned, based on the fact that several other photoacoustic brain imaging studies have been already published, I believe that a comparison to them (and not optical imaging techniques) would be needed. Taking into account the findings of the study and the claims of the authors, the comparison (apart from general features) would also focus on information about: 'injection sites of vectors, fluorescence expression intensity or signal loss during sample preparation'.

Methods

- PATTERN: Why did you select a transducer with central frequency of 5.5 MHz? And not a higher-frequency one?

- PATTERN: Which are the optical/absorption properties of the body fluid used in the perfusion module? Would its presence affect the signals measured in the brain? And how?

- Data reconstruction and postprocessing: Could you provide some more information on the light fluence simulation and its compensation scheme, please?

- DnCNN Method: How did you train the neural network regarding the artefacts? Did you manually delineate them? How did you ensure that no information was taken as an artefact?

Response to the Editor

Dear Editor,

We sincerely appreciate the efforts of the editorial team and the constructive comments from the reviewers. We believe that all raised concerns have been addressed through the performance of additional experiments, the incorporation of new data and figures, and the implementation of improvements based on the comments provided in the reviews. Our point-by-point responses to the reviewer's comments are provided below. The original reviewers' comments are indicated in blue, our responses are provided in black, and the revisions are highlighted in red.

Point-by-point response to the reviewers' comments

Reviewer #1 :

(RC: Reviewer's Comment, AR: Author's Response)

RC1.0:

This paper presents a novel implementation of photoacoustic tomography (PAT) that relies on temporal encoding instead of spectral encoding to achieve molecular imaging using exogenous chromophores (fluorophores). The basic instrumentation of using a half-circle sensor array for 3D imaging is not all that novel, and rotating and translating the array with spatial compounding to improve image quality and resolution is an incremental improvement. However, the idea of temporal encoding aided by deliberate photobleaching of fluorophores is elegantly simple and can aid in molecular and function photoacoustic (PA) imaging of a variety of readily available fluorophores provided that produce a strong enough PA signal and that the photobleaching is fast enough. The authors have prepared a high-quality manuscript with extensive data using multiple animal models to demonstrate the strengths of their new approach including imaging with genetically encoded probes. As such, I think that this submission is appropriate for this journal. However, several clarifying questions need to be addressed prior to publication. Therefore, I recommend relatively major revisions prior to consideration for publication.

AR1.0:

Thank you for your invaluable feedback. We deeply appreciate your thoughtful comments, and we have taken each of them into account. Consequently, we have made corrections and supplemented new experimental data to enhance the overall quality of our manuscript. Your insights have been instrumental in refining our work, and we are grateful for the opportunity to improve based on your guidance.

RC1.1:

- The main premise here is that photobleaching of fluorophores produces a time-dependent (exponential) decay in the PA signal whereas endogenous chromophores like blood (background) signal remains unchanged over time. However, this is a function of how much incident optical energy is reaching different locations within the biological sample. There appears to some simple correction for local light fluence using a Monte Carlo (MC) simulation. These details are lacking. Furthermore, it is not clear if this MC simulation adequately corrects for optical heterogeneity in real biological samples.

AR1.1:

During the process of photobleaching, the PA signal decay rate is dependent on the excitation light intensity. The PA amplitude A as a function of the scanning time t has the following relationship (for the derivation of the following equation, please refer to our response to Reviewer 2 on page 35):

$$A(t) \propto \frac{\Gamma \eta_{\text{th}} C_0 I}{k I^\beta} (1 - \exp(-k I^\beta T)) \exp(-k I^\beta t), \quad (\text{R1.1})$$

where Γ is the Grueneisen coefficient, and η_{th} is the percentage of the absorbed photon energy that is converted into heat. C_0 is the concentration of the molecules, I is the incident light intensity, and T is the scanning time of a single cycle. The factor $k I^\beta$ represents the bleaching rate in which k is a constant factor and $\beta \geq 0$ accounts for a generally nonlinear dependence on the incident light intensity. The underlying assumption is that the light intensity I remains constant or changes insignificantly during the bleaching process, indicating the absorption of fluorescent proteins is relatively weak compared to the absorption and scattering of the background, which was discussed in the original manuscript (page 27 lines 17–23, “Currently, temporal encoding schemes typically require a relatively weak bleached signal so that the light fluence distribution inside the sample is not significantly affected...”).

In our algorithm, we fit the PA amplitude decay curve using

$$A(t) = a \cdot \exp(-bt) + c, \quad (\text{R1.2})$$

where the parameter b represents the bleaching rate, a is the PA signal strength of the fluorescent tag which contributes to the useful signals, referred to as “PATTERN signals” in the current work, and c is the PA signal strength of any unbleached background chromophores, referred to as “PA signals” in the current work (page 12 line 23–page 14 line 1, “To prevent any ambiguity, we utilize the term “PA signal/image” in the current work to refer to signals and image features that correspond to the intrinsic tissue background... Conversely, we employ the term “PATTERN signal/image” to refer specifically to signals and image features that are derived from temporal encoding...”). By comparing Eq. R1.1 and Eq. R1.2, the strength of fluorescent tag a can be rewritten as:

$$a = \frac{\Gamma \eta_{\text{th}} C_0 I}{k I^\beta} (1 - \exp(-k I^\beta T)).$$

In biological samples, I is spatially-varying and typically known, thus it is difficult to derive C_0 (which represents the real object) directly from the measured PA signal a .

The PATTERN technology provides a viable means to reconstruct the distribution of C_0 . The location-dependent light intensity can be evaluated by the bleaching rate $b = k I^\beta$ (Figure R 1). Moreover, the nonlinear factor β is a fluorophore-dependent parameter which can be experimentally measured. For example, we have tested that $\beta = 2.64$ for

iRFP713 (Figure R2). Once these parameters are known, the influence of inhomogeneous I distribution can be compensated for. The detailed methodology was illustrated in the Methods Section (pages 38–42) and Supplementary Fig. 5d.

As for the Monte Carlo (MC) simulation, it was not utilized for quantitative calculations but rather for a post-processing procedure aimed at suppressing the excessively bright pixels/voxels on the sample surface. Applying a linear colormap subsequent to the Monte Carlo-based fluence compensation provides a reconstruction result that is more realistic and imbued with physical significance. The simulation procedure has been reported in the Methods Section (page 40 lines 14–22). Additionally, the code for the simulation has been listed at page 67.

Figure R 1 | Bleaching rates of a sample. a, Bleaching rate distribution of a cross-section of a brain. **b**, histogram of the bleaching rate b .

Figure R 2 | Photobleaching rate of iRFP713 shows a power-law dependence on intensity with an exponent of 2.64 ($\beta = 2.64$).

Revisions:

- (1) The approach to compensate for the excitation intensity was added in the Methods Section (page 40 line 26–page 42 line 21).

Due to the extensive nature of the revisions, please refer to the manuscript for details.

(2) The MC simulation procedure was further clarified in the Methods Section (page 40 lines 14–22):

“Subsequently, a 3D Monte Carlo simulation, conducted using MCXLAB⁶⁴ (Supplementary Methods), was employed to simulate the light fluence distribution, considering $\mu_a = 0.005$ and $g = 0.9$ for the brain, and $\mu_a = 0.002$ and $g = 0.98$ for the sample holder (agarose). We utilized a uniform light source covering the inner surface of the sample holder to replicate the experimental conditions. The resulting light fluence distribution was then applied to compensate for the optical attenuation within both the PA image (background) and the PATTERN image (fluorescent signal).”

(3) The methods of image visualization, including the Monte Carlo simulation procedure, are segregated into a distinct paragraph for enhanced clarity (page 40 lines 12–22).

Due to the extensive nature of the revisions, please refer to the manuscript for details.

(4) The codes for MC simulation were added to supplementary methods.

RC1.2:

- There is also no analysis of optimal bleaching rates (and its relation to this imaging approach and acquisition rates) as well as how the bleaching varies as a function of distance from surface in real biological materials or what is the minimum PA SNR needed at the onset for the PATTERN approach.

AR1.2:

Thanks for the reviewer’s comments related to the critical information about minimal detectable bleaching rate (signal reduction speed) and bleaching extent (signal reduction magnitude). In practice, there was no singular 'optimal' bleaching rate; instead, there existed a range of bleaching rates that yielded satisfactory performance. The lower bound of this range was defined by the minimal detectable bleaching rate, constrained by the system noise.

During imaging, the excitation light generates PA waves while simultaneously inducing bleaching in the sample. Consequently, there exists a correlation between the bleaching rates and PA signal-to-noise ratio (SNR), underscoring that the bleaching rate cannot be regarded as an independent parameter. In practice, the bleaching process cannot be excessively slow, as achieving a large reduction of the signal is essential for high detection sensitivity (Figure R 4). However, the bleaching rate cannot be excessively high either, as this would result in non-uniform k-space filling after a scan cycle, leading to a poorly shaped point spread function (Figure R 3, copied from Supplementary Fig. 7).

It is noteworthy that within a given sample, the bleaching rate varies due to the non-uniform distribution of light intensity (Figure R 1), posing a challenge in precisely controlling the bleaching rate to a single "optimal" value. Fortunately, based on the aforementioned

analysis, achieving an optimal bleaching rate is not imperative, as long as the rates within the sample fall within an acceptable range. The primary objective was to ensure that features with a minimal bleaching rate still exhibited a detectable bleaching extent. We have demonstrated that our method performs well across a wide range of bleaching rates encountered in all tested samples.

Figure R 3 | Optimization of rotation order for image quality. **a**, Diagram of the sequential scan of a point source which is exponentially bleached during the scan cycle. **b**, Reconstruction results from the scan and related FFT graph. **c**, **d**, The same as (a) and (b) but with jump scan. **e**, Reconstruction results with different bleaching rates. **f**, Cosine similarity between the point source image and the reconstructed image using sequential and jump scan.

To address the last two questions, we conducted a phantom experiment. The phantom provided a more controlled environment to study how the bleaching rate varied with depth. The optical properties of the phantom were designed to mimic those of brain tissue¹. In our experiment, we used a suspension of intralipid-30% with a v/v ratio of 3.6% to imitate brain tissue. We placed a PTFE tube filled with purified iRFP713, which had an inner diameter of 0.3 mm, in the intralipid suspension. Due to surface tension, the surface of the suspension exhibited a slight curvature. To calculate the distance, we selected a relatively

flat portion of the surface. In the image, the tube was manually segmented and served as the reference standard in our evaluation (Figure R 4a).

The measured bleaching curve exhibited slower rates with increasing distance from the surface (Figure R 4b), exhibiting an almost linear relationship between bleaching rate and depth (Figure R 4c). This phenomenon is likely due to the uneven distribution of the excitation laser beam. We also studied quantitatively how the bleaching rates varied as a function of incident intensity ($b = kI^\beta$). We were able to fit the parameter β using the experimental data (Figure R 2).

Figure R 4 | Optimization of bleaching rates. **a**, Imaging of a PTFE tube fulfilled with purified iRFP713, the inset shows the experimental schematic. **b**, Bleaching curves corresponding to different depths are indicated by the arrows in (a). **c**, Bleaching rate plotted against depth. **d**, Receiver operating characteristic (ROC) curve of three depths indicated by the arrows in (a). **e**, Area under the curve (AUC) of voxels with different bleaching extent.

Concerning the "minimum PA SNR needed at the onset for the PATTERN approach," what matters is the bleaching extent-to-noise ratio (BNR) rather than the absolute PA SNR. In this context, "bleaching extent" is defined as the magnitude of signal reduction:

$$BNR = 20 \lg \left(\frac{\max_t(PA \text{ amplitude}) - \min_t(PA \text{ amplitude})}{std(noise)} \right). \quad (R1.3)$$

The standard deviation was estimated using a manually selected region that did not contain fluorescent proteins.

To evaluate the precision of PATTERN, the receiver operating characteristic (ROC) curve at different depths was calculated. Remarkably, even at a penetration depth of 15.6 mm (indicated by the yellow curve) and with minimal bleaching (~10%), the ROC curve consistently outperformed random prediction (black dashed line), validating the efficacy of the PATTERN approach (Figure R 4d). The fitting analysis revealed that a minimum BNR of 25 dB is essential to surpass random prediction in terms of precision. For a detection rate of approaching 100%, a BNR higher than 47 dB is recommended. The area under the curve (AUC) approached 1 within this BNR range (Figure R 4e). It should also be noted that the intensity threshold-based segmentation method employed here can be further refined using our DnCNN algorithm, which integrates pre-trained morphological information from actual brain samples.

Revisions:

- (1) The optimization for the bleaching extent was clarified in the Results Section (page 12 lines 9–12):

“Based on the rotation-translation scan strategy and the temporal unmixing method, we performed two optimization steps: (1) for a single sample, we applied multiple scan cycles (typically 8 cycles) to effectively bleach the fluorescent tags, ensuring a sufficient extent of bleaching (Fig. 1c).”

- (2) The details of experimentally controlling the bleaching extent were added in the Methods Section (page 37 lines 9–18):

“During imaging, the laser fluence at the sample surface was 3.2 mJ/cm² at 690 nm for iRFP713, 3.4 mJ/cm² at 697 nm for SNIFP, 2.0 mJ/cm² at 532 nm for mScarlet, and 3.4 mJ/cm² at 800 nm for cross-modal registration. These fluence values were empirically determined to ensure both sufficient bleaching extent of fluorescent proteins (favor high fluence, Supplementary Fig. 10) and high image quality of the bleached features (favor low fluence, Supplementary Fig. 7). To conveniently adjust the light spot size on the sample, a customized expander comprising three lenses (GCL-010112A, GCL-010166A, and GCL-010329, Daheng Optics) was used to expand the beam. ”

- (3) Supplementary Fig. 7 has been revised to include the quantification results of the impact on image quality induced by jump scan.

(4) Figure R4 was added to the manuscript as Supplementary Fig. 10 and was referred in the Results Section (page 13 lines 8–15):

“We compared the histograms of the PATTERN signals before and after confocal-imaging-guided corrections, which shows that PATTERN is accurate (Fig. 2f). Additional experiments indicated that for best performance, a “bleaching extent to noise ratio” (BNR) (see Methods) exceeding 47 dB was required, whereas the minimum BNR for the PATTERN approach was approximately 25 dB (Supplementary Fig. 10). The weak PATTERN signal slightly above the noise floor exhibits some false positives because of artefacts in the raw PA images⁴².”

(5) Details of the above experiment were added to the Methods Section (pages 43-44).

Due to the extensive nature of the revisions, please refer to the manuscript for details.

RC1.3:

- Exponential fitting for PATTERN – was this the same for all biological models (different length scales) or was this done separately for each animal model?

AR1.3:

The bleaching rate exhibited variation not only among different samples but also within different regions of a single sample, as discussed in response to the previous query (AR1.1, Figure R 1). Accordingly, during the exponential fitting process, our methodology incorporates the automatic fitting of both amplitude and bleaching rates (Figure R 5; Supplementary Fig. 5d). This approach ensures adaptability to the diverse samples studied in the current work.

Figure R 5 | Unmixing process for PATTERN. (Copied from Supplementary Fig. 5d)

Revisions:

- (1) The calculation of bleaching rates is highlighted in the Methods section (page 39 lines 11–27):

“...In our algorithm, we fit the PA amplitude decay curve using

$$A(t) = a \cdot \exp(-bt) + c,$$

where the parameter b represents the bleaching rate, a is the PA signal strength of the fluorescent tag which contributes to the useful signals, referred to as “PATTERN signals” in the current work, and c is the PA signal strength of any unbleached background chromophores, referred to as “PA signals” in the current work. To optimize computational speed, the process of curve fitting is divided into the following steps: Initially, an exhaustive search for the bleaching rate b was conducted in parallel using GPU. The searching interval, $0 \leq b \leq 0.09$ per translational scan, was confirmed to fully cover the bleaching rates of different brain samples. A set of bleaching rates, typically comprising 12 rates to achieve a balance between accuracy and calculation speed, was employed to generate an equal number of corresponding bleaching curves. Subsequently, the correlation coefficient between the experimental data and each preset curve followed by its absolute value was calculated to estimate the confidence of the corresponding bleaching rate.”

- (2) Supplementary Fig. 5d has been revised for clarity.

RC1.4:

- PATTERN is not going to provide cell-level resolution, and that may be ok as this approach is complimentary to other destructive techniques. Authors speculate that increasing the center frequency and bandwidth of the transducers may improve resolution. This while true, may not always be good approach for PAT, as low frequencies are equally important for accurate representation of features and shapes. Really, bandwidth is the most important factor and then perhaps PA SNR.

AR1.4:

We appreciate the reviewer's insights and concur with the assessment. We cited a paper which featured an optically interrogated ultrasound transducer with a cutoff frequency of 22 MHz, demonstrated wideband ultrasound detection across a frequency range from DC to 22 MHz². This underscores the reviewer's emphasis on the significance of low frequencies in such measurements.

Revisions:

Bandwidths has been emphasized in the Discussion section (page 26 lines 20–28):

“Additionally, several measures can be implemented to enhance PATTERN’s performance and impact. Firstly, faster lasers or potentially, some deep-learning-based image fusion approaches⁵⁸ can be employed to accelerate the imaging process, while the imaging resolution can be improved by using ultrasound transducers with larger bandwidths. Specifically, the utilization of transducers with a frequency response ranging from direct current (DC) to 22 MHz has demonstrated the ability to achieve a resolution of approximately 50 μm ⁵⁷, which is comparable to the resolution achieved in all-optical, large-FOV brain imaging⁵².”

RC1.5:

- Please provide a table that compares the acquisition speed, resolution, max depth, and FOV for PATTERN, fMOST, and LSFM (also destructive vs non-destructive, fixed vs fresh) for each biological model for comparison. This might be a nice addition for readers to see the pros and cons at glance.

AR1.5:

We appreciate the reviewer's suggestion to provide a comparison table to help readers assess the advantages and limitations of each technology and our system. As suggested, we have conducted a comparison of our system, PATTERN, with the mentioned technologies as a table (Table R1) to provide an overview of the acquisition speed, resolution, maximum depth, field of view (FOV), as well as other relevant parameters. The table is also included in the revised manuscript as Supplementary Table 1.

Table R1 Performance comparison of different methods of whole-brain optical imaging

Table R1: Performance comparison of different methods of whole-brain optical imaging

Different Methods		Sample Preparation			Imaging		Data Analyzing	
Type	Typical method	Sample preparation method (Tissue Clearing)	Whether to fix the sample	Sample preparation time	Typical voxel size	FOV	Data collection time (single)	Reconstruction time
PATTERN	PATTERN (Our system)	None (non-destructive)	Both fixed and fresh samples are available	0	75 $\mu\text{m} \times 75 \mu\text{m} \times 75 \mu\text{m}$	2 cm \times 2 cm	20.5 mins per mouse brain	8 h
Sectioning Tomography	STP ⁷	Agarose-embedding, vibrating sectioning	Fixed	10-14 days	0.5 $\mu\text{m} \times 0.5 \mu\text{m} \times 50 \mu\text{m}$	1.66 mm \times 1.66 mm	24 h per mouse brain	(Not mention)
	FAST ⁸	Agarose-Embedding, vibrating sectioning	Fixed	12-25 h	0.7 $\mu\text{m} \times 0.7 \mu\text{m} \times 5 \mu\text{m}$	1.43 mm \times 1.43 mm	2.4-10 h per mouse brain	16-30 h
	fMOST ⁹	Resin- embedded sectioning	Fixed	5 days	0.32 $\mu\text{m} \times 0.32 \mu\text{m} \times 1 \mu\text{m}$	0.65 mm \times 0.65 mm	11h per mouse brain	3-5 days
Tissue Clearing + Light-sheet microscopy	Different clearing methods		Fixed	5 days				
	combined with Zeiss LightsheetZ.1 (5 \times detection lens, zoom 0.36 \times) (Our data)	uDISCO ³	Fixed	5 days				
		CUBIC ⁴	Fixed	7-21 days	2.5 $\mu\text{m} \times 2.5 \mu\text{m} \times 6.5 \mu\text{m}$	3.4 mm \times 3.4 mm	1.5 h	1 days
		PAGESOS ⁵	Fixed	7 days				
		CLARITY ⁶	Fixed	7 days				

Revisions:

Table R1 has been added to the revised manuscript as supplementary table 1 and referred in the Discussion section (page 28 lines 3–12):

“Due to the limitations of optical microscopy imaging depth, serial sectioning and tissue clearing methods are used to achieve complete whole-brain imaging with high spatial resolution (Supplementary Table 1)^{2,3,6,9}. However, both strategies require a sample preparation process that involves morphological and biological changes, significantly influencing accurate geometric measurement, precise fluorescence signal analysis, and automatic registration of brain samples.”

RC1.6:

- Please clearly articulate why this approach for acquiring PAT data is better than other implementations that use for instance an array of sensors arranged in a hemispherical bowl.

AR1.6:

Indeed, we abstained from asserting the superiority of the current system over other PAT schemes. Specifically, we favored an array with point-like transducers (page 26 lines 7–8, “...a system with point-like ultrasound transducers is preferred for the best imaging quality (isotropic resolution) ...”). However, we maintain that the imaging pipeline we describe is adaptable to various PAT platforms. The choice of the current system was made after comparing the available systems in our laboratory. It's worth noting that, although we have access to a commercial hemispherical array, budgetary constraints prevented us from customizing one ourselves.

The experimentally compared systems included a full ring array system¹⁰, a hemisphere array system (the Endra NEXUS 128 system), and an optical-fiber-sensor-based PA mesoscope¹¹. Below we summarize what we found in the tests.

- (a) The homemade full ring array system, as reported elsewhere¹⁰, exhibits limited resolution in the elevational direction, resulting in unsatisfactory imaging outcomes (Figure R 6a, b).
- (b) The NEXUS 128 system employs a hemispherical sensor array but demonstrates lower sensitivity compared to the PATTERN system. We illuminated the sample using a 700 nm laser and performed the angular scan in 120 steps with 30 averages per step, the entire scanning process took 3.9 minutes, in comparison to a scan time of 2.2 minutes for PATTERN. However, the brain sample remained almost undetectable. A horse tail mane was inserted into the sample for localization purposes (Figure R 6c, d). It is important to acknowledge that the NEXUS 128 system was acquired in 2017 and has undergone extensive use. Consequently, its current performance may not accurately reflect that of a typical hemispherical array system. Unfortunately, at present,

we lack the necessary funding to customize a hemispherical system for a more comprehensive comparison with the half-ring array system employed in the study.

- (c) The optical-fiber-sensor-based PA mesoscope, equipped with a custom-made ultrasound sensor at the fiber tip and a detection bandwidth ranging from DC to 23 MHz¹¹, could potentially achieve a spatial resolution of approximately 50 μm . This sensor underwent raster scanning, effectively representing a planar array. However, the signal-to-noise ratio appeared insufficient for imaging the perfused brain (Figure R 6e, f).

- (d) Figure R 6 | Brain imaging using other systems.** **a, b**, Imaging results using a full ring array system. **c**, Photos of the brain sample imaged by the NEXUS-128 system. **d**, Imaging result of the NEXUS-128 system. Dashed white circle delineates the position of the brain sample. **e**, Brain sample inside the fiber sensor-based photoacoustic mesoscope. **f**, Imaging results of the fiber-optic mesoscope, the profile of the brain was delineated with white dashed lines.

The imaging results clearly demonstrate that the alternatives were inadequate for the application in this study.

In principle, regardless of the array type, if the channel number is fixed, the required number of scanning steps will be approximately the same, whether employing a half-ring or hemispherical arrangement. The hemispherical array entails a slightly lower number of scanning steps due to the relatively more uniform filling of the k-space. In contrast, the half-ring array densely samples the center of k-space, necessitating the application of a filter to homogenize the transfer function.

However, the hemispherical arrangement may introduce additional complications, such as reduced sensor size, leading to larger electrical noise. Consequently, this increase in noise levels may necessitate more averaging to achieve satisfactory results. In contrast, the approach we reported benefits from a broader dynamic range. The elevationally focused design inherently enjoys higher signal amplitude, thereby enabling the detection of weaker signals that might fall below the quantization noise level of an unfocused design, such as a hemispherical arrangement. In such scenarios, our design exhibits superiority, as signals below the quantization noise threshold cannot be effectively recovered through mere averaging.

We recognize the existence of higher-performance systems based on hemispherical arrangements, such as the configuration with 1024 channels at a 40 MHz sampling rate and a 12-bit dynamic range¹², or 3072 channels at the same sampling frequency and dynamic range¹³. These systems potentially offer alternatives for the pipeline of PATTERN. We emphasize that the primary innovation of this paper lies in introducing the concept of bleaching-based temporal encoding and demonstrating its advantages in imaging genetically encoded fluorescent tags, rather than focusing on the design of a specific ultrasound array.

Revisions:

- (1) The imaging results obtained using other PACT systems are added to the manuscript as Supplementary Fig. 19 and referred in the Discussion section (page 26 lines 8–16):

“...However, the reported PA systems of this kind, to our knowledge, are not suitable for the tasks reported in the current work. Difficulties include low resolution or sensitivity to clearly distinguish brain regions, limited FOV for large brains, and excessive laser exposure for fluorescent proteins (Supplementary Fig. 19). Such challenges arise because manufacturing sensitive, broadband and small-footprint ultrasound transducers are difficult. In this study, we employed a relatively simple and cost-effective imaging configuration to achieve both good image quality and high sensitivity to molecular probes.”

- (2) The relationship between the PATTERN pipeline and the imaging platform was explained in the Discussion section (page 26 line 24–page 27 line 2):

“Specifically, the utilization of transducers with a frequency response ranging from direct current (DC) to 22 MHz has demonstrated the ability to achieve a resolution of

approximately $50 \mu\text{m}^{57}$, which is comparable to the resolution achieved in all-optical, large-FOV brain imaging⁵². Secondly, regardless of the detailed implementations, the PATTERN concept is adaptable to PAT platforms involving various scan strategies and could potentially enhance their sensitivity to fluorescent proteins.”

RC1.7:

- Multi-angle filter appears to be implemented in K-space as part of the image reconstruction workflow, but this is not clearly explained. This seems to be an important component. Please provide a detailed explanation on how this is implemented.

AR1.7:

In response to the reviewer's request, the details regarding the k-space filter are provided below. The filter was designed to normalize the uneven sampling density in k-space. Specifically, at each scanning angle, a thin disk oriented along that angle is accessed within k-space. After completing a full scanning cycle, the disk undergoes a full rotation. When these disks are superimposed, the central portion is weighted more heavily. As a result, we attenuated the amplitude of each frequency component by its overall weight, while preserving the original phase. To prevent the introduction of extraneous noise, amplitudes beyond the frequency support of the filter were set to zero. In essence, the filter $H(\theta)$ bears resemblance to a three-dimensional Ram-Lak Filter:

$$H(\theta) = S(\theta) \cdot \frac{1}{\sum_{\theta} S(\theta)},$$

in which $S(\theta)$ represents a transfer function corresponding to a translationally scanned tomogram at angle θ :

$$S(\theta) = \text{step}(k_x \cos(\theta) - k_y \sin(\theta) + W_e) - \text{step}(k_x \cos(\theta) - k_y \sin(\theta) - W_e).$$

W_e is the cut-off spatial frequency in the elevational direction, and $\text{step}(\cdot)$ denotes the step function.

The designed multi-angle filter was employed to reconstruct a single-angle image through the following steps: first, transforming the image to k-space using fast Fourier transform (FFT), then multiplying its spectrum by the filter, and finally transforming it back to the spatial domain using inverse FFT.

Revisions:

The detail of the filter was added in the Methods section (page 38 lines 10–28):

“To improve the elevational resolution, a multiangle fusion procedure was applied consisting of the following steps: (1) Rotating and registering 3D images of 32 different angles into the same coordinates. (2) Transforming the images into the frequency domain,

followed by filtering and summing. (3) Inversely transforming the uniformly-filled spatial frequency signal back into the spatial domain, resulting in a near isotropic 3D volume (Supplementary Fig. 2). The filter in (2) was designed to normalize the uneven sampling density in k-space. Specifically, at each scanning angle, a thin disk oriented along that angle is accessed within k-space. After completing a full scanning cycle, the disk undergoes a full rotation. When these disks are superimposed, the central portion is weighted more heavily. As a result, we attenuated the amplitude of each frequency component by its overall weight, while preserving the original phase. To prevent the introduction of extraneous noise, amplitudes beyond the frequency support of the filter were set to zero:

$$H(\theta) = S(\theta) \cdot \frac{1}{\sum_{\theta} S(\theta)},$$

where $S(\theta)$ represents a transfer function corresponding to a translationally-scanned tomogram at angle θ :

$$S(\theta) = \text{step}(k_x \cos(\theta) - k_y \sin(\theta) + W_e) - \text{step}(k_x \cos(\theta) - k_y \sin(\theta) - W_e).$$

W_e is the cutoff spatial frequency in the elevational direction, and $\text{step}(\cdot)$ denotes the step function.”

RC1.8:

- How does the current approach compare to filtered back-projection reconstruction in process and results?

AR1.8:

We tested the filtered back projection (FBP) reconstruction as the reviewer suggested and the outcome seemed to be worse (Figure R 7a). We found that the transducer's response in the elevational direction, which diverges from the physical model of the FBP method, requires consideration of the spatial impulse response (SIR). However, implementing SIR correction within the FBP framework poses technical challenges, with only a few methods documented¹⁴⁻¹⁶. We adopted a published method known as the "focal-line" technology¹⁵ for the correction, serving as an example to illustrate the modified FBP reconstruction results (Figure R 7b). Our proposed approach is superior due to the incorporation of a modified back-projection filter, which goes beyond merely correcting the projection line (in this case, the projection surface) as the modified FBP method does. For illustrative purposes, a comparison with the results of our approach is presented (Figure R 7c). In this respect, our approach was an enhanced FBP designed for the translation-rotational scanning strategy.

Figure R 7 | Comparison with filtered back-projection reconstruction.

Revisions:

A comparison with the FBP approach was added to Supplementary Fig. 20 and referred in the Discussion section (page 26 lines 12–19):

“Challenges arise due to the difficulty in manufacturing sensitive, broadband, and small-footprint ultrasound transducers. In this study, we employed a relatively simple and cost-effective imaging configuration to achieve both good image quality and high sensitivity to molecular probes. The filtered multi-angle reconstruction procedure, designed specifically for the translation-rotational scanning strategy (Supplementary Fig. 20), can be better understood by visualizing it in the spatial frequency domain, which allows for the optimization of scanning parameters³⁸.”

RC1.9:

- Why use delay-and-sum for the initial reconstruction? This is known to produce low-quality images. Better performing methods might allow 3D volumes with fewer angles and faster acquisition times?

AR1.9:

While delay-and-sum (DAS) may generate suboptimal images under specific conditions, its performance is notably influenced by the system configuration. In the case of most ring or half-ring array systems, DAS and FBP produce comparable reconstruction results due to two factors: 1) The linear dimension of the field of view is relatively small in comparison to the distance from the sensing element, minimizing the impact of the solid angle factor; 2) The piezoelectric (PZT) transducers together with the interfacing electronics exhibit a Gaussian-shaped frequency response, effectively implementing a ramp filter on the PA waveforms in the low-frequency range; while the ramp filter equivalently applies the differentiation operation required for FBP. In addition, image reconstruction and acquisition times are equally crucial for high-throughput imaging. Conventional model-based (iterative)

and time-reversal-based (finite-element analysis) methods are excessively slow. While the FBP reconstruction combined with deconvolution is fast, it relies on precise measurement of the electrical impulse response and inadvertently increases the noise level¹⁷, thereby impacting the sensitivity of the PATTERN signal. Consequently, we chose to employ the DAS method.

Figure R 8 | Maximum allowed rotation step, determining the minimum number of angles (copied from Supplementary Fig. 2d).

For the last question, as we depict in Supplementary Fig. 2d (Figure R 8), the minimum number of scanning angles is independent of the initial reconstruction technique employed. However, this reviewer's comment has prompted an alternative solution for accelerating image acquisition: harnessing fully sampled imaging results as the 'ground truth' to train a deep neural network. This approach aims to generate images with exceptional quality even when the number of scanning angles is significantly reduced. Exploration of such advanced approaches will be a focus of future research.

Revisions:

(1) The reason for applying DAS has been added in the Methods section (page 38 lines 4–8):

“The recorded data underwent reconstruction using the delay and sum (DAS) algorithm, resulting in the production of 2D PA image stacks. DAS was used because of the fast reconstruction speed without sacrificing image SNR for high throughput imaging. Subsequently, these stacks were interpolated using an FFT method to generate 3D images with relatively poor elevational resolution...”

(2) The idea of applying neural networks to accelerate image acquisition was added to the Discussion section (page 26 lines 20–24):

“Additionally, several measures can be implemented to enhance PATTERN’s performance and impact. Firstly, faster lasers or potentially, some deep-learning-based image fusion approaches⁵⁸ can be employed to accelerate the imaging process, while the imaging resolution can be improved by using ultrasound transducers with larger bandwidths....”

RC1.10:

- Fig. 2E, please change the x-axis to units of time.

AR1.10:

Thanks for the suggestion. We agree that the arbitrary unit could potentially confuse readers. Consequently, we have revised the unit to minutes. We'd like to note that the first temporal sampling point was regarded as the end of the first scan cycle (1024 pulse / 10 Hz \approx 1.7 min), providing readers with a more accurate understanding of the actual bleaching time.

Figure R 9 | Temporal profile of the signal from an image voxel revealing the simultaneous presence of a diminishing PATTERN signal and a constant PA background signal (from Fig. 2e).

Revisions:

The unit of the x-axis was changed to minutes in the revised Figure 2.

RC1.11:

- P11: “Compared to using intensity alone...a gain of 12 was achieved in sensitivity (Fig. 2f)” this is not clear, nor do I see how the figure shows a factor of 12 improvement in sensitivity. Please clarify.

AR1.11:

We apologize for the confusion and we have modified the manuscript and the notions of the figure for improved clarity. If the PA signals from the contrast agent are strong, applying a threshold to distinguish these signals from the background is a common practice, as documented in references^{18,19}. In this context, the appropriate threshold should be set at the maximum value of the intrinsic background PA signal, as indicated by the dark red dashed line of Fig. 2f, failure to do so may result in false positives in the detection of the fluorescent tags, leading to erroneous identification of background tissue components as fluorescent proteins. In contrast, PATTERN not only relies on signal amplitude but also

computes the bleaching rates. This approach effectively suppresses background signals whose bleaching rates approach zero. Consequently, this procedure enhances the success rate of fluorescent-tag detection, as evidenced by the very low noise floor depicted by the green dashed line in Fig. 2f. This baseline was established as three times the standard deviation (STD) of the noise in the PATTERN signals. This noise is commonly assumed to adhere to a Gaussian distribution, and a threshold of three times its STD effectively eliminates 99.7% of the noise. We ensured that the small region chosen to compute the STD of the noise contained no signal by cross-referencing it with the confocal imaging results, considered as the ground truth. After normalization, this noise floor has an amplitude of unity. This stands in stark contrast to traditional methods, where the signal detection threshold is twelve, as illustrated by the red dashed line. Consequently, we assert a twelvefold increase in sensitivity through the application of the PATTERN method.

Figure R 10 | Updates in Fig. 2f: Texts in this figure were modified: “maximum of background signal” was modified to “Background roof”, and the baseline defined as three times the noise standard deviation was labelled as “PATTERN noise floor”.

Revisions:

(1) Labels in Fig. 2f were modified and the legend was revised (page 10 lines 10–14):

“f, Histograms of both PA signals (gray) and PATTERN signals (orange) in Fig. 1g; the corrected PATTERN signals guided by confocal are marked individually in green. Green dashed line, the noise floor estimated as three times the standard deviation of PATTERN signals’ noise. Red dashed line, the roof of PA signals (maximum of background).”

(2) The description was modified for improved clarity (page 11 line 14–page 12 line 7):

“Compared to using intensity alone without employing temporal encoding, PATTERN’s detection sensitivity of fluorescent proteins is significantly enhanced (Fig. 2f). To clarify how

we quantified this sensitivity improvement, we examine a thin slice (Fig. 1g). The conventional approach for distinguishing fluorescent labels from intrinsic background involves applying a threshold set at the maximum value of the intrinsic background signal. In Fig. 2f, this threshold is referred to as the "background roof" and is depicted as the red dashed line. Signals exceeding this threshold are identified as originating from fluorescent tags, while signals with amplitudes below the threshold are considered background. It is evident that this thresholding method is highly insensitive, resulting in the discarding of a substantial portion of the actual signal. In contrast, the novel temporal decoding method implemented by PATTERN ensures the effective removal of the unbleachable background. The background-rejection threshold can be set to three times the noise standard deviation (STD), denoted as the "PATTERN noise floor" and illustrated by the green dashed line in Fig. 2f. This approach significantly improves sensitivity by preserving a larger portion of the real signal. We ensured that the small region chosen to compute the STD of the noise contained no signal by cross-referencing it with the confocal imaging results. By comparing the two thresholds, a gain of 12 was achieved, indicating the enhancement in detection sensitivity (Fig. 2f). Such enhancement was validated across the three different types of fluorescent proteins (Supplementary Fig. 6)."

RC1.12:

• P33: "...aCSF was used to minimize biological activity loss" – what activity is being referred to here, does this mean that the whole excised brain remains viable in aCSF?

AR1.12:

We appreciate the reviewer's question and have revised the term 'biological activity loss' as 'biological changes at the molecular level within brain tissue'. In this part of the manuscript, our primary aim was to utilize aCSF treatment to mitigate potential damage to mRNA and protein, facilitating subsequent multi-omics analysis. In our initial submission, we conducted Western Blot and RNA quality assessments (Supplementary Fig. 13f, g) to demonstrate that PATTERN imaging did not induce significant changes in protein or RNA levels. In addition, based on previous studies of ex vivo whole-brain functional imaging and molecular analysis^{20,21}, neuronal activity in acutely excised brains can also be preserved following a 20-minute imaging process.

Revisions:

The corresponding description in the Methods section was revised (page 37 line 24–page 38 line 2):

"In addition, a perfusion module was utilized to maintain a liquid environment consistent with body fluid. For the PA imaging procedure followed by other imaging modalities, the imaging chamber was perfused with PBS; for imaging followed by other biological

measurements, aCSF was used to minimize biological changes at the molecular level of the brain tissue²⁷ (Supplementary Fig. 18f-h).”

RC1.13:

- For DnCNN, please provide pertinent details on number of training and testing data, number of hyperparameters, what strategies were used for data augmentation, etc.

AR1.13:

As suggested, we have included more detailed information of the DnCNN method in the revised manuscript (located on page 46, line 9). Take the hippocampus as an example, our training process involved a dataset comprising 156 pairs of images, each with and without artefacts, while testing was conducted on a separate set of 60 pairs. Noteworthy hyperparameters employed during training included a batch size of 1, 50 training epochs, and a learning rate set at 1e-3. At the 30th epoch, we implemented a learning rate reduction by a factor of 10. We utilized traditional approaches of data augmentation techniques including flipping, rotating, and intensity adjustments. All codes and the pre-trained network utilized in this study are available at <https://github.com/CaA2318777/PATTERN>.

Revisions:

Detailed information of the DnCNN method has been included in the revised Methods section, (page 46 lines 9–20).

“The neural network implicitly learned the artefacts of each image to achieve the distribution of the artefacts across the entire dataset. The Mean Squared Error (MSE) loss function was employed as the training criterion over 50 epochs, during which the network effectively filtered out pure artefacts to generate the output. For each brain region, a dataset consisting of more than 100 pairs of images with and without artefacts were used for training and more than 50 pairs for testing. Some of the representative hyperparameters used during training are provided below: batch size = 1, training epochs = 50, and learning rate = 1e-3. When the epoch reached 30, the learning rate was reduced by a factor of 10. To make the network more generalized, traditional data augmentation strategies were also applied, including flipping, rotating, and intensity changes.”

Reference:

- 1 Michels, R., Foschum, F. & Kienle, A. Optical properties of fat emulsions. *Optics Express* **16**, 5907-5925 (2008). <https://doi.org:10.1364/OE.16.005907>
- 2 Jathoul, A. P. *et al.* Deep in vivo photoacoustic imaging of mammalian tissues using a tyrosinase-based genetic reporter. *Nature Photonics* **9**, 239 (2015).
- 3 Pan, C. *et al.* Shrinkage-mediated imaging of entire organs and organisms using uDISCO. *Nature Methods* **13**, 859-867 (2016). <https://doi.org:10.1038/nmeth.3964>

- 4 Matsumoto, K. *et al.* Advanced CUBIC tissue clearing for whole-organ cell profiling. *Nature Protocols* **14**, 3506-3537 (2019). <https://doi.org:10.1038/s41596-019-0240-9>
- 5 Jing, D. *et al.* Tissue clearing of both hard and soft tissue organs with the PEGASOS method. *Cell Research* **28** (2018). <https://doi.org:10.1038/s41422-018-0049-z>
- 6 Tomer, R., Ye, L., Hsueh, B. & Deisseroth, K. Advanced CLARITY for rapid and high-resolution imaging of intact tissues. *Nature Protocols* **9**, 1682-1697 (2014). <https://doi.org:10.1038/nprot.2014.123>
- 7 Ragan, T. *et al.* Serial two-photon tomography for automated ex vivo mouse brain imaging. *Nature methods* **9**, 255-258 (2012). <https://doi.org:10.1038/nmeth.1854>
- 8 Seiriki, K. *et al.* High-Speed and Scalable Whole-Brain Imaging in Rodents and Primates. *Neuron* **94**, 1085-1100.e1086 (2017). <https://doi.org:10.1016/j.neuron.2017.05.017>
- 9 Zhong, Q. *et al.* High-definition imaging using line-illumination modulation microscopy. *Nature Methods* **18**, 309-315 (2021). <https://doi.org:10.1038/s41592-021-01074-x>
- 10 Cui, M. *et al.* Adaptive photoacoustic computed tomography. *Photoacoustics* **21**, 100223 (2021).
- 11 Chen, Y. *et al.* Photoacoustic Mouse Brain Imaging Using an Optical Fabry-Pérot Interferometric Ultrasound Sensor. *Frontiers in neuroscience* **15**, 572 (2021).
- 12 Lin, L. *et al.* High-speed three-dimensional photoacoustic computed tomography for preclinical research and clinical translation. *Nature communications* **12**, 1-10 (2021).
- 13 Cao, R. *et al.* Single-shot 3D photoacoustic computed tomography with a densely packed array for transcranial functional imaging. arXiv:2306.14471 (2023). <<https://ui.adsabs.harvard.edu/abs/2023arXiv230614471C>>.
- 14 Lu, T. *et al.* Full-frequency correction of spatial impulse response in back-projection scheme using space-variant filtering for photoacoustic mesoscopy. *Photoacoustics* **19**, 100193 (2020). <https://doi.org:https://doi.org/10.1016/j.pacs.2020.100193>
- 15 Xia, J. *et al.* Three-dimensional photoacoustic tomography based on the focal-line concept. *Journal of Biomedical Optics* **16**, 090505-090505-090503 (2011). <https://doi.org:10.1117/1.3625576>
- 16 Seeger, M. *et al.* Pushing the boundaries of photoacoustic microscopy by total impulse response characterization. *Nature Communications* **11**, 2910 (2020). <https://doi.org:10.1038/s41467-020-16565-2>
- 17 Van de Sompel, D., Sasportas, L. S., Jokerst, J. V. & Gambhir, S. S. Comparison of Deconvolution Filters for Photoacoustic Tomography. *PLoS ONE* **11** (2016).
- 18 Zhang, J. *et al.* In vivo characterization and analysis of glioblastoma at different stages using multiscale photoacoustic molecular imaging. *Photoacoustics* **30**, 100462 (2023).
- 19 Yao, J. *et al.* Multiscale photoacoustic tomography using reversibly switchable bacterial phytochrome as a near-infrared photochromic probe. *Nature Methods* **13**, 67-73 (2016). <https://doi.org:10.1038/nmeth.3656>

- 20 Margrie, T. W. *et al.* Targeted Whole-Cell Recordings in the Mammalian Brain In Vivo. *Neuron* **39**, 911-918 (2003).
[https://doi.org:https://doi.org/10.1016/j.neuron.2003.08.012](https://doi.org/https://doi.org/10.1016/j.neuron.2003.08.012)
- 21 Gottschalk, S. *et al.* Rapid volumetric optoacoustic imaging of neural dynamics across the mouse brain. *Nature biomedical engineering* **3**, 392-401 (2019).

Reviewer #2 :

(RC: Reviewer's Comment, AR: Author's Response)

RC2.0:

I am highly impressed by the innovative work of Dr. Ma's team in developing the PATTERN photoacoustic imaging technology. Their manuscript presents a significant advancement in the field, demonstrating the capability of PATTERN for high-resolution, non-destructive, and high-speed imaging of fluorescent proteins in fresh ex vivo brains. The paper is well-written and effectively communicates the technical details and results. I offer several major and minor comments that I believe will further enhance the quality and impact of this excellent study.

AR2.0:

Thank you for your invaluable feedback. We deeply appreciate your thoughtful comments, and we carefully considered each of them. Accordingly, we have made corrections and supplemented new experimental data. Your insights have been instrumental in refining our work, and we are grateful for the opportunity to improve based on your guidance.

RC2.1:

Major Comments:

1. Sensitivity Limit of PATTERN:

The authors successfully highlight the substantial enhancement in detection sensitivity achieved through the photobleaching approach in PATTERN. To gain a deeper understanding of PATTERN's performance, I suggest exploring the sensitivity enhancement across different fluorescent proteins.

AR2.1:

Thank you for this valuable suggestion, we conducted a series of phantom experiments to evaluate various fluorescent proteins. We utilized a v/v 3.6% intralipid-30% suspension, to simulate brain tissue. Within this suspension, a PTFE tube with an inner diameter of 0.3 mm, containing purified proteins, was placed. We tested three types of fluorescent proteins: iRFP713, mScarlet, and SNIFP, excited at wavelengths of 690 nm, 532 nm, and 697 nm, respectively (Figure R 11a, c, e). Each protein had a molar concentration of 30 $\mu\text{mol/L}$. For each depth assessed, we manually selected the pixels within the tube as the region of interest (ROI) for calculating the Signal-to-Noise Ratio (SNR) for both the initial PA images and the PATTERN images (derived from temporal encoding, refer to manuscript page 12 lines 26–28).

Before delving into the calculation of the SNR, we first provide an explanation of how the signal was defined for the data acquired in PATTERN. The photobleaching process contributes to an exponentially decaying signal, which can be characterized as follows:

$$A(t) = a \cdot \exp(-bt) + c,$$

where $A(t)$ is the PA amplitude and the parameter b represents the bleaching rate, a is the PA signal strength of the fluorescent tag, referred to as “PATTERN signals”. It’s worth noting that if the calculated bleaching rate b was very small, the corresponding PATTERN signal a was directly set to zero to avoid any unstable inverse operation.

Figure R 11 | Sensitivity enhancement of different fluorescent proteins. a, Imaging results of iRFP713. **b,** SNR enhancement by PATTERN for iRFP713. **c,** Imaging results of mScarlet. **d,** SNR enhancement by PATTERN for mScarlet. **e,** Imaging results of SNIFP. **f,** SNR enhancement by PATTERN for SNIFP.

For both the traditional method and the PATTERN method, we characterized the sensitivity using SNR. Signals outside the ROI were confirmed to be generated from the system noise, and inside was the superposition of protein signals and system noise. We computed SNR according to the following equation:

$$SNR = \frac{P_{in ROI} - P_{out ROI}}{P_{out ROI}}, \quad (R2.1)$$

where P is the power of the corresponding signal, calculated by:

$$P_{region} = \frac{\sum_{region} pixel\ value^2}{pixel\ number}. \quad (R2.2)$$

According to the experimental results, iRFP713 was the brightest, followed by SNIFP, with mScarlet being the least bright. Notably, the SNR varied across different ROIs, so a direct comparison was made between a pair of SNRs—one from the initial PA image and the other from the PATTERN image—pertaining to the same ROI, each represented as a scatter point (Figure R 11b, d, f). The results indicate that the PATTERN method can enhance SNR by at least 2-fold and up to 20-fold. The most significant enhancement was observed with mScarlet, potentially due to its lower initial PA image SNR. This suggests that the PATTERN method's sensitivity enhancement is more pronounced for darker fluorescent proteins in PA imaging.

Figure R 12 | Copied from Fig. 2f. Histograms of both PA signals (gray) and PATTERN signals (orange) in Fig. 1g; the PATTERN signals corrected using the confocal images are marked individually in green. The standard deviation (STD) of the noise was calculated using signals identified as true negatives. Three times the noise STD was then designated as the noise floor, denoted by the green dashed line. The maximum value of the background signal was labeled as the "background roof," identified by the red dashed line.

Regarding the real brain sample, the signals arising from fluorescent proteins and intrinsic tissue background were superimposed, making it challenging to assess their respective amplitudes without a definitive gold standard. Consequently, quantifying the SNR corresponding to the fluorescent proteins proved unfeasible. Instead, for the real brain samples, we characterized the noise-suppressing capability of the PATTERN approach by the threshold value we used for rejecting the intrinsic tissue background, as illustrated in Figure 2f (copied in Figure R 12 below). It's noteworthy that the estimation of SNR enhancement (Figure R 11) was influenced by the detection rate, a factor dependent on the extent of bleaching. A detailed analysis of this aspect is presented in another response (AR2.4) on page 33.

Revisions:

(1) Additional details regarding the calculation of the PATTERN signal have been incorporated into the Methods section (page 39 line 27–page 40 line 7):

“Ultimately, the preset curve associated with the rate with the highest confidence was used to determine both the amplitude of the decreasing part (PATTERN signal) and the amplitude of the constant part (PA signal) through linear unmixing. To guarantee the stability of the calculation, when the bleaching rate was determined to be exceptionally small (e.g., $b = 0.0001$), the PATTERN signal was set to zero. This precaution was taken to prevent the inversion of a low-rank matrix. Physically, setting the PATTERN signal to zero signifies classifying the voxel as unbleachable.”

(2) The data above was added in Supplementary Fig. 6 and referred in the Results section (page 12 lines 4–7):

“By comparing the two thresholds, a gain of 12 was achieved, indicating the enhancement in detection sensitivity (Fig. 2f). Such enhancement was validated across the three different types of fluorescent proteins (Supplementary Fig. 6).”

(3) Fig. 2f was revised for clarity.

(4) The experiment detail was added to the Methods section (pages 43–44).

Due to the extensive nature of the revisions, please refer to the manuscript for details.

RC2.2:

Additionally, investigating its performance in the presence of strong background signals, such as blood, would provide insights into its potential for in vivo applications.

AR2.2:

We have validated the efficacy of the PATTERN approach in suppressing strong background signals, as evidenced in Supplementary Fig. 13, where melanin was detected

in the brain sample. Despite being five times more intense than the fluorescent protein in photoacoustic amplitude (Figure R 13b), melanin did not impede the algorithm's performance. Another example is shown in Figure R 11, where the red arrows highlight the successful rejection of the intense signals from dusts by the PATTERN approach.

Figure R 13 | Brain sample with strong background signals. **a**, Three-dimensional image showing the PATTERN signal in green color and background PA signals in gray scale. (Copied from Supplementary Fig. 13a) **b**, The maxima of the PATTERN signal (orange) and the PA background signal (blue), indicated by the arrows in (a).

PATTERN can resolve the signals of fluorescent proteins even when co-located with strong background signals, as demonstrated in Fig. 2e of the manuscript (copied below).

Figure R 14 | Temporal profile of the signal from an image voxel revealing the simultaneous presence of a diminishing PATTERN signal and a constant PA background signal (copied from Fig. 2e).

The background-suppression capacity of PATTERN is further validated by a phantom experiment, in which a tube was filled with 30 $\mu\text{mol/L}$ purified iRFP713 and bovine blood in a 1:1 ratio. Another tube filled with 15 $\mu\text{mol/L}$ purified iRFP713 was placed in the same holder for comparison (Figure R 15a). The bleaching curves of the two tubes exhibited similar decay rates and noise levels, suggesting that the noise observed was additive

rather than multiplicative (Figure R 15c). Hence, we have demonstrated that PATTERN is capable of isolating the signal from fluorescent tags even in the presence of a strongly co-located background (Figure R 15 b, d).

Figure R 15 | Performance of PATTERN in the presence of blood. **a**, Photoacoustic image depicting two tubes, one filled with purified iRFP713 (upper left) and the other with a mixture of iRFP713 and bovine blood (lower right). Inset: experimental schematic. **b**, PATTERN image corresponding to (a). **c**, Bleaching curves of the two points indicated by the arrows in (a). **d**, Profiles along the dashed lines in (a) and (b).

Revisions:

The above analysis and figure were added to supplementary Fig. 23 and referred in the Discussion section (page 27 lines 3–7):

“Thirdly, with the bleaching non-linearity of the fluorescent proteins calibrated, their molecular concentrations can potentially be quantified by PATTERN (Methods and Supplementary Figs. 21 and 22). Lastly, the background rejection capability of PATTERN can be further explored for in vivo applications (Fig. 2e and Supplementary Fig. 23).”

RC2.3:

A comparison of PATTERN's sensitivity with that of photoswitching-based techniques, which offer reversible sensitivity enhancement, would strengthen the manuscript.

AR2.3:

Thank you for highlighting this promising alternative tagging strategy. The rapid switching rate and potential for multiple cycles in the time domain translate into a narrower modulation bandwidth in the time-frequency domain. The narrow bandwidth facilitates the suppression of noise, thereby enhancing both sensitivity and specificity in distinguishing between fluctuating objects compared to the methods presented in the current work. Nevertheless, achieving this reversible capability necessitates more stringent experimental conditions, and it may be less compatible with the downstream fluorescence imaging workflow. On the contrary, the bleaching-based method, employing routine labeling strategies, can be applied with nearly all currently available fluorescent proteins.

Our experiments indicate that the switching rate of photoswitchable proteins far exceeds the bleaching rate of conventional proteins. Consequently, our current demodulation scheme lacks the capability to detect these photo-switchable proteins with sufficient quality (Figure R 20). We attempted to implement photoswitching with an alternative demodulation scheme but have not achieved success, and unfortunately, we are currently unable to furnish a comprehensive quantitative comparison between the two strategies.

Revisions:

- (1) The discussion of photoswitching-based techniques was added to the Discussion section (page 27 lines 8–14):

“It has been shown that the use of photoswitchable proteins significantly improves intrinsic tissue background rejection, thus enhancing sensitivity to molecular labels²². However, compared to traditional fluorescent proteins⁴⁰, further development of photoswitchable proteins is required to address the diverse needs in various neuroscience studies. Thus, in PATTERN, we employed the photobleaching process instead of the photoswitching process, to ensure compatibility with routine labeling strategies (Supplementary note 2).”

- (2) The discussion of photoswitching was added as Supplementary note 2:

“The photoswitching-based technique is another promising alternative for fluorescent tagging. The rapid switching rate and potential for multiple cycles in the time domain translate into a narrower modulation bandwidth in the time-frequency domain. The narrow bandwidth facilitates the suppression of noise, thereby enhancing both sensitivity and specificity in distinguishing between fluctuating objects compared to the methods presented in the current work. Nevertheless, achieving this reversible capability relies on changes in the protein conformation of specific types of fluorescent tags. On the contrary, the bleaching-based method, employing routine labeling strategies, can be applied to nearly all currently available fluorescent proteins.”

RC2.4:

Furthermore, an investigation into the extent of photobleaching required to achieve the desired sensitivity, through phantom studies with purified fluorescent proteins in scattering media, would add valuable information.

AR2.4:

We appreciate the reviewer's constructive suggestion and are eager to address this question in a more practical context. The degree of photobleaching is influenced by local optical fluence and the concentration of fluorescent proteins. In our experiments, we modulated photobleaching by varying the light fluence at different positions within the scattering media, calibrating it against the noise floor of our system. For ease of reference, we quantified this as the “bleaching extent to noise ratio” (BNR), defined as:

$$BNR = 20 \lg \left(\frac{\max_t(PA \text{ amplitude}) - \min_t(PA \text{ amplitude})}{std(noise)} \right).$$

The standard deviation was estimated using a manually selected region that did not contain fluorescent proteins.

Figure R 16 | The relationship between bleaching extent to noise ratio (BNR) and PATTERN signal's SNR.

We used a single tube containing purified iRFP713, whose concentration was maintained constant, as the test target. For each position, we computed and plotted the BNR and corresponding SNR (refer to Eqs. R1.1 and R1.2 for definition) of the PATTERN signals (Figure R 16 a). The results indicate that the PATTERN approach yields an SNR exceeding 2 provided the extent of bleaching reaches a minimum of 25 dB, which corresponds to a detection accuracy of above 50%. Above a bleaching extent of 45 dB, the SNR in PATTERN signals ceases to increase, suggesting that the PATTERN approach does not introduce additional noise, thus the SNR was fully decided by the raw PA signals (from

fluorescent proteins). At such high BNR, the detection accuracy reaches almost 100%. This outcome is indicative of an almost ideal differentiation between fluorescent proteins and other background signals. It is also noteworthy that this segmentation efficacy could be further augmented through the use of our DnCNN algorithm, which incorporates pre-trained morphological information from actual brain samples.

Revisions:

- (1) The illustration of the optimal and suboptimal bleaching extent was added in the Results section (page 13 lines 8–15):

“We compared the histograms of the PATTERN signals before and after confocal-imaging-guided corrections, which shows that PATTERN is accurate (Fig. 2f). Additional experiments indicated that for best performance, a “bleaching extent to noise ratio” (BNR) (see Methods) exceeding 47 dB was required, whereas the minimum BNR for the PATTERN approach was approximately 25 dB (Supplementary Fig. 10). The weak PATTERN signal slightly above the noise floor exhibits some false positives because of artefacts in the raw PA images⁴².”

- (2) The details of the data analysis method were added to the Methods section (page 44 lines 4–24).

Due to the extensive nature of the revisions, please refer to the manuscript for details.

RC2.5:

2. Signal Non-linearity of PATTERN:

The authors' creative use of photobleaching to enhance detection sensitivity in PATTERN raises questions about the non-linear relationship between its signals and local optical fluence. Photobleaching was explored before for PAM as a non-linear approach to enhance the spatial resolution (Physical review letters 112, no. 1 (2014): 014302.). It is innovative to turn photobleaching, an otherwise unwanted side effect, into a useful tool for improving detection sensitivity. Because the photoacoustic rate (or the signal deduction speed) usually has a power dependence on the optical fluence (may be larger than 1 for some strong optical absorbers), the PATTERN signals should have a non-linear dependence on the local optical fluence. To address this, I recommend a quantitative investigation into the non-linear behavior of PATTERN signals such as the signal decay with increasing depth, particularly in deep brain targets with reduced optical fluence. Comparing the non-linear behavior of PATTERN signals to traditional PA and fluorescence signals would shed light on its implications for cross-model validation and analysis.

AR2.5:

We are grateful for the reviewer's insightful feedback, which has prompted us to delve deeper into quantitative imaging – by calibrating the non-linearity of bleaching, we have now developed a methodology for signal quantification. However, our proposed method of investigating bleaching non-linearity for image quantification is in its early stages, and comprehensive analyses, including noise immunity and system parameter optimization, are still ongoing. Consequently, we decided to maintain the original PATTERN signal processing method in the paper, while introducing an analysis of PATTERN's non-linearity as supplementary material. Further exploration of this aspect is planned for future research.

Our current analysis of the non-linearity is as follows. According to the paper provided by the reviewer (Physical review letters 112, no. 1 (2014): 014302.), the PA amplitude $P(t)$ (as a function of the scanning time t) can be expressed as

$$P(t) \propto \Gamma \eta_{\text{th}} C_0 I \exp(-kI^\beta t),$$

where Γ is the Grueneisen coefficient, and η_{th} is the percentage of the absorbed photon energy that is converted into heat. C_0 is the concentration of molecules, and I is the excitation intensity. The factor kI^β represents the bleaching rate in which k is a constant factor and β is the intensity power dependence with $\beta \geq 0$. In PATTERN, a 3D image is computed by integrating a number of 2D frames, mathematically:

$$A(t) \propto \int_t^{t+T} P(\tau) d\tau.$$

Here $A(t)$ is the amplitude of the signal acquired by the multiangle fusion process, and T is the scanning time of a single cycle. Collectively, we get:

$$A(t) \propto \frac{\Gamma \eta_{\text{th}} C_0 I}{kI^\beta} (1 - \exp(-kI^\beta T)) \exp(-kI^\beta t).$$

This equation represents the exponential decay of raw photoacoustic signals over the entire scanning procedure (including many cycles), a relationship that has been experimentally validated and applied in our study. Furthermore, the interconnection between amplitude and bleaching, mediated by the excitation intensity, offers a methodology for calibrating the parameter β using the aforementioned data through the subsequent steps:

(1) Fit $A(t)$ of different positions with exponential function $A(t) = a \cdot \exp(-bt)$, obviously we get

$$kI^\beta = b, \tag{R2.3}$$

and

$$\frac{\Gamma \eta_{\text{th}} C_0 I}{kI^\beta} (1 - \exp(-kI^\beta T)) = a. \tag{R2.4}$$

(2) Calculate the equivalent excitation intensity $I_{eq} \propto I$:

Substitute Eq. R2.3 into Eq. R2.4, we can calculate the excitation intensity I using:

$$I = \frac{1}{\Gamma \eta_{th} C_0} \cdot \frac{ab}{1 - \exp(-bT)}$$

Notice that Γ and η_{th} are constants for the same type of protein, and C_0 was kept as a constant for the whole tube, we define the equivalent excitation intensity I_{eq} as follow:

$$I_{eq} = \frac{ab}{1 - \exp(-bT)}$$

Obviously, we get $I \propto I_{eq}$.

(3) Calculate the intensity power dependence factor β using least squares method:

$$\begin{bmatrix} \ln(k) - \beta \ln(\Gamma \eta_{th} C_0) \\ \beta \end{bmatrix} = (\mathbf{X}\mathbf{X}^T)^{-1} \mathbf{X} \ln(\mathbf{b}_{n \times 1}),$$

in which \mathbf{X} is a matrix of 2 by n:

$$\mathbf{X} = \begin{bmatrix} \mathbf{1}_{1 \times n} \\ \ln(I_{eq}) \end{bmatrix}_{2 \times n}.$$

All three proteins were calibrated and the β parameter of iRFP713, mScarlet and SNiFP was measured to be 2.64, 1.8 and 1.63, respectively (Figure R 17).

Figure R 17 | The non-linearity of different fluorescent proteins. a-c, Photobleaching rates of different proteins have different intensity power dependences. (a), iRFP713, (b), mScarlet, (c), SNiFP.

By calibrating the nonlinearity parameter β , we have now developed a methodology for quantifying C_0 , as detailed in the following steps:

- (1) Use PATTERN to get the PATTERN signal amplitude a , bleaching rate b and PA signal (background) of a voxel.
- (2) Calculate the equivalent molecule concentration $C_{eq} \propto C_0$:

Eliminate excitation intensity I in Eq. R2.3 and Eq. R2.4 and extract C_0 :

$$C_0 = \frac{k^{\frac{1}{\beta}}}{\Gamma \eta_{th}} \cdot \frac{ab^{\frac{\beta-1}{\beta}}}{1 - \exp(-bT)}$$

Notice that Γ , η_{th} , k and β are all constants for the same type of protein, we define equivalent molecule concentration C_{eq} as follow:

$$C_{eq} = \frac{ab^{\frac{\beta-1}{\beta}}}{1 - \exp(-bT)}$$

which is related to the real molecule concentration ($C_{eq} \propto C_0$).

Figure R 18 | Comparison of PATTERN reconstruction and quantified PATTERN reconstruction. **a**, PATTERN image of a tube filled with SNIFP. **b**, Molecule concentration image corresponding to (a) reconstructed using the non-linear-based quantification method. **c**, Profiles of the lines in (a) and (b).

An additional experiment was conducted using SNIFP to evaluate the effectiveness of this method (Figure R 18). The advantages and disadvantages of this approach are: on one hand, the quantified PATTERN signals depend on the relationship between the bleaching rates and the initial PA amplitude. Consequently, artefacts or false positive PATTERN signals can be effectively reduced, as their bleaching rates are typically too rapid to align with their amplitudes. On the other hand, this method demands high precision in fitting

since errors or noise are magnified during the additional calculation process. This requirement can hinder accurate quantification (Figure R 18c).

To mitigate noise, the quantified PATTERN approach could be adapted to encompass a 'super voxel', averaging values across several voxels. However, it is crucial to maintain the super voxel size sufficiently small to ensure that the optical fluence is approximately constant within these voxels²². This adjustment, while effective in reducing noise, would result in a compromise on spatial resolution. Additionally, the actual bleaching rates of the sample may vary depending on the microenvironment of the proteins²³, indicating that the parameters k or β might not be constant even for the same type of fluorescent protein. In light of these considerations, we have chosen to retain the original PATTERN signal processing as described in the paper, while incorporating an analysis of PATTERN's non-linearity in the Supplementary Materials.

Revisions:

(1) The method for calibrating the intensity power dependence was added to the Methods section (pages 40–42).

Due to the extensive nature of the revisions, please refer to the manuscript for details.

(2) Figure R 17 and Figure R 18 were added to the manuscript as Supplementary Figs. 21, 22, respectively, and referred in the Discussion section (page 27 lines 3–7).

“Thirdly, with the bleaching non-linearity of the fluorescent proteins calibrated, their molecular concentrations can potentially be quantified by PATTERN (Methods and Supplementary Figs. 21 and 22). Lastly, the background rejection capability of PATTERN can be further explored for in vivo applications (Fig. 2e and Supplementary Fig. 23).”

RC2.6:

Minor Comments:

1. Clarification Needed:

- Page 4, line 10: The term "a large missing cone" should be explained more comprehensively for readers who are not experts in imaging, specifically in relation to spatial frequencies (k space).

AR2.6:

Many PACT systems employ ultrasound transducer arrays that are geometrically focused in the elevational direction, resulting in a large missing cone in this direction, within which all spatial frequency components are lost. The presence of the missing cone in the spatial-frequency domain is a significant contributor to image artifacts and resolution degradation.

Revisions:

The term “missing cone” was explained in the Introduction section (page 4 lines 20–27):

“Many PACT systems employ ultrasound transducer arrays that are geometrically focused in the elevational direction^{19,28–30}, resulting in a large missing cone in this direction, within the missing cone all spatial frequency components are lost. The presence of the missing cone in the spatial-frequency domain is a significant contributor to image artifacts and resolution degradation. As a result, for three-dimensional (3D) imaging, the elevational resolution is usually more than five times larger than the in-plane resolution.”

RC2.7:

- Page 27: Further clarification is required regarding how the translation and rotation effectively address the limited view problem in PATTERN, especially concerning the detection of vertical structures along the acoustic axis.

AR2.7:

The adoption of half-ring array ultrasound transducers has alleviated the limited-view problem within the imaging plane, i.e., the first two dimensions, as the 180° coverage ensures angularly uniform sampling in these dimensions. Introducing translation along the elevational direction added a third dimension; however, the small numerical aperture corresponding to the elevational direction results in limited-view artifacts. To address this problem, rotational scanning employs a multi-view strategy to synthetically create a large acceptance angle in the third dimension, effectively achieving nearly omnidirectional sampling in the spatial-frequency domain and minimizing the impact of the limited-view problem.

Revisions:

(1) Clarification on how the translation and rotation effectively address the limited-view problem was added to Supplementary note 1:

“The adoption of half-ring array ultrasound transducers has alleviated the limited-view problem within the imaging plane, i.e., within the first two dimensions, all spatial frequency components are captured by the 180° coverage of the ultrasound array. Introducing translation along the elevational direction added a third dimension; however, the small numerical aperture corresponding to the elevational direction results in significant limited-view artifacts. To address this problem, the rotational scanning employs a multi-view strategy to synthetically create a large acceptance angle in the third dimension, effectively achieving nearly omnidirectional sampling in the spatial-frequency domain and minimizing the impact of the limited-view problem.”

(2) The added Supplementary note 1 was referred in the Results section (page 7 lines 5–10):

“The original PACT system⁴¹ only scans linearly along the elevational direction in 3D imaging. The restricted acceptance angle of the transducer array leads to diminished resolution along the scanned direction. In PATTERN, we address this issue by rotating the system to mitigate the limited-view problem (Supplementary note 1). The 3D images acquired at different scan angles are combined to reconstruct a 3D volume with near isotropic resolution (Fig. 1c-g).”

RC2.8:

2. Figure and Terminology:

- Page 8, Figure 1a: Consider whether PATTERN should follow the "brain retrieval".

AR2.8:

Thanks for the reminder and we agree that such an expression would cause some confusion for readers. In the initial submission, we wanted to emphasize that the sample (brain tissue) would not be damaged after PATTERN imaging. We have already changed it to "Further analysis" in the revised version of Figure 1a (see Figure R 19).

Figure R 19 | The pipeline for PATTERN imaging (Figure 1a).

Revisions:

Figure 1a has been revised (page 8 line 1).

RC2.9:

- Background Signals: Please provide additional details about the PA contrast of background signals, including their composition (somas) and potential sources (lipids, cell plasma, water).

AR2.9:

The source was physiological structures within the brain, notably in some wild-type mice, which possess pigment cells in their brains (Supplementary Fig. 13). The reviewer may be particularly interested in those common sources that delineate brain structure. A prior study has examined these features, categorizing them as proteins and lipids²⁴. We hypothesize

that these signals predominantly arise from lipids, a conclusion supported by the observation that lipids do not exhibit a fixed photoacoustic spectrum but vary depending on specific components²⁵. This variability aligns with the diverse photoacoustic spectra we observed. Additionally, such background signals of the brain were almost invisible after a degreasing process, further indicating the contributions of lipids (details will be published in a future manuscript). Consequently, we propose that such background signals lack a specific composition, with the somas appearing more prominently.

Revisions:

The potential sources of the background signals were discussed in the Results section (page 12 lines 23–26):

“To prevent any ambiguity, we utilize the term "PA signal/image" in the current work to refer to signals and image features that correspond to the intrinsic tissue background. Such background contrast, depicted in grayscale across all subfigures (except Fig. 2g), is potentially attributed to lipids¹⁹.”

RC2.10:

3. Reconstruction and Processing:

- Positive PA Signals: Elaborate on the specific technique used for image reconstruction (e.g., DAS) and whether the Hilbert transform is applied to eliminate negative signals, along with its potential impact on spatial resolution.

AR2.10:

Our primary reason for utilizing the Delay-and-Sum (DAS) method is its rapid processing speed. Notably, we did not apply a Hilbert transform, which is evident from the presence of negative signals in the photoacoustic amplitude images. While the PATTERN approach inherently disregards negative PA amplitudes, these negative values in the PATTERN signal amplitude are converted into absolute values for the purpose of visualization. Based on experience, most of the PATTERN signals were positive, possibly due to the small size of the neurons contributing primarily high frequency signals. We believe this procedure does not compromise the spatial resolution of the PATTERN images.

Revisions:

Detailed discussion about the reconstruction method was added (page 38 lines 4–8):

“The recorded data underwent reconstruction using the delay and sum (DAS) algorithm, resulting in the production of 2D PA image stacks. DAS was used because of the fast reconstruction speed without sacrificing image SNR for high throughput imaging.

Subsequently, these stacks were interpolated using an FFT method to generate 3D images with relatively poor elevational resolution.”

RC2.11:

4. Jumping Sequence and Quantitative Analysis:

- The innovative jumping sequence for reducing k-space non-uniformity warrants further discussion, particularly regarding its potential implications on the quantitative analysis of PATTERN signals.

AR2.11:

In the original version of our paper, we provided a qualitative analysis, concluding that the implementation of a jumping sequence notably reduced k-space non-uniformity relative to a standard sequential scan, thereby enhancing image quality and minimizing signal distortion. Nevertheless, it's important to acknowledge that this non-uniformity was not entirely eliminated and continued to affect the system's performance. Consequently, the extent of its impact remains somewhat ambiguous. To address this, the updated version of our simulation incorporates the precise parameters and processing codes used in the actual experimental data for a more quantitative assessment.

Initially, we assessed the image degradation resulting from photobleaching by comparing the point source image with the reconstructed image using cosine similarity. For comparison purposes, a commonly used sequential order, $-90^\circ, -84.375^\circ \dots, 0^\circ \dots, 84.375^\circ$, was set as the control group. The imaging quality was the primary consideration. Assuming that a point source (1×1 pixel at the center of a 256×256 grid) was bleached during a scan cycle. We assessed the image degradation resulting from photobleaching by comparing the point source image with the reconstructed image using cosine similarity. As the rate of bleaching increased, it was observed that sequential scan yielded progressively poorer image quality (Figure R 20e), with a corresponding rapid decrease in cosine similarity. Conversely, the jumping scan approach maintained a more stable performance (Figure R 20f).

Figure R 20 | Optimization of rotation order for image quality. **a**, Diagram of sequential scan of a point source which is exponentially bleached during the scan cycle. **b**, Reconstruction results from the scan and related FFT graph. **c**, **d**, The same as (a) and (b) but with jump scan. **e**, Reconstruction results with different bleaching rates. **f**, Cosine similarity between the point source image and the reconstructed image using sequential and jump scan.

Subsequently, we conducted simulations to analyze the impact of scan sequence on the bleaching curve and its fitting accuracy. A line source (1×256 pixels at the middle of a 256×256 grid) was bleached during 8 scan cycles. The center pixels were used to plot the 'bleaching curve'. By altering the orientation of the line source, we generated a series of distorted bleaching curves. The envelopes of these curves, plotted alongside the actual bleaching curve, demonstrate that distortion exacerbates with increasing bleaching rates (Figure R 21e, f). Utilizing these distorted bleaching curves, we quantified the estimation errors for both the PATTERN signal and the bleaching rate, presenting them as an upper limit of the relative error. Notably, the jump scan strategy effectively suppresses the error in both the PATTERN signal and the bleaching rate compared to the sequential scan (Figure R 21g, h). It is important to highlight that high relative errors predominantly occur when the feature size becomes comparable to the field of view.

Figure R 21 | Optimization of rotation order for quantification. **a**, Diagram of the sequential scan of a line source which is exponentially bleached during eight scan cycles. **b**, The same as (a) but with jump scan. **c**, The bleaching curve calculated from the reconstruction results. **d**, The same as (c) but with jump scan. **e**, **f**, Measured distortion of bleaching curve induced by feature's directionality: (e) sequential scan, (f) jump scan. **g**, **h**, Maximum relative error of PATTERN signal amplitude and bleaching rate, induced by feature's directionality: (g) sequential scan, (h) jump scan.

Revisions:

- (1) The original Extended Data Fig. 6 has been replaced by Figure R 20 (Supplementary Fig. 7) and Figure R 21 (Supplementary Fig. 8) and referred in the Results section (page 12 lines 18–22):

“In contrast, if we were to scan in the angular direction in an incremental manner (0° , 5.625° , 11.250° , ...), uneven filling of k-space in the azimuthal direction occurs due to the gradual signal decay from photobleaching, leading to a reduction in both image quality and fidelity of the bleaching curve (Fig. 2a and Supplementary Figs. 7, 8).”

(2) A description of the above simulation process was added to the Methods section (pages 44–45):

“The rotation order of PATTERN was designed and verified via simulation using MATLAB (Supplementary Figs. 7, 8). All parameters used in the simulation were consistent with those employed for the real samples. For comparison purposes, a sequential scan scheme, -90° , -84.375° ..., 0° ..., 84.375° , was simulated as the control group. During the simulation, we assumed that a point source (1×1 pixel at the center of a 256×256 grid) was bleached during a scan cycle. We assessed the image degradation resulting from photobleaching by comparing the reconstructed image with the ground truth using cosine similarity. Next, a line source (1×256 pixels at the middle of a 256×256 grid) was bleached during an eight-cycle scan. The center pixels were counted to plot the ‘bleaching curve’. By altering the orientation of the line source, we generated a series of distorted bleaching curves. The envelopes of these curves, plotted alongside the actual bleaching curve, demonstrate that distortion exacerbates with increasing bleaching rates. Utilizing these distorted bleaching curves, we quantified the estimation errors for both the PATTERN signal and the bleaching rate, presenting them as an upper limit of the relative error.”

RC2.12:

5. Deep Learning and Elevational Resolution:

- The use of DnCNN for enhancing PATTERN signals raises an interesting possibility of leveraging multi-angled scanning results as the ‘ground truth’ to improve the elevational resolution of single-angled scanning results. Exploring this idea further could potentially lead to enhanced elevational resolution without extensive scanning or improved results with reduced scanning angles.

AR2.12:

We appreciate the reviewer's insightful suggestion and agree that it is indeed a promising proposal. In recent literature, several deep-learning-based methods have been developed to address the challenges of sparse sampling and limited-view problems in PA imaging. These methods align well with the reviewer's idea. We evaluated a UNet-based network²⁶, focusing on xz-plane slices as the network inputs. However, the frequency component loss was too serious in each translation scan (Supplementary Fig. 2). Moreover, the three-dimensional PA data exhibit a strong correlation among neighboring slices in the xz-plane. Therefore, the dataset needed for training would be much larger than in the traditional two-

dimensional situation, and advanced network design is necessary. We recognize the feasibility of this idea, but definitive conclusions are not available at present.

Revisions:

The idea of implementing deep-learning to further improve system performance was added to the Discussion section (page 26 lines 20–28):

“Additionally, several measures can be implemented to enhance PATTERN’s performance and impact. Firstly, faster lasers or potentially, some deep-learning-based image fusion approaches⁵⁸ can be employed to accelerate the imaging process, while the imaging resolution can be improved by using ultrasound transducers with larger bandwidths. Specifically, the utilization of transducers with a frequency response ranging from direct current (DC) to 22 MHz has demonstrated the ability to achieve a resolution of approximately 50 μm ⁵⁷, which is comparable to the resolution achieved in all-optical, large-FOV brain imaging⁵².”

Reference

- 1 Michels, R., Foschum, F. & Kienle, A. Optical properties of fat emulsions. *Optics Express* **16**, 5907–5925 (2008). <https://doi.org:10.1364/OE.16.005907>
- 2 Jathoul, A. P. *et al.* Deep in vivo photoacoustic imaging of mammalian tissues using a tyrosinase-based genetic reporter. *Nature Photonics* **9**, 239 (2015).
- 3 Pan, C. *et al.* Shrinkage-mediated imaging of entire organs and organisms using uDISCO. *Nature Methods* **13**, 859–867 (2016). <https://doi.org:10.1038/nmeth.3964>
- 4 Matsumoto, K. *et al.* Advanced CUBIC tissue clearing for whole-organ cell profiling. *Nature Protocols* **14**, 3506–3537 (2019). <https://doi.org:10.1038/s41596-019-0240-9>
- 5 Jing, D. *et al.* Tissue clearing of both hard and soft tissue organs with the PEGASOS method. *Cell Research* **28** (2018). <https://doi.org:10.1038/s41422-018-0049-z>
- 6 Tomer, R., Ye, L., Hsueh, B. & Deisseroth, K. Advanced CLARITY for rapid and high-resolution imaging of intact tissues. *Nature Protocols* **9**, 1682–1697 (2014). <https://doi.org:10.1038/nprot.2014.123>
- 7 Ragan, T. *et al.* Serial two-photon tomography for automated ex vivo mouse brain imaging. *Nature methods* **9**, 255–258 (2012). <https://doi.org:10.1038/nmeth.1854>
- 8 Seiriki, K. *et al.* High-Speed and Scalable Whole-Brain Imaging in Rodents and Primates. *Neuron* **94**, 1085–1100.e1086 (2017). <https://doi.org:10.1016/j.neuron.2017.05.017>
- 9 Zhong, Q. *et al.* High-definition imaging using line-illumination modulation microscopy. *Nature Methods* **18**, 309–315 (2021). <https://doi.org:10.1038/s41592-021-01074-x>
- 10 Cui, M. *et al.* Adaptive photoacoustic computed tomography. *Photoacoustics* **21**, 100223 (2021).
- 11 Chen, Y. *et al.* Photoacoustic Mouse Brain Imaging Using an Optical Fabry-Pérot Interferometric Ultrasound Sensor. *Frontiers in neuroscience* **15**, 572 (2021).
- 12 Lin, L. *et al.* High-speed three-dimensional photoacoustic computed tomography for

- preclinical research and clinical translation. *Nature communications* **12**, 1-10 (2021).
- 13 Cao, R. *et al.* Single-shot 3D photoacoustic computed tomography with a densely packed array for transcranial functional imaging. arXiv:2306.14471 (2023). <<https://ui.adsabs.harvard.edu/abs/2023arXiv230614471C>>.
- 14 Lu, T. *et al.* Full-frequency correction of spatial impulse response in back-projection scheme using space-variant filtering for optoacoustic mesoscopy. *Photoacoustics* **19**, 100193 (2020). [https://doi.org:https://doi.org/10.1016/j.pacs.2020.100193](https://doi.org/https://doi.org/10.1016/j.pacs.2020.100193)
- 15 Xia, J. *et al.* Three-dimensional photoacoustic tomography based on the focal-line concept. *Journal of Biomedical Optics* **16**, 090505-090505-090503 (2011). <https://doi.org:10.1117/1.3625576>
- 16 Seeger, M. *et al.* Pushing the boundaries of optoacoustic microscopy by total impulse response characterization. *Nature Communications* **11**, 2910 (2020). <https://doi.org:10.1038/s41467-020-16565-2>
- 17 Van de Sompel, D., Sasportas, L. S., Jokerst, J. V. & Gambhir, S. S. Comparison of Deconvolution Filters for Photoacoustic Tomography. *PLoS ONE* **11** (2016).
- 18 Zhang, J. *et al.* In vivo characterization and analysis of glioblastoma at different stages using multiscale photoacoustic molecular imaging. *Photoacoustics* **30**, 100462 (2023).
- 19 Yao, J. *et al.* Multiscale photoacoustic tomography using reversibly switchable bacterial phytochrome as a near-infrared photochromic probe. *Nature Methods* **13**, 67-73 (2016). <https://doi.org:10.1038/nmeth.3656>
- 20 Margrie, T. W. *et al.* Targeted Whole-Cell Recordings in the Mammalian Brain In Vivo. *Neuron* **39**, 911-918 (2003). [https://doi.org:https://doi.org/10.1016/j.neuron.2003.08.012](https://doi.org/https://doi.org/10.1016/j.neuron.2003.08.012)
- 21 Gottschalk, S. *et al.* Rapid volumetric optoacoustic imaging of neural dynamics across the mouse brain. *Nature biomedical engineering* **3**, 392-401 (2019).
- 22 Li, L. *et al.* Small near-infrared photochromic protein for photoacoustic multi-contrast imaging and detection of protein interactions in vivo. *Nature communications* **9**, 2734 (2018).
- 23 Li, X. *et al.* Three-dimensional structured illumination microscopy with enhanced axial resolution. *Nature Biotechnology* **41**, 1307-1319 (2023). <https://doi.org:10.1038/s41587-022-01651-1>
- 24 Olefir, I. *et al.* Spatial and spectral mapping and decomposition of neural dynamics and organization of the mouse brain with multispectral optoacoustic tomography. *Cell Reports* **26**, 2833-2846. e2833 (2019).
- 25 Li, L. *et al.* Label-free photoacoustic tomography of whole mouse brain structures ex vivo. *Neurophotonics* **3**, 035001 (2016). <https://doi.org:10.1117/1.NPh.3.3.035001>
- 26 Davoudi, N., Deán-Ben, X. L. & Razansky, D. Deep learning optoacoustic tomography with sparse data. *Nature Machine Intelligence* **1**, 453-460 (2019). <https://doi.org:10.1038/s42256-019-0095-3>
- 27 Allouche-Arnon, H. *et al.* Computationally designed dual-color MRI reporters for noninvasive imaging of transgene expression. *Nature Biotechnology* **40**, 1143-1149 (2022). <https://doi.org:10.1038/s41587-021-01162-5>
- 28 Chen, Z. *et al.* Hybrid magnetic resonance and optoacoustic tomography (MROT) for preclinical neuroimaging. *Light: Science & Applications* **11**, 332 (2022). <https://doi.org:10.1038/s41377-022-01026-w>

- 29 Machado, T. A., Kauvar, I. V. & Deisseroth, K. Multiregion neuronal activity: the forest and the trees. *Nature Reviews Neuroscience* **23**, 683-704 (2022).
<https://doi.org/10.1038/s41583-022-00634-0>
- 30 Lichtman, J. Imaging the Connectome. *Biophysical Journal* **108**, 23a (2015).
<https://doi.org/https://doi.org/10.1016/j.bpj.2014.11.148>
- 31 Abbott, L. F. *et al.* The Mind of a Mouse. *Cell* **182**, 1372-1376 (2020).
<https://doi.org/https://doi.org/10.1016/j.cell.2020.08.010>
- 32 Fang, Q. & Boas, D. Monte Carlo Simulation of Photon Migration in 3D Turbid Media Accelerated by Graphics Processing Units. *Optics express* **17**, 20178-20190 (2009).
<https://doi.org/10.1364/OE.17.020178>

Reviewer #3:

(RC: Reviewer's Comment, AR: Author's Response)

RC3.0:

In the work with the title 'Photoacoustic Tomography with Temporal Encoding Reconstruction (PATTERN) for Cross-Modal Individual Analysis of the Whole Brain', Yang et al. introduce an imaging platform (PATTERN) for the non-invasive, fast, three-dimensional ex-vivo imaging of small animal brains. The manuscript is generally well-written with high quality figures but I think some more clarifications and additional work (major revision) would be needed:

AR3.0:

Thank you for your invaluable feedback. We deeply appreciate your thoughtful comments, and we have taken each of them into account. Consequently, we have made corrections and supplemented new experimental data to enhance the overall quality of our manuscript. Your insights have been instrumental in refining our work, and we are grateful for the opportunity to improve based on your guidance..

We recognize that your main questions may arise from a lack of clarity in our initial manuscript describing the advantages of the PATTERN system compared to existing PA systems. We will now concisely outline the primary novelty of PATTERN here, with detailed information provided in the point-to-point response:

Compared to existing PA systems, the PATTERN system is uniquely designed to improve the detection and high-fidelity imaging of fluorescent proteins—an aspect not fully explored in previous designs. Notably, the PATTERN technique excels in reconstructing the spatial distribution of fluorescent protein markers with unparalleled quality, representing a groundbreaking achievement with vast potential for advancing the applications of PA imaging.

By employing a temporal encoding strategy based on photobleaching, our system markedly boosts sensitivity in detecting fluorescent proteins, surpassing existing PA imaging methods by an order of magnitude, as shown in Figures 1 and 2. In the context of circuit-level brain fluorescence analysis, our advancement positions PATTERN on par with certain optical methods, which are conventional but destructive approaches for measuring fluorescent signals.

We have revised both the Introduction and Discussion sections of the manuscript to improve clarity on the above aspects. Detailed responses addressing your concerns are outlined in the point-to-point response below.

RC3.1:

Introduction

- The authors present their method as an improvement compared to optical imaging methods without explaining why optical imaging would be accepted as the gold-standard small animal imaging method. In other words, a brief overview (with main pros and cons of each one) regarding other imaging technologies (e.g., magnetic resonance imaging, MRI) available for whole-brain small animal imaging is needed.

AR3.1:

We appreciate this suggestion. In our initial submission, we mainly compared our system to other optical imaging methods because the distinguishing advantage of our system, compared to previous photoacoustic methods, is the ability to detect genetically encoded fluorescent tags. Thus, we regarded optical imaging as a suitable reference specifically for fluorescent imaging, it was not meant to regard it as a gold standard at other imaging levels. However, we agree with the importance of discussing the potential and limitations of other non-optical whole-brain imaging technologies, such as MRI-based approaches²⁷. Related discussion is helpful to lead readers to the future directions of non-destructive whole-brain imaging and we have incorporated discussions in the revised manuscript.

Revisions:

Discussion about MRI-based fluorescent imaging has been added (page 29 lines 14–25):

“Due to its capacity to obtain the whole-brain 3D fluorescence distribution in a fast, nondestructive, and high-throughput manner, PATTERN provides a new way of quickly and conveniently previewing the fluorescence signal quality of samples. It provides effective evaluation and guidance before conducting subsequent high-resolution imaging experiments or other high-cost experiments, such as single-cell sequencing and spatial transcriptomics. Therefore, incorporating PATTERN into the specimen analysis pipeline enhances the success rate and reduces overall costs. Therefore, incorporating PATTERN into the specimen analysis pipeline enhances the success rate and reduces overall costs. Notably, emerging magnetic resonance (MR)-based reporter gene imaging demonstrates the potential for non-destructive, in vivo whole-brain detection of gene-expression patterns⁶⁰. However, the diversity of available labels for this approach is still limited.”

RC3.2:

- Line 24: I suppose that the word ‘metabonomic’ should be changed into ‘metabolomic’.

AR3.2:

As suggested, we indeed intended to use a more systematic term that represents a category of disciplines. Therefore, we replaced 'metabonomic' with 'metabolomic' in revised manuscript (page 3 line 24).

RC3.3:

- Indeed, the method provides high quality morphological brain imaging. However, based on the fact that several studies focusing not only on morphological but also functional brain imaging with photoacoustics have been already published (DOI: 10.1038/s41551-019-0372-9, DOI: <https://doi.org/10.1038/s41377-022-01026-w>, DOI: 10.1016/j.celrep.2019.02.020), I would like to ask the authors whether the system could be also used for functional brain imaging. If yes, could you please provide us with some data? If not, could you please explain more clearly what is the exact novelty of the work? Lower cost? Faster scanning times? Isotropic resolution? Higher resolution? A thorough comparison with other photoacoustic brain studies would be required.

AR3.3:

The key advancement of PATTERN over the previous system lies in the improved detection of weak signals from fluorescent proteins, as depicted in Figs. 1-2. These enhancements notably improve both resolution and sensitivity. While preliminary works have reported photoacoustic-based functional imaging of activity-dependent fluorescent sensors, critical questions persist, including sensitivity to fluorescent tags and fidelity of the reconstructed distribution of reporter genes. These challenges represent obstacles to the widespread application of PA imaging in neuroscience.

The major innovation presented in the current work is the utilization of photobleaching-based temporal encoding for achieving highly sensitive and accurate imaging of genetically encoded fluorescent proteins within optically opaque biological tissues. We have developed a system alongside a comprehensive imaging protocol to implement temporal encoding, resulting in the generation of 3D photoacoustic images of unparalleled quality and sensitivity to genetically encoded fluorescent tags. Most of the existing studies have not addressed the specific problem explored in the current work, namely, accurately and sensitively delineating the whole-brain distributions of genetically encoded fluorescent proteins in the rodent model (Table R 2 columns 2, 3).

To achieve functional brain imaging via activity-dependent fluorescent proteins (commonly used in animal models, such as GCaMP and voltage sensors) in a photoacoustic (PA) system, enhancing sensitivity to these sensors and improving temporal resolution are two key points. The work mentioned by the reviewer (DOI: 10.1038/s41551-019-0372-9) primarily focuses on improving the temporal resolution to detect the GCaMP signal, while our research is optimized to increase the sensitivity to a broader range of fluorescent

proteins. Such optimization benefits both morphological brain imaging and functional imaging, in the context of utilizing fluorescent proteins. While functional imaging is not explicitly demonstrated in the current work, the PATTERN technology represents a significant stride towards better PA-based functional imaging strategies.

Although PATTERN focuses on improving the detection of fluorescent proteins, it also achieves higher performances in other key features. Table R2 compares the specifications and representative images of various recently published PA brain imaging technologies, alongside those of PATTERN.

Table R 2 | Comparison of recent studies on PA brain imaging technologies.

	Fluorescent protein imaging	Contrast mechanism	Representative images	Field of view/ Resolution
UZH ²¹ (Nature Biomed. Eng. 2019)	Yes	Whole brain expressed with GCaMP6f, signal was relatively strong.		FOV: 0.256 cm ³ Resolution: 150 μm (Brain images appear to have worse resolutions)
Caltech ^{1 2} (Nature Comm. 2022)	No	Blood (strong signal)		FOV: 2.2 cm ³ Resolution: Lateral: 390 μm Axial: 370 μm
HZM ²⁴ (Cell Reports 2019)	No	Blood (strong signal) and weak signals from lipids		FOV: 3.5 cm ³ Resolution: In plane: 150 μm Elevational: 800 μm (Blurry in the elevational direction)

UZH ²⁸ (Light: Science & Appl., 2022)	No	Blood (strong signal)		FOV: 0.256 cm ³ Resolution: 163 μm (Brain images appear to have worse resolutions)
This work	Yes	Weak signals from lipids and genetically encoded proteins		FOV: 11 cm ³ Resolution: <140 μm (isotropic; within the entire FOV)

The comparison above indicates that PATTERN outperforms existing technologies in almost all key aspects, including overall image quality, FOV, and resolution (Table R2, columns 4, 5). More importantly, it demonstrates unprecedented reconstruction quality for the spatial distribution of fluorescent labels.

Furthermore, considering the experimental conditions varied across different studies, we have incorporated several new experiments by utilizing several classic photoacoustic imaging systems that are similar to the mentioned PA systems. To ensure a fair comparison between PATTERN and other photoacoustic imaging platforms, we obtained brain images using additional photoacoustic imaging systems available in our lab, striving to maintain consistency in experimental conditions. From this series of experiments, we also concluded that PATTERN consistently produced superior images (Figure R 22).

The experimentally compared systems included a full ring array system¹⁰, a hemisphere array system (the Endra NEXUS 128 system), and an optical-fiber-sensor-based PA mesoscope¹¹. Below we summarize what we found in the tests.

Figure R 22 | Brain imaging using other systems. a, b, Imaging results using a full ring array system. **c,** Brain sample inside NEXUS-128. **d,** Imaging results using NEXUS-128, the brain should be in the dashed white circle. **e,** Brain sample inside a fiber sensor based photoacoustic mesoscope. **f,** Imaging results using the mesoscope, the brain contours was sketch by white line.

- (a) The homemade full ring array system comprises a circular ultrasound array with 256 sensing elements (Imasonics Inc.), geometrically focused in the elevational direction for acoustic sectioning. The sensing elements have a central frequency of 5.5 MHz and a receiving bandwidth of 60%. The system exhibits a limited resolution of ~ 1 mm in the elevational direction, resulting in unsatisfactory imaging outcomes (Figure R 22a, b).
- (b) The NEXUS 128 system employs a hemispherical sensor array but demonstrates lower sensitivity compared to the PATTERN system. We illuminated the sample using 700 nm laser and performed angular scan in 120 steps with 30 averages per step, the entire scanning process took 3.9 minutes, in comparison to a scan time of 2.2 minutes

for PATTERN. However, the brain sample remained almost undetectable. A horse tail mane was inserted into the sample for localization purposes (Figure R 22c, d). It is important to acknowledge that the NEXUS 128 system was acquired in 2017 and has undergone extensive use. Consequently, its current performance may not accurately reflect that of a typical hemispherical array system.

- (c) The optical-fiber-sensor-based PA mesoscope, equipped with a custom-made ultrasound sensor at the fiber tip and a detection bandwidth ranging from direct current (DC) to 23 MHz¹¹, could potentially achieve a spatial resolution of approximately 50 μm . This sensor underwent raster scanning, effectively representing a planar array. However, the signal-to-noise ratio appeared insufficient for imaging the perfused brain (Figure R 22e, f).

The imaging results demonstrate that the alternative systems were inadequate for the application in this study.

Based on the additional experimental observations reported above, we summarized the advantages of the PATTERN system as below:

Improved resolution: Using the multiangle image fusion strategy, resolution was enhanced dramatically, as indicated in Fig. 2c. (Figure R 23a). This is also highlighted by the comparison with published results (Figure R 23b, c). It is essential to note that the images obtained using other systems shown in rows 3 and 4 of Table R1, also illustrated in Figure R 23d below, appear to have lower image qualities, although the claimed resolutions are similar to ours. This is due to the use of different statistical methods for resolution estimation.

High sensitivity: The photoacoustic signals originating from fluorescent proteins and perfused brain tissue were notably weaker in comparison to those from blood. In the context of this work, the ability to detect signals from the perfused brain is a fundamental requirement. Brain samples designated for the PATTERN system underwent imaging using other platforms in our lab, including a commercial hemispherical array system (Figure R 22c, d) and a homemade photoacoustic mesoscope (Figure R 22e, f). However, their performances exhibited significantly lower sensitivity.

Figure R 23 | Comparison on image resolution. **a**, A copy of Fig. 2c. **b**, Imaging results of platforms listed in Table R1 row 2 (upper) and our system (lower). **c**, Imaging results of a ring-array-based system (upper) and our system (lower). **d**, Imaging results of a hemisphere system listed in Table R1 rows 3 and 4.

Revisions:

(1) The major novelty was further emphasized in the Introduction section (page 5 lines 10 –19):

“Taking advantage of the photobleaching of the fluorescent tags³⁹ that exhibit a characteristic temporal decay, we designed the scanning sequence to specifically extract the signals from the tags, using a single illumination wavelength. We utilized photobleaching, which is typically seen as a hindrance, in a new way to suppress the intrinsic tissue background, resulting in the generation of 3D photoacoustic images of enhanced detection sensitivity specifically for these fluorescent tags (Fig. 1a). We further enhanced the detection accuracy by deep learning. Accordingly, we named our new

imaging technology Photoacoustic Tomography with Temporal-Encoding Reconstruction (PATTERN).”

(2) The above imaging results using other systems in our lab are added to the manuscript as Supplementary Fig. 19 and referred in the Discussion section (page 26 lines 6–14):

“To meet the requirement of high-resolution and high-throughput whole-brain PA imaging, a system with point-like ultrasound transducers is preferred for the best imaging quality (isotropic resolution)^{14,27,34,57}. However, the reported PA systems of this kind, to our knowledge, are not suitable for the tasks reported in the current work. Difficulties include low resolution or sensitivity to clearly distinguish brain regions, limited FOV for large brains, and excessive laser exposure for fluorescent proteins (Supplementary Fig. 19). Challenges arise due to the difficulty in manufacturing sensitive, broadband, and small-footprint ultrasound transducers.”

(3) The adaptability of the PATTERN pipeline to various imaging platforms was discussed in the Discussion section (page 26 line 24–page 27 line 2):

“Specifically, the utilization of transducers with a frequency response ranging from direct current (DC) to 22 MHz has demonstrated the ability to achieve a resolution of approximately 50 μm ⁵⁷, which is comparable to the resolution achieved in all-optical, large-FOV brain imaging⁵². Secondly, regardless of the detailed implementations, the PATTERN concept is adaptable to PAT platforms involving various scan strategies and could potentially enhance their sensitivity to fluorescent proteins.”

RC3.4:

Results

- Near isotropic photoacoustic imaging of PATTERN: The authors provide resolutions of approximately 100-150 μm . Based on the fact that several even clinical photoacoustic systems, which would definitely have (or even need) lower resolutions to image human tissues, almost achieve these resolutions (also in real-time), I would like to ask whether the authors believe that the spatial resolutions achieved are enough to image the fine small animal brain structures. What do the other preclinical photoacoustic platforms achieve? For example, what is the smallest biologically- (or disease-) relevant small animal brain structure that could be resolved with the developed system?

AR3.4:

We believe this concern also arises from the unclear expression of the major novelty of our system in the initial submission and we mainly aimed to enhance the imaging quality of fluorescent proteins in this new PA imaging method. It's important to note that while most preclinical PA systems are optimized for imaging blood and vascular structures, or for rapidly capturing functional signals, few are specifically designed to detect fluorescent protein signals in the brain. Thus, the distinct advantage of our system is its ability to image

fluorescent proteins at the whole-brain level. This specialized capability is important and unique for brain studies considering fluorescent proteins are widely used in numerous biological studies.

The resolution provided by PATTERN is sufficient for brain studies in small animal models. Presently, brain imaging predominantly centers on three distinct scales: synaptic levels, neuronal levels, and circuitry levels. A recent review paper introduced multiple techniques for imaging the brain at different levels²⁹. It showed that regional methods like OEG (resolution: about 100µm) and photoacoustic computed tomography (resolution about 150 µm) are capable of studying the brain at the circuit level in mice (Figure R 23)²⁹. The resolution of PATTERN is comparable to these methods. From another perspective, as illustrated in Figure 4, the PATTERN-derived projection map for many brain regions closely aligns with that obtained from the Allen Brain database. Thus, we believe PATTERN is effective in investigating the mouse brain at circuitry levels.

Figure R 24 | Examples of brain imaging technologies with different spatial resolution²⁹.

RC3.5:

- Temporal encoding and unmixing of fluorescent tags by PATTERN: The authors characterize the spectral unmixing 'extremely complex'. I find it absolutely useful to provide more objectified data on the superiority of the temporal unmixing approach followed. Which are exactly the weaknesses of the spectral unmixing approaches?

AR3.5:

The term 'extremely complex' in our context refers to the considerable challenges involved in applying spectral unmixing to our 3D imaging system. The intricacy of this phenomenon is accentuated by the data presented in Supplementary Fig. 5b (copied below as Figure R 25), illustrating a time-varying spectrum for the same image voxel in consecutive scans. The spectral fluctuations observed were attributed to photobleaching. It is noteworthy that a time-varying photoacoustic spectrum can pose significant challenges to spectral unmixing. Another example is illustrated in Figure R 26, where fluorescent proteins can be almost entirely bleached in a single scan cycle (note that $\exp(-0.09 \times 32) \approx 0.056 \ll 1$, where the bleaching rate $b = 0.09/\text{translational scan}$ for some image voxels, and a single scan cycle took 32 translational scans).

Figure R 25 | Left: a typical coronal slice of a mouse brain with fluorescent protein. Right: a time-varying spectrum for the same image voxel in consecutive scans. (from Supplementary Fig. 5a, b)

Figure R 26 | **Bleaching rates of a sample.** **a**, Bleaching rates distribution of a cross-section of a brain. **b**, histogram of the bleaching rates b .

Figure R 27 | Spectral distortion occurs during data acquisition

Figure R 27 further illustrates the impracticality of spectral unmixing in the presence of photobleaching. Let's assume a sequential brain imaging scenario starting at discrete wavelengths, such as 680 nm, 690 nm, 700 nm, and so on. At each wavelength, the sample undergoes a complete translation-rotation scan cycle involving hundreds of excitation pulses before the mechanical scan restarts with the excitation switched to the next wavelength. During this process, due to photobleaching, the measured PA signal significantly decreases at the subsequent wavelength of 690 nm (red curve in Figure R 27), even though the actual absorption at 690 nm remains high (blue curve in Figure R 27). Since accurate spectral measurement is essential for any spectral-unmixing methods, the influence of photobleaching inherently renders spectral analysis unreliable.

The novelty of this work lies in transforming photobleaching, typically considered detrimental to optical imaging, into a mechanism for highly accurate detection of fluorescent tags through temporal encoding. What's even more intriguing is our capability to conduct quantitative analysis of fluorophore concentration by leveraging the nonlinearity inherent in the bleaching process (AR2.5).

Revisions:

The impracticality of spectral unmixing was emphasized in the Results section (page 11 lines 3–13):

“To show the superiority of temporal unmixing over spectral unmixing in the translation-rotational scan scheme, we tested both methods. During the scan process, repeated laser exposure caused signal decay due to photobleaching (Supplementary Fig. 5a, b). The change in the signal strength distorts the measured photoacoustic spectrum, making spectral unmixing extremely complex. In addition, in our spectral sweeping window (680-1064 nm), there is a limited variety of usable fluorescent tags (Supplementary Fig. 5c). Conversely, the temporal unmixing method involves single-wavelength operation only, thus making full use of the pump laser (532 nm) to detect red fluorescent proteins (i.e., mScarlet).

These factors prompted us to adopt the temporal unmixing method to improve the sensitivity to signals from the labels.”

RC3.6:

- Temporal encoding and unmixing of fluorescent tags by PATTERN: I would be grateful to have some more information on how the ‘artefacts’ were identified? Could you provide extra clarifications on the usual artefacts observed with the technology presented?

AR3.6:

In theory, these artefacts (i.e., 'false positive signals') primarily arise from imperfections in raw photoacoustic images, particularly due to the suboptimal response of ultrasound transducers. Although deconvolution methods can mitigate these issues, they often do so at the expense of image SNR¹⁷. To clarify, we consider the impulse response of an ultrasound transducer: ideally, the transducer has an infinite bandwidth, resulting in a single peak in the time domain that precisely matches the input pressure waveform (Figure R 28a, b). In practice, however, the response is bandlimited and may exhibit harmonically oscillating tails following the excitation (Figure R 28c). These tails are also projected into the image during signal reconstruction, as indicated by the blue transducer and its projection line. This leads to 'shadows' around the point source (highlighted by the red arrow) and 'ripples' (encircled by the white dashed line), which are recognized as artefacts in raw PA images (Figure R 28d). While these artefacts are weak in amplitude, they follow the same temporal decoding curve as the point source in PATTERN, leading to their erroneous interpretation as PATTERN signals. Traditional methods, especially those used in spectral unmixing, aim to enhance raw PA image quality by employing reconstruction techniques to compensate for the transducers' nonideal response. However, these solutions either compromise the SNR or introduce other types of artefacts.

Based on the knowledge about artefacts, the identification of these artefacts is facilitated by the combination of PATTERN imaging with confocal imaging. We conducted sequential imaging of the same brain with both the PATTERN system and a confocal microscope, utilizing the confocal images as a reliable reference for artefact identification (Figure R 29, copied from Supplementary Fig. 9a, b in revised manuscript). More specifically, the presence of EGFP (for confocal imaging) and iRFP713 (for PATTERN imaging) signals within the same brain regions allows the alignment and registration of each confocal image to the PATTERN image. This alignment served as a template for human experts to accordingly label artefact regions within the PATTERN data. To ensure accuracy, only signals evident in the PATTERN images but absent in the reference were designated as artefacts, mitigating potential mislabeling (Figure R 29).

Figure R 28 | Illustration on the artefact generation mechanism. **a**, Response of ultrasound transducers of ideal (top) and nonideal (bottom) situations. **b**, Corresponding reconstructed raw photoacoustic images.

Figure R 29 | Working flowchart of DnCNN performance and artefact identification. **a**, Illustration of the workflow of DnCNN. Datasets are constructed in the training phase based on confocal microscopy as the reference. Network after training are used in the evaluation phase to remove the artefacts. **b**, Pipeline of artefact identification. Images with artefacts and the artefacts-removed ones were collected as datasets.

Revisions:

The above illustration shown in Figure R 28 and Figure R 28 was added to Supplementary Figs. 11, 9b and was referred in the Results section (page 13 lines 16–23):

“For further improving our system, we next used confocal microscopy images as a reliable reference and employed a denoising convolutional neural network (DnCNN)⁴³ to remove the potential false-positive signals (Fig. 2a). Relying on a single-batch learning strategy, DnCNN was capable of extracting artefacts in the vicinity of true signals (Fig. 2g, Supplementary Fig. 9). The majority of the artefacts caused by the unideal response of the ultrasound transducers (Supplementary Fig. 11) were effectively eliminated without causing severe false-negative signals compared to traditional deconvolution methods (Fig. 2h).”

RC3.7:

- PATTERN-based whole-brain optical imaging: I believe that the best way to characterize the technology developed is photoacoustic and not optical (2nd line). This is why the main comparisons should be done against other photoacoustic (and not optical) imaging platforms.

AR3.7:

We agree that the comparison between our system with other PA systems is clearer to show the advancement of the PATTERN system, as we have addressed in our response to the Reviewer's comment RC3.3. Our comparison led to the conclusion that utilizing other photoacoustic systems as a basis for comparison may not effectively yield the necessary fluorescent information.

More importantly, in this part of the manuscript, we mainly aimed to demonstrate the ability of the PATTERN to visualize the fluorescent proteins in the brain, as well as its potential to measure the expression of viral vectors and the long-range projection from specific brain regions. Therefore, we employed optical imaging for validation but not for comparison, considering that confocal imaging is a widely utilized technique for these analyses. Notably, a recent study has also used optical imaging methods to validate the effectiveness of a non-optical (MRI-based) imaging technology (Figure R 30)²⁶.

Furthermore, PATTERN imaging, along with photoacoustic imaging in general, exhibits image contrast that closely aligns with that of fluorescence imaging. Both photoacoustic imaging and fluorescence imaging derive their image contrast from light absorption. While photoacoustic imaging detects light absorption through nonradiative relaxation (i.e., photoacoustic emission), fluorescence imaging captures the absorption signal through radiative relaxation (i.e., fluorescence emission). Therefore, employing optical imaging as the basis for validation is physically accurate.

Figure R 30 | Comparison of MRI and confocal imaging. Left: Pseudo-color images of MRI reporter genes overlaid on anatomical MRI images; Right: corresponding fluorescent images of brain sections of the same mice²⁶.

Revisions:

Descriptions have been modified in the Results section (page14 lines 3–13).

“Utilizing photobleaching-based temporal encoding, the PATTERN system has been enhanced to detect fluorescent proteins within brain tissue effectively. To further validate the accuracy of our system for fluorescent protein detection, we employed confocal data from the same brain samples, which contained both AAV-expressed iRFP713 (for PATTERN imaging) and EGFP (for confocal imaging). Using confocal images as the references, PATTERN images showed consistent fluorescent signals (Fig. 3d). This consistency encouraged us to explore more neuroscience applications. Firstly, PATTERN exhibited the ability to visualize the implants in the brain such as the optical fibers used in optogenetics and the electrodes used in electrophysiology (Supplementary Fig. 12a-d)...”

RC3.8:

- PATTERN for visualizing neural connectivity of the brain: Since exploration of neural connectivity is indeed important, the reader should be provided with more information about its importance.

AR3.8:

We appreciate this constructive suggestion. We revised the expression and added some citations^{30,31} to help readers understand the significance of investigating neural connectivity.

Revisions:

More descriptions and citations have been added in the main text (page 16 lines 16–17):

“Revealing the connectivity of the brain is crucial for understanding its functions and dysfunctions^{44,45}. Since PATTERN can achieve reliable 3D whole-brain optical imaging, we sought to explore its potential to visualize neural circuits. For that purpose, we injected an AAV vector to express iRFP713 fused with EGFP in the brain area of interest, allowing us to validate the PATTERN-traced projections by subsequent other optical imaging methods (Fig. 4a).”

RC3.9:

Discussion

- Comparison and combination of PATTERN with other whole-brain optical imaging technologies: As already mentioned, based on the fact that several other photoacoustic brain imaging studies have been already published, I believe that a comparison to them (and not optical imaging techniques) would be needed. Taking into account the findings of the study and the claims of the authors, the comparison (apart from general features) would also focus on information about: ‘injection sites of vectors, fluorescence expression intensity or signal loss during sample preparation’.

AR3.9:

We agree that comparing our system with existing photoacoustic systems in the discussion will enhance the clarity of our manuscript. Following the suggestion, we have incorporated the distinctions and novelty of our system compared to the published PA systems mentioned by the reviewer in the revised Discussion section. Notably, since not all referenced works imaged fluorescent proteins and none employed viral vectors, we did not compare all suggested features and conducted respective comparisons with each method in the revision, taking into account their specific characteristics.

Revisions:

The comparison of the PATTERN system with existing photoacoustic systems was added in the Discussion section (page 25 line 16–page 26 line 5):

"Comparison of PATTERN with other PA brain imaging studies

Given its capacity to identify optical absorption contrasts in deep tissue with high resolution, PA imaging has gained much attention in brain research. Recently, various PA imaging systems have been developed to meet the specific needs of different neuroscience studies. Contrasts between oxyhemoglobin and deoxyhemoglobin have rendered PA imaging an effective tool for visualizing the distribution of oxygen saturation^{14,19}, especially when combined with functional magnetic resonance imaging (fMRI)⁵⁶. Using exogenous fluorescent probes, PA imaging has been also explored for visualizing enhanced molecular details deep within the brain¹⁹. With the enhancement of temporal resolution, PA imaging has also demonstrated its capability to detect GCaMP signals²⁷. However, the ability and sensitivity of detecting fluorescent proteins, which are prominent requirements for neuroscience studies, are still not further optimized in most photoacoustic imaging techniques. Thus, the primary focus of the PATTERN system was to improve sensitivity and image fidelity across a broader range of fluorescent proteins, thereby extending the potential applications of photoacoustic imaging in neuroscience studies."

RC3.10:

Methods

- PATTERN: Why did you select a transducer with central frequency of 5.5 MHz? And not a higher-frequency one?

AR3.10:

Firstly, we agree with the reviewer that 5.5 MHz was not optimal for imaging resolutions, but the choice of the current transducer was based on a compromise among image resolution, detection sensitivity, field of view, scanning time, and system cost. During system design, these parameters exhibit a trade-off, with each influencing the other.

Secondly, as discussed in our response to RC3.4 of the Reviewer, PATTERN provides circuitry-level resolution for brain imaging. However, achieving resolution down to the cellular level is currently constrained by the limitations of commercially available transducer arrays, as illustrated in Figure R23. Moreover, the attenuation of high-frequency ultrasound in brain tissue prevents the attainment of cellular-level resolution, even with the use of ultrahigh-frequency transducers. Consequently, additional theoretical breakthroughs are necessary to propel photoacoustic computed tomography into higher levels of image resolution that hold significance for neuroscience.

Revisions:

The possibility of utilizing higher frequency transducers was discussed in the Discussion section (page 26 lines 21–28):

“Firstly, faster lasers or potentially, some deep-learning-based image fusion approaches⁵⁸ can be employed to accelerate the imaging process, while the imaging resolution can be improved by using ultrasound transducers with larger bandwidths. Specifically, the utilization of transducers with a frequency response ranging from direct current (DC) to 22 MHz has demonstrated the ability to achieve a resolution of approximately $50\ \mu\text{m}$ ⁵⁷, which is comparable to the resolution achieved in all-optical, large-FOV brain imaging⁵².”

RC3.11:

- PATTERN: Which are the optical/absorption properties of the body fluid used in the perfusion module? Would its presence affect the signals measured in the brain? And how?

AR3.11:

The fluids utilized in our experiments included Phosphate Buffered Saline (PBS), artificial Cerebrospinal Fluid (aCSF), and Paraformaldehyde (PFA). All these fluids are colorless and transparent. PBS and aCSF were also employed in the imaging chamber and sample holder to facilitate acoustic coupling. Notably, no significant signals attributable to these fluids were observed, indicating that they did not interfere with the photoacoustic signals of the brain. PFA was used primarily for fixing the brain in experiments involving tissue clearing and fMOST techniques (Figs. 1d, 2c, 3a, 3g-j, 3k,4b-d, 3f-g, 5b-f, 5i-j, 5k-m, Supplementary Figs. 12, 13, 14, 15, 16, 17, 18a-e). In terms of PA imaging, brains perfused with PFA showed no discernible differences compared to those perfused with PBS or aCSF, suggesting that PFA also does not impact the PA signal.

RC3.12:

- Data reconstruction and postprocessing: Could you provide some more information on the light fluence simulation and its compensation scheme, please?

AR3.12:

Computer simulation and optical fluence compensation were used for image rendering. Initially, the image features, such as a mouse brain along with the agarose-based sample holder, were segmented using Amira software. Subsequently, binary masks of the brain and holder were modeled using reasonable optical parameters (brain: $\mu_a = 0.005$ and $g = 0.9$; agarose: $\mu_a = 0.002$ and $g = 0.98$) in a Monte Carlo simulation for light fluence computation. We used a uniform light source covering the inner diameter of the sample holder as in real experiments. Such simulation was performed using MCXLAB³². The resulting light fluence was then utilized to compensate for the optical attenuation within the brain tissue by dividing both the PA image and the PATTERN image. The compensated results were rendered using Amira. The codes for the simulation are listed as below:

```

load([datapath2,'simulate_label_brain.mat']); % load labels of brain and holder
vol = uint8(simulate_label_brain);

cfg.vol = vol;
cfg.unitinmm=19.2/256; % set the grid size to be the same with the images
cfg.prop=[0.002,1,1,1.37; % water
          0.002,20,0.98,1.37; % agar
          0.005,20,0.9,1.37; % brain
          ]; % standard tissue

cfg.nphoton=1e9;
cfg.issrcfrom0=1;
cfg.srcpos=[128 1 128]; % center of the beam
cfg.tstart=0;
cfg.tend=5e-9;
cfg.tstep=5e-9;

% set light source
cfg.srcdir=[0 1 0];
cfg.srctype='disk';
cfg.srcparam1=[106 0 0 0]; % beam radius of 7.95 mm
cfg.isreflect=0;
cfg.autopilot=1;
cfg.gpuid=1;
cfg.debuglevel='P';

cfg.outputtype='energy';
flux=mcxlab(cfg);

```

Revisions:

The code was added to supplementary methods and referred in the Methods section (page 40 lines 14–22):

“Subsequently, a 3D Monte Carlo simulation, conducted using MCXLAB⁶⁴ (Supplementary Methods), was employed to simulate the light fluence distribution, considering $\mu_a = 0.005$ and $g = 0.9$ for the brain, and $\mu_a = 0.002$ and $g = 0.98$ for the sample holder (agarose). We utilized a uniform light source covering the inner surface of the sample holder to replicate the experimental conditions. The resulting light fluence distribution was then applied to compensate for the optical attenuation within both the PA image (background) and the PATTERN image (fluorescent signal).”

RC3.13:

- DnCNN Method: How did you train the neural network regarding the artefacts? Did you manually delineate them? How did you ensure that no information was taken as an artefact?

AR3.13:

As mentioned by the reviewer, we manually identified artefacts in photoacoustic images to build up datasets for DnCNN training. Our approach, as depicted in Figure R 30, mainly involved the combination of PA fluorescent signals and confocal fluorescent imaging, as well as the human experts' identification. Specifically, each brain sample subjected to PATTERN processing was also imaged by confocal microscopy. The presence of EGFP (for confocal imaging) and iRFP713 (for PATTERN imaging) signals within the same brain regions allows the alignment and registration of each confocal image to the PATTERN image. This alignment served as a template for human experts to accordingly label artefact regions within the PATTERN data. To ensure accuracy, only signals evident in the PATTERN images but absent in the reference were designated as artefacts, mitigating potential mislabeling. In addition, this labeling method is able to avoid the generation of false positives, yet there still exists the potential for unlabelled artefacts, which can be further improved. Therefore, we performed a method based on DnCNN, utilizing the capabilities of artificial neural networks to remove all artefacts as effectively as possible. Subsequently, artefact-removed images and their corresponding originals were paired to form the dataset used for training the DnCNN. Moreover, we have included information of DnCNN training in the revised method. For the sake of transparency and reproducibility, all codes and the pre-trained network employed in this study have been made available at <https://github.com/CaA2318777/PATTERN>

Figure R 30 | Working flowchart of DnCNN performance and artefact identification.

a, Illustration of the workflow of DnCNN. Datasets are constructed in the training phase based on confocal microscopy as the reference. Network after training is used in the evaluation phase to remove the artefacts. **b**, Pipeline of artefact identification. Images with artefacts and the artefacts-removed ones were collected as datasets.

Revisions:

The above illustration was included as Supplementary Fig. 9b in the revised manuscript (page 70) and the clarification was included in the Methods section (page 45 lines 17–24 and page 46 lines 9–20):

Page45:

"The brain samples were first subjected to PATTERN to collect the iRFP713 signal prior to sectioning and confocal microscopy to collect the EGFP signal. The two modalities were then equally normalized. Well-trained human experts would subsequently manually annotate the artefacts in the PATTERN signals, based on the results of confocal microscopy, which serve as the reference. In addition, this labeling method is able to avoid the generation of false positives, yet there still exists the potential for unlabelled artefacts, which can be further improved."

Page46:

"The neural network implicitly learned the artefacts of each image to achieve the distribution of the artefacts across the entire dataset. The Mean Squared Error (MSE) loss function was employed as the training criterion over 50 epochs, during which the network effectively filtered out pure artefacts to generate the output. For each brain region, a dataset consisting of more than 100 pairs of images with and without artefacts were used for training and more than 50 pairs for testing. Some of the representative hyperparameters used during training are provided below: batch size = 1, training epochs = 50, and learning rate = 1e-3. When the epoch reached 30, the learning rate was reduced by a factor of 10. To make the network more generalized, traditional data augmentation strategies were also applied, including flipping, rotating, and intensity changes."

Reference

- 1 Michels, R., Foschum, F. & Kienle, A. Optical properties of fat emulsions. *Optics Express* **16**, 5907–5925 (2008). <https://doi.org:10.1364/OE.16.005907>
- 2 Jathoul, A. P. *et al.* Deep in vivo photoacoustic imaging of mammalian tissues using a tyrosinase-based genetic reporter. *Nature Photonics* **9**, 239 (2015).
- 3 Pan, C. *et al.* Shrinkage-mediated imaging of entire organs and organisms using uDISCO. *Nature Methods* **13**, 859–867 (2016). <https://doi.org:10.1038/nmeth.3964>
- 4 Matsumoto, K. *et al.* Advanced CUBIC tissue clearing for whole-organ cell profiling. *Nature Protocols* **14**, 3506–3537 (2019). <https://doi.org:10.1038/s41596-019-0240-9>
- 5 Jing, D. *et al.* Tissue clearing of both hard and soft tissue organs with the PEGASOS method. *Cell Research* **28** (2018). <https://doi.org:10.1038/s41422-018-0049-z>
- 6 Tomer, R., Ye, L., Hsueh, B. & Deisseroth, K. Advanced CLARITY for rapid and high-resolution imaging of intact tissues. *Nature Protocols* **9**, 1682–1697 (2014).

- <https://doi.org:10.1038/nprot.2014.123>
- 7 Ragan, T. *et al.* Serial two-photon tomography for automated ex vivo mouse brain imaging. *Nature methods* **9**, 255–258 (2012). <https://doi.org:10.1038/nmeth.1854>
- 8 Seiriki, K. *et al.* High-Speed and Scalable Whole-Brain Imaging in Rodents and Primates. *Neuron* **94**, 1085–1100.e1086 (2017). <https://doi.org:10.1016/j.neuron.2017.05.017>
- 9 Zhong, Q. *et al.* High-definition imaging using line-illumination modulation microscopy. *Nature Methods* **18**, 309–315 (2021). <https://doi.org:10.1038/s41592-021-01074-x>
- 10 Cui, M. *et al.* Adaptive photoacoustic computed tomography. *Photoacoustics* **21**, 100223 (2021).
- 11 Chen, Y. *et al.* Photoacoustic Mouse Brain Imaging Using an Optical Fabry-Pérot Interferometric Ultrasound Sensor. *Frontiers in neuroscience* **15**, 572 (2021).
- 12 Lin, L. *et al.* High-speed three-dimensional photoacoustic computed tomography for preclinical research and clinical translation. *Nature communications* **12**, 1–10 (2021).
- 13 Cao, R. *et al.* Single-shot 3D photoacoustic computed tomography with a densely packed array for transcranial functional imaging. arXiv:2306.14471 (2023). <<https://ui.adsabs.harvard.edu/abs/2023arXiv230614471C>>.
- 14 Lu, T. *et al.* Full-frequency correction of spatial impulse response in back-projection scheme using space-variant filtering for photoacoustic mesoscopy. *Photoacoustics* **19**, 100193 (2020). <https://doi.org:https://doi.org/10.1016/j.pacs.2020.100193>
- 15 Xia, J. *et al.* Three-dimensional photoacoustic tomography based on the focal-line concept. *Journal of Biomedical Optics* **16**, 090505–090505–090503 (2011). <https://doi.org:10.1117/1.3625576>
- 16 Seeger, M. *et al.* Pushing the boundaries of photoacoustic microscopy by total impulse response characterization. *Nature Communications* **11**, 2910 (2020). <https://doi.org:10.1038/s41467-020-16565-2>
- 17 Van de Sompel, D., Sasportas, L. S., Jokerst, J. V. & Gambhir, S. S. Comparison of Deconvolution Filters for Photoacoustic Tomography. *PLoS ONE* **11** (2016).
- 18 Zhang, J. *et al.* In vivo characterization and analysis of glioblastoma at different stages using multiscale photoacoustic molecular imaging. *Photoacoustics* **30**, 100462 (2023).
- 19 Yao, J. *et al.* Multiscale photoacoustic tomography using reversibly switchable bacterial phytochrome as a near-infrared photochromic probe. *Nature Methods* **13**, 67–73 (2016). <https://doi.org:10.1038/nmeth.3656>
- 20 Margrie, T. W. *et al.* Targeted Whole-Cell Recordings in the Mammalian Brain In Vivo. *Neuron* **39**, 911–918 (2003). <https://doi.org:https://doi.org/10.1016/j.neuron.2003.08.012>
- 21 Gottschalk, S. *et al.* Rapid volumetric photoacoustic imaging of neural dynamics across the mouse brain. *Nature biomedical engineering* **3**, 392–401 (2019).
- 22 Li, L. *et al.* Small near-infrared photochromic protein for photoacoustic multi-contrast imaging and detection of protein interactions in vivo. *Nature communications* **9**, 2734 (2018).
- 23 Li, X. *et al.* Three-dimensional structured illumination microscopy with enhanced axial resolution. *Nature Biotechnology* **41**, 1307–1319 (2023). <https://doi.org:10.1038/s41587-022-01651-1>
- 24 Olefir, I. *et al.* Spatial and spectral mapping and decomposition of neural dynamics and organization of the mouse brain with multispectral photoacoustic tomography. *Cell*

- Reports* **26**, 2833-2846. e2833 (2019).
- 25 Dawson, G. Measuring brain lipids. *Biochimica et Biophysica Acta (BBA) - Molecular and Cell Biology of Lipids* **1851**, 1026-1039 (2015).
[https://doi.org:https://doi.org/10.1016/j.bbalip.2015.02.007](https://doi.org/https://doi.org/10.1016/j.bbalip.2015.02.007)
- 26 Davoudi, N., Deán-Ben, X. L. & Razansky, D. Deep learning optoacoustic tomography with sparse data. *Nature Machine Intelligence* **1**, 453-460 (2019).
<https://doi.org:10.1038/s42256-019-0095-3>
- 27 Allouche-Arnon, H. *et al.* Computationally designed dual-color MRI reporters for noninvasive imaging of transgene expression. *Nature Biotechnology* **40**, 1143-1149 (2022). <https://doi.org:10.1038/s41587-021-01162-5>
- 28 Chen, Z. *et al.* Hybrid magnetic resonance and optoacoustic tomography (MROT) for preclinical neuroimaging. *Light: Science & Applications* **11**, 332 (2022).
<https://doi.org:10.1038/s41377-022-01026-w>
- 29 Machado, T. A., Kauvar, I. V. & Deisseroth, K. Multiregion neuronal activity: the forest and the trees. *Nature Reviews Neuroscience* **23**, 683-704 (2022).
<https://doi.org:10.1038/s41583-022-00634-0>
- 30 Lichtman, J. Imaging the Connectome. *Biophysical Journal* **108**, 23a (2015).
<https://doi.org:https://doi.org/10.1016/j.bpj.2014.11.148>
- 31 Abbott, L. F. *et al.* The Mind of a Mouse. *Cell* **182**, 1372-1376 (2020).
<https://doi.org:https://doi.org/10.1016/j.cell.2020.08.010>
- 32 Fang, Q. & Boas, D. Monte Carlo Simulation of Photon Migration in 3D Turbid Media Accelerated by Graphics Processing Units. *Optics express* **17**, 20178-20190 (2009).
<https://doi.org:10.1364/OE.17.020178>

REVIEWERS' COMMENTS:

Reviewer #1 (Remarks to the Author):

This paper presents a novel implementation of photoacoustic tomography (PAT) that relies on temporal encoding instead of spectral encoding to achieve molecular imaging using exogenous chromophores (fluorophores). The idea of temporal encoding aided by deliberate photobleaching of fluorophores is elegantly simple and can aid in molecular and function photoacoustic (PA) imaging of a variety of readily available fluorophores provided that produce a strong enough PA signal and that the photobleaching is fast enough. The authors have prepared a high-quality manuscript with extensive data using multiple animal models to demonstrate the strengths of their new approach including imaging with genetically encoded probes. All of my critiques and suggestions have been adequately addressed and I recommend this manuscript for publication in its current form.

Reviewer #2 (Remarks to the Author):

The authors have adequately addressed all of my comments and suggestions, with new experimental data and comprehensive discussions. I believe the proposed PATTERN method can be broadly applied to many neuroscience research, particularly for molecular imaging. I highly recommend the publication of this fine work in Nat Communications.

Response to the Editor

Dear Editor,

We sincerely appreciate the efforts of the editorial team and reviewers.

The point-by-point responses to the reviewer's comments are as follows:

Point-by-point response to the reviewers' comments

Reviewer #1 :

“This paper presents a novel implementation of photoacoustic tomography (PAT) that relies on temporal encoding instead of spectral encoding to achieve molecular imaging using exogenous chromophores (fluorophores). The idea of temporal encoding aided by deliberate photobleaching of fluorophores is elegantly simple and can aid in molecular and function photoacoustic (PA) imaging of a variety of readily available fluorophores provided that produce a strong enough PA signal and that the photobleaching is fast enough. The authors have prepared a high-quality manuscript with extensive data using multiple animal models to demonstrate the strengths of their new approach including imaging with genetically encoded probes. All of my critiques and suggestions have been adequately addressed and I recommend this manuscript for publication in its current form.”

Response to reviewer #1:

We sincerely thank the constructive suggestion and insightful comments from the reviewer.

Reviewer #2 :

“The authors have adequately addressed all of my comments and suggestions, with new experimental data and comprehensive discussions. I believe the proposed PATTERN method can be broadly applied to many neuroscience research, particularly for molecular imaging. I highly recommend the publication of this fine work in Nat Communications.”

Response to reviewer #2:

We sincerely thank the constructive suggestion and insightful comments from the reviewer.